

# Properties of biomass burning aerosol mixtures derived at fine temporal and spatial scales from Raman lidar measurements: Part I optical properties

Lucja Janicka[1], Iwona S. Stachlewska[1]

[1]University of Warsaw, Faculty of Physics, Institute of Geophysics, Warsaw, 02-093, Poland

*Correspondence to*: Lucja Janicka (lucja.janicka@igf.fuw.edu.pl)

**Abstract.** The analysis of the aerosol optical properties derived at fine temporal and spatial scales were performed based on measurements obtained during heat wave event in vicinity of a cold weather front in Warsaw on August 9[th]-11[th], 2015. The signals collected by the PollyXT-UW lidar allowed for the calculation of 23 sets of so-called 3β+2α+2δ+wv profiles

averaged by 30-minutes periods during 2 nights. The total number of 11 different aerosol types and aerosol mixtures were identified with reference to properties within 116 sub-layers in the profiles and were characterized by the mean values. The statistical sample of various optical properties being in agreement for consecutive profiles allowed to assess the spatio-temporal extent of aerosol/mixture types. The mean lidar ratio values of 53-73 sr (355 nm) and 31-45 sr (532 nm) in the layers dominated by the anthropogenic pollution were found. For the layers dominated by the biomass burning aerosol

(fresh, moderately fresh, moderately aged) mean lidar ratio was of 69-114 sr (355 nm) and 57-85 sr (532 nm). The colour ratio of lidar ratio (532/355) higher than 1, characteristic for aged biomass burning aerosol, was found only in one scattered layer, accompanying with low value of extinction related Ångström exponent of 0.60±0.32 and low particle depolarization ratio. The maximum of the particle depolarization ratio of 4.8-5.0 % at 532 nm were observed in a layer likely contaminated with pollen and in a layer dominated by fresh biomass burning aerosol. This study provides an excellent data set for

exploration of separation algorithms, aerosol typing algorithms and microphysical inversion.

**Keywords:** mixtures of aerosol, biomass burning aerosol, lidar

## 1 Introduction

Lidar measurements, as they provide vertical profiles of the optical properties in the atmosphere, are an excellent tool for studying of atmospheric aerosols. This research is extremely important in context of the aerosol impact on the earth radiative

budget (Stocker et al., 2013). Small particles suspended in the atmosphere affect the climate system directly by scattering or absorbing solar radiation and indirectly by changing microphysics of clouds forming in the presence of aerosols. The aerosol effect depends on the particles type and the earth surface properties, over which they occur, thus the related radiative forcing can be both positive or negative. Moreover the altitude of aerosol layer suspension in the atmosphere is relevant, especially in the case of these containing black carbon (Zarzycki and Bond, 2010; Samset et al., 2013), which is the main aerosol



component absorbing solar radiation. Absorbing aerosols cause heating at the altitude of layer presence and cooling below, what leads to an increase of the atmospheric stability (Menon et al., 2002; Babu et al., 2011). The main aerosol species containing significant amount of black carbon, produced in the process of incomplete combustion, are biomass burning aerosol (BBA) (Eck et al., 2003; Reid et al., 2005; China et al., 2013) and anthropogenic pollution from combustion of fossil and bio fuels (Johnson et al., 2005; Schwarz et al., 2008). Nevertheless the composition of individual aerosol released during specific burning event is variable and depends on the burning process and the burnt material. Smoldering combustion in the natural habitats produce more soot than the flaming one (Dubovik et al., 2002; Pósfai et al., 2003). Fresh biomass burning aerosol consist of the organic (brown) carbon, black carbon and inorganic particles. After the emission, biomass burning aerosol is the subject of an intense aging processes (Nicolae et al., 2013; Li et al., 2015) which lead to the chemical, microphysical and optical evolutions of the particles, thus the estimation of the BBA impact on the radiative budget is a very complex task. The fact, that in a large number of cases the observed aerosol is a mixture of different types found in various proportions (Clarke et al., 2004; Hara et al., 2017) gives an additional challenge to the atmospheric aerosol study.

In recent years development of lidar techniques yield in the capability of high quality measurements which allow characterization of an observed aerosol by sounding the atmosphere in several wavelengths and measuring a signal depolarization by unspherical particles. Using the theory of Raman scattering, providing independent information of the aerosol extinction, gives the opportunity of a reliable intensive optical properties retrieval. In the literature one can find a variety of analysis of different aerosol types properties observed during the special cases of advection (Ortiz-Amezcua et al., 2017; Ritter et al., 2018), measurements campaigns (Tesche et al., 2011; Groß et al., 2015; Chazette et al., 2016) and also the climatological studies at specific stations or within the research network, where the standard requirement of lidar profiles should be fulfilled (Amiridis et al., 2005; Sicard et al. 2011; Marinou et al., 2017). In many studies the lidar measurements are assisted by the other instruments to increase the research quality and reliability. Nevertheless, the relations between the intensive aerosol optical properties derived from multiwavelength Raman-depolarization lidar measurements have the potential of independent aerosol typing (Burton et al., 2012; Groß et al., 2013; Voudouri et al., 2018). The automatic aerosol typing algorithms operate with the satellite-based lidar CALIPSO (Omar et al. 2009), CATS (Proestakis et al., 2019) and EarthCARE (Illingworth, et al., 2015). Recent studies present the algorithms for automatic typing of the aerosols observed by the lidar using an artificial neural network (Nicolae et al., 2018) and the Mahalanobis distance function (Papagiannopoulos et al., 2018). The typing schemes can provide maximum number of 14 aerosol types/mixtures as an output, in the case of the neural network, and up to 8 aerosol classes in the case of the Mahalanobis distance. This constitute a very convenient and useful tools for the climatological analysis. However, to use the lidar data by the specialist in the radiative forcing and climate modelling calculations, the microphysical aerosol parameters are needed. The algorithms that deal with the inversion problem of microphysical retrieval require an accurate aerosol layer selection and a high quality $3\beta+2\alpha$ optical data as well as the depolarization information (Veselovskii et al., 2002; Böckmann et al., 2005; Müller et al., 2016). Thus the manual data evaluation which allows for an insightful analysis of lidar signals with the individual approach to the considered case is still much-needed.





In the vicinity of places where huge events of aerosol release happen high aerosol load and pronounced layers occur and therefore create good conditions for analyses of pure aerosol types and keep the retrieval uncertainties small. A well defined aerosol plumes originated from huge emissions can be distributed over a long distances by reaching the high troposphere (Mamouri et al., 2016) or even the stratosphere (Haarig et al., 2018; Hu et al., 2018). An issue of the pure and long range

transported aerosols properties retrieval was the subject of various measurement campaigns and found in the systematic observations (Freudenthaler et al., 2009; Groß et al., 2015; Ortiz-Amezcua et al., 2017). When the aerosol emissions are small, the plumes strongly evolve during the transport in the atmosphere, including mixing with the another particles, and therefore do not represent the distinct layers. In these cases the evaluation of optical properties becomes a challenge. The heat waves create a favourable conditions of multiple, different type, aerosols layers formation which are advected or locally

emitted as pollution, smoke from uncontrolled fires of crops and natural ecosystems or pollen (Stachlewska et al., 2018). Not only the heat waves become more intense in recent years (Della-Marta et al., 2007; Tomczyk and Bednorz, 2016), moreover, the heat wave in 2015 was one of the most persistent events (Wibig, 2018). Thus, the sensitive study of such kind of complicated aerosol load is required.

The presented study refers to the August 2015 heat wave. High aerosol optical thickness exceeding 0.63 according with the

unusually elevated boundary layer height above 2.5 km above the station in the southern Poland during this event were reported by Szkop and Pietruczuk, 2017. The research were conducted based on the synergy of ceilometer and sunphotometer data, supported by the newly developed scheme for backward air mass trajectory analysis to recognize type and source of aerosol layers. In the period of August 6th-13th, 2015, which coincide with the analyzed in this paper, the aerosol were recognized as the biomass burning one form the Ukrainian wildfires. The preliminary results for the case of

August 9th-11th, 2015 by Janicka et al., 2018 showed the complexity of this case, which was the motivation for deepening study in the fine temporal and spatial scales to reflect atmospheric variability and to 'catch' all individual sub-layers observed by the multiwavelength lidar. Assuming that the observed aerosols can be likely kinds of mixtures, the obtained impressive set of optical parameters in multiple selected sub-layers can act as the perfect test-bed for the separation algorithms (Mamouri and Ansmann, 2017).

The uniqueness of this study is the statistical approach for the aerosol properties evaluation, as the considerable amount of about 100 sub-layers, evaluated for 2 nights, with full set of optical parameters were taken into account in the analysis. The specific aerosol sub-layers are described by the numerous, consistent sets of optical data. The aerosol optical properties in particular discriminated layers classified as the one aerosol or mixture type stand as the mean value of the ones evaluated separately in each sub-layer. The mentioned method is considered as more appropriate for the aerosol determination than the

evaluation of only a few profiles during an event, performed as a standard in lidar community. For such analysis the continuous 24/7 lidar measurements with multiple detection channels are required. In presented study the PollyXT-UW 8-channel Raman-polarization and water vapour lidar that provides $3\beta+2\alpha+2\delta+wv$ set of optical profiles was used. High quality of the obtained profiles is provided as the EARLINET ACTRIS quality assurance (QA; Freudenthaler et al., 2018) and quality control (QC; ref) procedures are in place.



The paper is organized as follows. In section 2, an overview of lidar technical aspects is given. Section 3 contain the retrieval and analysis methodology. The general meteorogical situation during the considered days, the trajectory analysis and the analysis of possible aerosol sources are given in Section 4. In Section 5 the aerosol optical properties analysis and the discussion is presented. The summary of the findings and outlooks potential use of the obtained results are contained in Conclusions.

## 2 Instrument

The ground-based quasi-continuous 24/7 zenith-aiming lidar measurements were performed at the Remote Sensing Laboratory (RS-Lab) of the Institute of Geophysics at the Faculty of Physics of the University of Warsaw, Poland (52.21ºN, 20.98ºE, 112 m a.s.l.). High quality measurement data are obtained, as the EARLINET ACTRIS quality assurance (QA) and the quality control (QC) procedures are performed at lidar station (Pappalardo et al., 2014; Freudenthaler et al., 2018). The PollyXT-UW lidar (Engelmann et al., 2016) is equipped with the Nd:Yag laser, emitting laser pulses of 180, 110, 60 mJ at 1064, 532, 355 nm, respectively, with the repetition rate of 20 Hz. The laser beams are expanded and sent coaxially into the atmosphere (the beam diameter at the output mirror is 45 mm and the beam divergence is 0.2 mrad). The collection receiver consist of the Newtonian telescope with 300 mm primary mirror and 0.9 mm pinhole resulting in the field of view of 1 mrad. The signal detection is performed in 8 channels. The elastic backscattered radiation is detected in three wavelengths (355, 532, 1064 nm), for the UV and VIS also elastic cross-polarized scattering and non-elastic vibrational Raman scattering on atmospheric $N_2$ is measured (355s, 532s, 387, 607 nm). Additionally, for the UV, vibrational Raman scattering on $H_2O$ is recorded at 407 nm. For all channels photon-counting detection is used with Hamamatsu H10721P-110 photomultipliers (dead time of ~2 ns), except for the 1064 nm wavelength where extra-cooled Hamamatsu R3236 photomultiplier is used. The signals are recorded with 7.5 m height resolution up to altitude of 48 km using the 600 MHz counters.

## 3 Methodology

The sets of the optical properties profiles ($3\beta+2\alpha+2\delta+wv$), i.e. particle backscatter coefficient ($\beta$), particle extinction coefficient ($\alpha$), particle depolarization ratio ($\delta$) and water vapour mixing ratio (wv) with the related errors were obtained using the VerlaufNG7.vi software provided by TROPOS (Leibniz Institute of Tropospheric Research) in the frame of the PollyNET. Details on the evaluation procedures can be found in Baars et al., 2016 for aerosol optical properties and in Stachlewska et al., 2017a for water vapour mixing ration and relative humidity. The sets of the optical properties profiles were calculated for each consecutive half an hour averaging time from 19 UTC to 02 UTC on 9[th]/10[th] and 10[th]/11[th] of August 2015 (Fig. 1; red rectangles mark the periods chosen for analyses). Due to passing cumulus clouds, which appeared during the first night, in one period the time averaging was modified for 19:15-19:45 UTC, August 9[th], and four periods were



excluded from the analysis (22:00-23:00 UTC, August 9[th]; 00:30-01:30 UTC, August 10[th]). On the second night, one period was shortened in order to capture a brief interval of significantly varying meteorology conditions (21:17-21:25 UTC, August 10[th]). Finally, 9 sets of profiles on 9/10 of August 2015 and 14 sets of profiles on 10/11 of August 2015 were obtained. For retrieval of the aerosol extinction ($\alpha$) at 355 and 532 nm and backscatter at 355, 532, 1064 nm ($\beta$) coefficients profiles, the

classical Raman approach was used (Ansmann et al., 1990). Nighttime measurements were chosen for the evaluation since the signal to noise ratio (SNR) of daytime measurements was not sufficient for Raman unlike in other studies in Warsaw, where the daytime Raman profiles were successfully used for the microphysical retrieval (Janicka et al., 2017). The particle linear depolarization ratios ($\delta_{par}$) at 355 and 532 nm were obtained using the classical $\pm 45^{o}$ calibration method (Freudenthaler et al., 2016). No overlap correction was applied to the profiles since the focus of the study is on the aerosol layers in free

troposphere (above full overlap range). The profiles were evaluated with 7.5 m height resolution and vertical smoothing with running mean over 112.5 m applied on the raw lidar signals. The pressure, temperature and humidity profiles used for calculations and calibration were obtained from radiosounding (RS) data from Legionowo WMO 12374 station (52.40ºN, 20.96ºE, 96 m a.s.l., ~25 km North from Warsaw). The columnar value of the extinction related Ångström exponent of 1 was assumed for the extinction coefficient profiles calculation. An additional smoothing procedure was applied to the latter. The

height resolution in the pre-smoothed extinction profiles was reduced to 150 m in four cycles starting from 0, 37.5, 75, 112.5 m. The original 7.5 m height resolution of the profiles was preserved. In the next step all four profiles were averaged and finally smoothed with the running mean of 450 m averaging window and the flat base function starting from the altitude of 750 m. The smoothing with narrower averaging windows has been tested and also used for the layers discrimination. Comparing of the results obtained with different averaging windows showed the consistency, but due to lower variation, the

profiles smoothed with 450 m window were used for calculation of the mean extinction coefficient values in the selected layers. The multiple aerosol sub-layers were discriminated in the sets of optical properties profiles, mainly based on the extinction coefficient and relative humidity (RH) data. In total 115 aerosol layers on the first night and 72 on the second night were identified, whereof 69 aerosol layers of the first and 44 of the second night were taken into further analysis of the intensive optical properties because at the altitude above 3.4 km on the first night and above 2 km on the second night there

was no complete sets of all intensive and extensive properties. The minimum and the maximum thickness of the sub-layers selected in the analysis were of 89 and 350 m respectively.

The intensive aerosol properties: lidar ratio (LR=$\alpha/\beta$), colour ratio of lidar ratios (CRLR=$LR_{532}/LR_{355}$), extinction related Ångström exponent (Å$E_{355/532}$=-ln($\alpha_{355}/\alpha_{532}$)/ln(355/532)), backscattering related Ångström exponent (Å$B_{355/532}$=-ln($\beta_{355}/\beta_{532}$)/ln(355/532), Å$B_{532/1064}$=-ln($\beta_{532}/\beta_{1064}$)/ln(532/1064)) were calculated using the mean backscatter and extinction

coefficients values in the selected sub-layers. The related uncertainties were derived using the error propagation.

The aerosol optical depth (AOD) was integrated with respect to the extinction coefficient profiles at 355 and 532 nm in the range from the height of 1.2 km (above the boundary layer) up to the end of the pronounced aerosol layers occurrence (4 km for the night of 9/10 August, 2.5 km for the night of 10/11 August). The aerosol boundary layer height and residual layer height (red and black dots in Fig. 1, respectively) were calculated with the wavelet method, as in Wang et al., 2018. The



purple bars in Fig. 1 are the boundary layer height estimated from the RS from Legionowo. Clearly, for a complicated, unusual aerosol composition on both analyzed nights the aerosol layers up to 2 km are recognized by the wavelet algorithm as the residual layer. The lowermost analyzed layer at about 1 km on August 10[th] was not taken into the AOD calculation as it was on the top part of the boundary residual layer, relying on the RS estimation. Therefore the lidar derived AOD values

are representative for the observed free troposphere aerosols (Table 2).

## 4 Background of the event

### 4.1 General meteorogical situation

On the beginning of August 2015, persistent high pressure system located over Central and Eastern Europe controlled the weather situation and caused the heat wave over Europe (Tomczyk and Bednorz, 2016; Stachlewska et al., 2017b; Wibig,

2018). Inflow of the warm air from over Western Sahara was dominating in the troposphere. Extremely high temperatures were recorded at numerous countries of Europe (Janicka et al., 2018). During the analyzed period from 9[th] to 11[th] August 2015, above the territory of Poland the quasi-stationary wavy weather front was located from the south-western to the north-eastern border of Poland. The hot tropical air-masses were on the South side of the front line and the polar cooler air-masses on the North side. Within the three days the front line insensitively moved, taking finally the latitudinal direction in the

middle of Poland territory. In the three days period, the weather front line was at the direct neighbourhood of the lidar site in Warsaw. The weather front line is well visible in Fig. 2 panel (a) on the synoptic map of August 10[th], 2015, 12 UTC (more maps are available via www.pogodynka.pl). The spatial distribution of the AOD at 550 nm from the MODIS detector (accessed via www.polandaod.pl site), shown in Fig. 2 panel (b), depict distinct difference in the AOD between the northern Poland, with the value of about 0.05, and the southern Poland with AOD values up to 0.5, splitted by the front line. The

dynamical weather condition near the front can contribute to the advection of complicated aerosol structures (Zhang et al., 2009, Janicka et al., 2017, Stachlewska et al., 2018). Weather fronts are usually related to specific types of clouds occurrence. However, during the days of 9[th]-11[th] August 2015, due to the air settlement in the conditions of high pressure system, intensive formation of clouds was braked, which allowed for continuous lidar measurements of aerosol structures.

### 4.2 Air-mass transport analysis

The analysis of the possible directions of the aerosols inflow over Warsaw is based on the 6-days HYSPLIT backward trajectories (Stein et al., 2015; Rolph et al., 2017). The simulations were performed using the meteorogical data from the reanalysis, as well as the GDAS model. The comparison of the simulations with the same input data concerning the time and the altitude but performed with different meteorogical data show evident discrepancies between the results. In Fig. 3, the results of the 6-days HYSPLIT backward trajectories are shown. The simulations have been performed for Warsaw location

for the altitudes from 1.2 km to 5.5 km with the spatial interval of 0.3 km for 19 UTC on August 9[th] and for 02 UTC on August 10[th] and also from 1.2 km to 2.7 km with the same high interval for the following night - 19 UTC on August 10[th] and




UTC on August 11[th] to reflect the air mass transport for the altitude range and the time periods in which the analysis of the optical properties have been performed. During the night of 09/10 August 2015 in the low troposphere from 1.2 to 1.8 km general path of the inflow over Warsaw ran starting on the territory of Romania and Hungary and further northward to western Poland and finally latitudinally toward Warsaw. Within the analyzed time of 7 hours the trajectories at mentioned

altitudes changed only a little, becoming more divergent with time. At the altitudes from 2.1 km to 3 km on August 9[th], 19 UTC trajectories have evolved with height from the mentioned before direction of the lower trajectories to the direction from western Europe, circulating above Czech Republic before reaching Warsaw. Over the time, at 02 UTC on August 10[th] air mass advection at 2.7 and 3 km changed to the one deriving from over Spain. Lower, at the altitudes of 2.1 and 2.4 km the trajectories moved westward. At the altitudes of 3, 3.5 and 4 km in the evening on August 9[th] the trajectories passed over

Spain and Portugal, however the two uppermost ran faster and source from over Canary Islands thus it is possible that can have some contamination of mineral particles from Sahara Desert. Above the altitude of 5 km the air originate from Canada and Greenland. Later, at 02 UTC on August 10[th] the arctic air was advected over Warsaw at the lower altitude of 3.5 to 4 km. Comparing the results for 02 UTC on August 10[th], obtained with the reanalysis and the GDAS model meteorogical data, one can see consistency of the results with reference to both trajectories paths and altitudes only in the case of the reanalysis data.

The altitudes of the trajectories obtained using the GDAS model are very chaotic, moreover in some cases they move directly along the ground. The patches of the trajectories also look less reliable due to its discrepancy and meandering shape. The important difference between the two simulation is origin place of aerosols suspended in the atmosphere over Warsaw at about 3 km. The simulation with reanalysis data point as a potential biomass burning source wildfires in Portugal and Spain, while at the same time the simulation with GDAS model point Ukrainian territory.

The 6-days HYSPLIT backward trajectories calculated for the following night do not vary significantly between the time of 19 UTC on August 10[th] and 02 UTC on August 11[th]. The trajectories were calculated at the analogous altitudes as for the previous night, however only for the maximum height of 2.7 km, as above this altitude no aerosol layers were observed. The patch of the lowermost trajectories at 1.2 and 1.4 km reminds the ones observed at the analogous altitudes on the night before.  At the altitude of 1.8 km, the beginning of the trajectory is shifted to West, as it was on the previous night at higher

altitudes of  about 2.1 - 2.4 km. Above the altitude of 2.1 km the trajectories, also as the previous night, come from Portugal and northern Atlantic Ocean, even Greenland. Comparing the trajectories obtained with different meteorogical data for 02 UTC on August 11[th] we can see again high variability in the trajectories height profile in simulation with the GDAS model. The course of the trajectories with using GDAS model mimic the ones with reanalysis data, but also, as it was observed earlier, trajectory at 1.8 km with GDAS model point Ukraine as the place of origin of the air masses, while in the simulation

with reanalysis data Western Europe is showed. Based on the above analysis we decided to rely on the simulations with the reanalysis meteorological data in this particular study dealing with the dynamical conditions related to the weather front. However, taking into account difficult meteorological situation the GDAS model performs surprisingly well in comparison with reanalyses. However, we need as precise indication as possible thus we chose the latter.





### 4.3 Aerosol source analysis

Figure 4 presents the potential sources of the biomass burning aerosol observed in Warsaw on 9[th]-11[th] August 2015. In colours the areas appointed on the basis of the trajectory analysis for 02 UTC on August 10[th] (panel (a)) and 02 UTC on August 11[th], 2015 (panel (b)) are shown. Each colour indicates the area from which the air at the altitudes of 1.2 – 5.5 km (panel (a)) and 1.2 – 2.7 km (panel (b)) have originated for the particular time period. The trajectory altitude is rising north-westward (analogously as in Fig. 3). The data of the active fires are obtained from the NASA Fire Information for Resource Management System (FIRMS) based on the satellite data from the MODIS detector on the Terra and Aqua satellites. The collection 6 MODIS Active Fire Data (Giglio et al., 2016; MCD14DL accessed on 28 August 2018) are presented with black dots in Fig. 4. The presented fire points refer to the fires that occurred from 4[th] to 10[th] of August 2015 and are in the time collocation with each time period depicted in colours.

The analysis shows that biomass burning aerosols advected over Warsaw on August 9[th]-11[th], 2015 have originated mainly from Western Europe. Both panels in Fig. 4 point that biomass burning aerosols observed in Warsaw were older than one day (no black points in the yellow area), but it cannot be excluded that some small local fires on the territory of Poland, especially of crops, were not captured by the satellite detector. In the night of 9/10 of August (Fig. 4 panel (a)) 2 days old smoke might have come from the territory of Germany, France, Czech Republic and also Poland. However, the two active fire points in Poland were identified as the copper smelters, thus it could be possible source of the pollution at altitude of about 1.4 km on the first night. Above 3 km, the biomass burning aerosol could origin from wildfires in Portugal, aged of roughly 3 days. Below, at the altitude of about 2.7 km, the air was moving much more slowly and eventual biomass burning aerosol originating from Iberian Peninsula could be aged up to 6 days. At the altitudes of about 2.1-2.4 km biomass burning aerosols of age 3-6 days could origin from southern Germany, Austria and France. In the lowermost layers of 1.2-1.8 km aged biomass burning aerosol from Hungry and Romania was possible. Considering the altitudes of the trajectories, it can be interpreted that at lower altitudes (up to about 2.1 km) the moderately fresh ~2 days aerosol was present, while above, the age could be larger, specially at 2.7 km, where the trajectory has the lowest position at about 5 days before reaching Warsaw. On the night of 10/11 of August smoke aged up to 2 days old, might arrive almost only from the territory of Poland, where the only active fire was detected by MODIS (one black point in the pink area). Biomass burning aerosol aged of about 3 days could origin from Germany, Czech Republic and northern France. Aerosol aged up to 6 days could origin, as previously, from Western Europe or Romania. The lowest altitude of the trajectories along the way suggest 3-4 days aged aerosol.

### 5. Optical properties of the BBA and pollution mixtures

In Fig. 5 (a), (b) a representative for the layers obtained on each night sets of the optical properties profiles with the mean values in the selected sub-layers are shown. The sub-layers selection was based mainly on RH and α profiles. All of the 23 derived sets of profiles are stored in the EARLINET (Earlinet Publishing Group, 2018) and the PolandAOD data bases and





shown in the Appendix A. The data considered in further study of the intensive properties come from the layers below the yellow line, as above the sets of optical properties were not complete.

In the analyzed case, the aerosol load was present in the atmosphere from the ground up to about 5 km at the beginning of the event on August 9[th], gradually descending to about 2.5 km on August 10[th]. However, the optical properties derived within

the multiple sub-layers show heterogeneity of the aerosol and likely occurrence of aerosols mixtures. Up to an altitude of 4 km in Fig. 5 (a) continuous aerosol load with multiple sub-layers, recognized as mainly due to the biomass burning particles of different age in some cases likely mixed with pollution, is present. The layers observed on the second night show less diversity (Fig. 5 (b)). Above the height of 4 km in Fig. 5 (a) very weak layer with relatively higher depolarization ratio ($\delta_{par532}$ ~8%) allows to assume some contribution of mineral dust likely polluted with smoke particles (the West coast of

Sahara pointed by the trajectories at 4 km, Fig. 3 (a), (c)). The layer with relatively higher depolarization is visible in all profiles on the night of 9/10 of August 2015, descending from ~ 5 km to ~ 3.8 km during the course of night. Similar layer was observed also on the second night at the altitude of ~ 2.2 km (Fig. 5 (b)), just after the short sequence of stationary front fluctuation (21:17-21:25 UTC on August 10[th]), as from 12 UTC on August 10[th] to 12 UTC on August 11[th] the front line was in the direct vicinity of Warsaw site (www.pogodynka.pl). It is worth noting that on data presented in Fig. 5 backscatter

coefficient is anticorrelated with the depolarization ratio, i.e. the pronounced depolarization peaks have low backscattering. Similar behaviour was found by Chazette et al., 2017, during the heat wave event over Paris in July 2015. The possible explanation can be due to slight contamination with mineral dust particles, advected from over Sahara with tropical air during the heat wave.

Above the mentioned layers with higher depolarization the region of very low relative humidity of about 5% is visible at

about 5 km on the first night (Fig. 5 (a)) and at about 2.6 km on the second night (Fig. 5 (b)) . Such a low value of RH can be characteristic for the arctic air (Treffeisen et al., 2007; Stachlewska and Ritter, 2010), which is confirmed by the trajectory analysis. The arctic air layer was descending during both analyzed nights from ~5.5 km to ~2.5 km, what reflects the air subsidence in the high pressure system and inflow of cold air on the north-western side of the front line. However, further analysis of the arctic and the depolarizing layers are not the subject of this paper.

The aerosol sub-layers were grouped into the layers of likely similar aerosol species. The choice was performed based on the RH and $\delta_{par532}$ data primarily (Stachlewska and Ritter, 2010). The highlighted layers of specific kind of aerosols are marked in colours in Fig. 6. In several cases, from the initially selected layers the sub-types were discriminated in reference to the analysis of the intensive optical properties relations. The aerosol depicted in dark yellow rectangles, L6(a) in Fig. 6, differs from the light yellow (L5(a)) as exhibits the properties typical for the aged biomass burning aerosol (CRLR >1). The aerosol

in dark pink colour (L4(a)) is unclear, have some features of light pink or cyan layers. The aerosol in dark green (L2(b)) was discriminated from the light green (L1(b)) due to higher particle depolarization ratio.

The mutual relations between the intensive optical properties for all selected sub-layers as well as for the mean optical properties in each aerosol/mixture type selected as the individual layer are presented in the scatter plots in Appendix B. For transparency, the detailed results are shown for each night separately, while the plots with the mean values present data of



both nights together. Chosen for the analysis, relations plots of mean lidar ratios versus depolarization and extinction related Ångström exponent versus colour ratio of lidar ratios as well as the dependences in relative humidity, backscatter and depolarization are depicted in Fig. 7. Table 1 contains the mean values for the aerosol/mixture layers with the standard deviation.

The lidar ratio values for the layers vary in the range of 53-114 sr (355 nm) and 31-85 sr (532 nm). The lowest values of 55±6 sr (355 nm) and 43±4 sr (532 nm) on the night of 9/10 August and 53±5 sr (355 nm) and 31±4 sr (532 nm) on 10/11 August were observed in the two lowermost aerosol layers (L1(a), L1(b)), with the value of $\delta_{par532}$ of 3.6-3.8 %. However, in the second night the lowermost layer at about 1.1 km (L1(b)) was splitted into two separate ones. In the additional layer (L2(b)) higher $\delta_{par532}$ of 4.8±0.3 % and higher LR (especially at 355 nm) of 73±12 sr (355 nm) and 45±11 sr (532 nm) were

noticed. In this layer and in the L1(b), on the same night, the extinction related Ångström exponent ($\text{ÅE}_{355/532}$) had the highest values in the whole analyzed episode of 2.03±0.21 and 1.82±0.37, respectively.

Relative humidity was of 55-58 % in these layers, whereas in the first night (layer L1(a)) the aerosol was more humid with the RH value of 76±2 % and the particles relatively larger ($\text{ÅE}_{355/532}$ of 1.48±0.10). Based on the trajectory analysis it is likely that in the first night industrial pollution from cooper smelters could be observed in the layer at the altitude of about

1.4 km (L1(a)), just above the boundary layer top. However, the daily evolution of the boundary layer on August 9[th] was atypical. The range corrected signal in Fig. 1 indicates that during the day of August 9[th], the boundary layer was rising up to roughly 1.5 km in the afternoon, afterwards another air-mass (with lower aerosol load) forced this layer to move upwards to an altitude of about 1.4 km to 1.6 km. At the same time, unusual decrease of the temperature (~4°C) at the site occurred already at 13 UTC (available via www.polandaod.pl), indicating the weather front passage (typically, during fair weather,

temperature decrease occurs only after 16 UTC, e.g. August 10[th], 11[th]). Therefore, it is possible, that the lowermost layer on the first night (L1(a)) contains the pollution from the smelters and the pollution from the boundary layer.

The relatively low LR and high $\text{ÅE}_{355/532}$ in the layers L1(b) and L2(b) on the night of 10/11 August also indicates pollution domination. The polluted aerosol was located at the top part of the boundary layer (according to the RS estimations). The relatively high value of $\delta_{par532}$ of 4.8 ± 0.3, decreasing with time, along with slight increase of $\text{ÅE}_{355/532}$ may, reflect some

contamination of pollen (Sicard et al., 2016) waning after the sunset. This day was recognized as the intense event of pollination in the boundary layer. To easy compare the depolarization spectral dependence between the layers, the ratio of $\delta_{par355}/\delta_{par532}$ was calculated. The results did not vary significantly (0.36-0.49). The lowest value was found in the layer L2(b), which means that in this aerosol mixture some particles (which depolarize light) has relatively larger size, what confirms the hypothesis of pollen contamination.

In the layers at the altitudes range of 1.5-3.5 km (L2-6(a)), on the first night and at 1.3-2 km (L3-5(b)) on the second night, the LR mean values were higher than in the lowermost ones, which indicates domination by the biomass burning aerosol. Layer L2(a) in the night of 9/10 August have similar properties as L1(a) (with reference to $\delta_{par}$, Ångström exponents, CRLR (Table 1)). Only the lidar ratio of 76±7 sr (355 nm) and 62±4 sr (532 nm) are of about 20 sr higher in L2(a) at both wavelengths. Thus it is possible that the upper layer is stronger contaminated with soot. Microphysical retrieval revealed the





values of imaginary part of refractive index of about 0.12i in the lower layer (L2(a)) and about 0.20-0.30i in the upper one (L2(a)) (not shown here for brevity) (Böckmann et al., in preparation). The relative humidity of 68±4 % in L2(a) is a little lower than in the layer L1(a).

In the night of 9/10 August in the layer L3(a) (observed after advection of aerosols in layer L2(a)), the highest $\delta_{par532}$ of 5.0±0.3 % and $\delta_{par355}$ of 2.46±0.3 % was observed. As the LR values were relatively high of 81±6 sr and 60±6 sr at 355 and 532 nm, respectively, the layer can be treated as fresh biomass burning aerosol, although the $\text{ÅE}_{355/532}$ value of 1.34±0.33 is rather low for fresh BBA. This aerosol layer is weakly pronounced in the backscatter coefficient profile (the signal drop between two other layers). Considering general negative dependence of backscatter and depolarization during this event (Fig. 7 (f)) it is also probable that the increased depolarization is associated with the higher participation of background aerosol in the tropical air containing some mineral dust particles. Nevertheless, the ratio of $\delta_{par355}/\delta_{par532}$ was the highest (~0.49) of the analyzed layers what can suggest bigger amount of the relatively small depolarizing particles in comparison with the other layers.

In the layer L5(a), suspended in the atmosphere at the altitude of about 2.5-3.5 km on 9/10 of August, high lidar ratios values of 71±10 sr and 57±8 sr at 355 and 532, respectively, were observed. In this layer as well as in the layer L6(a) the highest, during this event, values of the relative humidity of 82-85 % were recorded. The particle linear depolarization ratio in both layers was low, with the values of $\delta_{par355}$ ~ 1 % and $\delta_{par532}$ ~ 2.4 %. Such low depolarization ratios are characteristic for aged biomass burning aerosol. However, the threshold of CRLR>1, indicative for aged BBA, was obtained only in five sub-layers. This sub-layers were collected in the layer L6(a) ($\text{LR}_{355}$ of 60±12 sr and $\text{LR}_{532}$ of 67±7 sr). The value of $\text{ÅE}_{355/532}$ was of 1.60±0.22 in the layer L5(a) and 0.60±0.32 in the layer L6(a) and they represent the highest and the lowest values observed on that day. In general, the value of extinction related Ångström exponent is expected to decrease with time and range after of the BBA emission (Müller et al., 2007; Nicolae et al., 2013). The $\text{ÅE}_{355/532}$ value of about 0.6 was reported by Müller et al., 2007 for 5-6 days aged biomass burning aerosol. The value of $\text{ÅB}_{355/532}$ in the layer L6(a) is of 0.89±0.34 and it is higher than the $\text{ÅE}_{355/532}$ which is 0.60±0.32. Similar behaviour of the extinction (ÅE) and backscattering (ÅB) related Ångström exponents for aged BBA was reported by Wandinger et al., 2002 and Janicka et al., 2017. In the mentioned layer the $\text{ÅE}_{355/532}$ is also lower than the $\text{ÅB}_{532/1064}$ value of 0.68±0.12. It cannot be excluded that such a low value of $\text{ÅE}_{355/532}$ of 0.60±0.32 is related to early cloud formation, although the altitudes of passing cumulus clouds at the night of 9/10 August in the Layer 6(a) are in agreement only in this one case (Fig. 6).

The layer L3(b) at the night of 10/11 August is characterized by the highest lidar ratios of 114±33 sr (355 nm) and 78±25 sr (532 nm) during the analyzed event. At the same time, the RH was the lowest (48±4 %) in L3(b). The mean depolarization ratios of 1.6±0.3 % (355 nm) and 3.8±0.5 % (532 nm) are the same in given uncertainty range as in the layer L1(b). The $\text{ÅE}_{355/532}$ value of 1.50±0.19 was observed. Gradual changes in the optical properties can be observed with height, going upward from the mentioned layer L3(b) to the layer L5(b), with the lidar ratios values decreasing with height (80±7 sr (355 nm) in L4(b) to 69±9 sr (355 nm) in L5(b) and 66±9 sr (532 nm) in L4(b) to 61±7 sr (532 nm) in L5(b)). The RH was increasing with altitude up to the value of 76±4 %. The particle linear depolarization ratios as well as the extinction related



Ångström exponent were decreasing with height from the $\delta_{par355}$ value of 1.6±0.3 % to 1.0±0.1 %, from the $\delta_{par532}$ value of 3.8±0.5 % to 2.3±0.3 % and from $\text{ÅE}_{355/532}$ value of 1.50±0.19 to 1.30±0.15. The backscattering related Ångström exponent $\text{ÅB}_{355/532}$ and colour ratio of lidar ratios were increasing with height. Only for the $\text{ÅB}_{532/1064}$ the value in the layer L4(b) was higher than the one in the L3(b). It shows, that in the case of the layers L3-5(b) the biomass burning aerosol rather dominate

in the layers, whereby the BBA was more aged with height, which is also confirmed by the descending value of CRLR.

Concerning the plots containing mean values (Fig. 7 and Appendix B, right panel) one can see that in general, for many cases linear dependence of the aerosol properties is visible. By going into the details, analyzing many values for each night separately (plots with all obtained properties for each of 113 sub-layers, e.g. Fig. B2, B5 in Appendix B left and middle panels), such relations can be seen only for one night alternatively, or the dependence is different than the linear. For several

cases (e.g. Fig. B4) the points that scattered off the others on the one night fit better to the trend obtained on the second night. Often, no single but more trend lines are discernable, although for some cases even this left some outsiders. This behaviour is interpreted as due to the observation of mixtures and not pure aerosol types.

Grouping of the aerosol can be seen with respect to LR and $\delta_{par}$ for both wavelengths (Fig. 7 (a), (b)). In circles, the aerosol mixture with the domination of biomass burning aerosol (black circle) and pollution (yellow circle) are depicted. Based on

these two quantities the discrimination of aged BBA (dark yellow) is not clearly seen. Several layers cannot be distinctly classified due to either the contamination with another species or another stage of the aerosol lifecycle in the atmosphere. Analysing detailed relations (Appendix B) it is visible that the Layer 3 (b) (in red) evolved with time from the higher values of lidar ratio and depolarization to the lower ones, thus probably from a more fresh to aged smoke, then eventually it could be splitted into two separate layers.

A strong negative relation is seen in the $\text{ÅE}_{355/532}$ and CRLR scatter plot (Fig. 7 (c)). Similar result of negative linear dependence of this two parameters and positive dependence of the CRLR and particle effective radius was reported by Samaras et al., 2015, for the biomass burning aerosol, demonstrating increasing particles size with age of the aerosol. The effective radius changed also with an increase of the relative humidity, indicating hygroscopicity of the particles. Nevertheless, concerning the origin of the $\text{ÅE}_{355/532}$ and CRLR, one see that they are not totally independent and both

contains the same extinction coefficients ratio. ÅE is a function of ÅB and CRLR for the same pair of wavelengths. The indicator of aged biomass burning aerosol CRLR>1 yields the same information as the expression $\text{ÅB}_{355/532} > \text{ÅE}_{355/532}$, and conversely. Veselovskii et al., 2015 reported, based on the model simulations, that $\text{ÅE}_{355/532}$ depends mainly on the particle size, but the $\text{ÅB}_{355/532}$ depends on both the particle size and the real part of the refractive index and that $\text{ÅB}_{355/532}$ is more sensitive for the refractive index variation than $\text{ÅB}_{532/1064}$. Note that in their study imaginary part value was constant and

small (0.005i). The CRLR can be written as a distinct exponential function of the difference between ÅB and ÅE (ÅB-ÅE) for the same pair of wavelengths, and indeed derived mean values for the layers meet this relationship.

Presented in Fig. 7 (d), (e), (f) scatter plots of the relative humidity, depolarization ratio and backscatter coefficient shows positive correlation in the case of the RH and $\beta_{532}$, and the negative for the RH and $\delta_{532}$. The relation of growing sphericity of the particles due to the water uptake, and hence decreasing of light depolarization was reported for the marine aerosol



(Haarig et al., 2017). The positive correlation of the RH and $\beta_{532}$ resulting from the particle size increasing because of the water uptake is seen in current study as in Haarig but for smoke mixture. Therefore, the negative dependence of the $\beta_{532}$ and $\delta_{par532}$ is unlikely related to mineral dust particles contamination possible in the advected tropical air but should be understood rather as a consequence of the relations of RH - $\beta_{532}$ and RH - $\delta_{532}$ mentioned before.

Table 2 presents the aerosol optical depth (AOD) calculated from the lidar measurements for free troposphere aerosols, above the boundary layer top assessed to 1.2 km. The calculations were performed to verify the consistency of the obtained results. The $AOD_{FT}$ was calculated from 1.2 km to the upper limit of the aerosol layers occurrence on each night, i.e. on the first night of 9/10 August the $AOD_{FT}$ was integrated up to 4 km, whereas on the second night up to 2.5 km. The mean daily columnar $AOD_{CL}$ and the first and the last $AOD_{CL}$ record on each day at 414 nm and 496 nm form the solar radiometer

MFR-7 operated close the Warsaw lidar site within the PolandAOD-NET are listed in Table 1 for comparison. The uncertainty of $AOD_{CL}$ from MFR-7 is at the level of ±0.025.

The $AOD_{FT}$ values observed during the analyzed event were high up to 0.85 (355 nm) and 0.46 (532 nm). Comparison of the closest in time radiometer and lidar data show very good agreement, indicating that the free troposphere aerosol load dominated over the boundary layer aerosol load. The AOD values are similar in spite of the fact that the boundary layer

AOD was not taken into the mean. The climatological value of the boundary layer $AOD_{BL}$ reported by Wang et al., 2018, indicates that the lidar $AOD_{FT}$ value was 2-3 times higher. This situation is possible only in the case of very strong intrusion of aerosols in free troposphere. The majority of aerosol load observed on that days were suspended above the boundary layer. The lidar $AOD_{FT}$ values calculated for each consecutive half an hour present a consistent course during the night, with respect to both wavelengths. Mean daily AOD results from radiometer in Warsaw and lidar $AOD_{FT}$ reflect decreasing

aerosol load during the analyzed period. Note that this results of PolandAOD-NET MFR-7 are partly consistent with the data from the nearby AERONET (https://aeronet.gsfc.nasa.gov/) site in Belsk (~50 km to the south of Warsaw). Thus, the data obtained at both sites in this period are not comparable, what reflects complicated meteorogical conditions of the quasi-stationary weather front.

**Conclusion and Outlook**

In this study the lidar observations of an aerosol mixtures, dominated by the biomass burning smoke were analysed. The quasi-continuous 24/7 measurements were performed with the multiwavelength Raman-polarization and water vapour lidar in Warsaw during the heat wave event on August 9[th]-11[th], 2015. The air-mass transport and the aerosol source analysis show that the 2-3 days transport from over Germany, France and Czech Republic brought moderately fresh biomass burning

aerosol. Furthermore, at the altitude of about 2.7-3 km on the first night, the inflow of aged (up to 5 days) aerosol from wildfires on the Iberian Peninsula was possible, confirmed further by the optical properties is some sub-layers. The comparison of the HYSPLIT trajectories simulated with the GDAS and the reanalysis meteorological data showed surprisingly good result obtained with model data under such challenging dynamical conditions related to the weather front occurrence.



The detailed analysis was based on specifying the multiple individual aerosol/mixture sub-layers in the sets of $3\beta+2\alpha+2\delta+wv$ optical profiles averaged over each consecutive 30 minutes during the two nights, so that fine temporal and spatial scales were derived. Total number of 113 sub-layers with set of the mean values of the extensive and intensive optical parameters in each o them were used for general aerosol/mixture layer discrimination and description in the statistical

approach.

In the analysis the lowermost layers during both nights were identified to have dominant contribution of anthropogenic pollution and were characterized by low lidar ratios ($LR_{355} \sim 55$ sr, $LR_{532}$ 31-43 sr), high $AE_{355/532}$ up to 2 and depolarization ratio of about 1.6 % (355 nm) and 3.6 % (532 nm). At the beginning of the second night slight contain with residual daytime released pollen particles is possible. In upper layers, domination of biomass burning aerosol of different age were observed.

In one layer the aged BBA (>5 days) is indicated by the CRLR>1 and $AB_{355/532}>AE_{355/532}$, recognized to be equivalent information. Nevertheless, this aerosol do not show an uniform layer but occurs in parts within other aerosol type layer, what points inhomogeneous mixing or cloud effects on the aerosol (low value of AE355/532 of 0.6 can be related not only to the aged BBA but also to early cloud formation). In other layers, fresh (<1 day) to moderately fresh BBA (2-3 days) was observed as the dominating aerosol species. The $LR_{355}$ of 69-114 sr and $LR_{532}$ of 57-85 sr from mean layer characteristic

were found, whereas the $\delta_{par355}$ was of 1-2.5 % and $\delta_{par532}$ of 2.3-5 %. The layers had different relative humidity, which was negatively correlated with the depolarization ratio and positively with the backscatter coefficient.

As the atmospheric aerosol load in this study was expected to be of some kinds of mixtures, the use of algorithms for specific aerosol component separation with this data set would be beneficial (not subject to this paper). Moreover, the results obtained constitute an excellent input for the mathematical inversion algorithms for the aerosol microphysical parameters

retrieval as they provide the required good quality set of $3\beta+2\alpha$ optical data, with an indication of the particles polarization degree and relative humidity. The assumption of spherical particles can be done for presented case, as the observed depolarization ratios are relatively low (<10%). Due to the ill-posedness the microphysical inversion is a challenging task and is burdened with high uncertainties. Doing the calculations for many individual sub-layers, which are a part of the group with similar optical properties allow to expect similar results in the layer classified as one aerosol/mixture type. This

statistical method can meaningly lower the retrieval uncertainties. The successful microphysical properties retrieval with such a rich data set can provide temporal and spatial distribution of the refractive index and the particle size distribution in the atmosphere. Such study was not performed so far and we are looking forward to compute it in the future.

Finally, the provided set of data would be excellent for testing the aerosol typing algorithms such as the newly developed NATALI code.

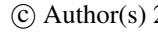



**Acknowledgements:**

Algorithms used in this paper developed in the frame of Development of a European High Spectral Resolution Lidar Airborne Facility (MULTIPLY) project founded by ESA-ESTEC Contract No. 4000112373/14/NL/CT for purpose of validation of Multiply lidar data with ground based lidar.

The PollyXT-UW lidar was developed in a collaboration of scientists from the University of Warsaw and the Institute of Tropospheric Research (TROPOS) in the PollyXT development group of Dietrich Althausen. The development of the lidar was financed by the Polish Foundation of Science and Technology (FNTP-No. 519/FNITP/115/2010).

The RS-Lab located in the city center of Warsaw, central Poland, is a part of the Polish Aerosol Research Network (PolandAOD-NET, www.polandaod.pl). The station represents the only Polish Raman lidar site within the European Aerosol

Research Lidar Network (EARLINET-ACTRIS, www.earlinet.org) and the worldwide Polly.NET lidar network (www.polly.tropos.de). The RS-Lab in Warsaw performs measurements for AERONET (https://aeronet.gsfc.nasa.gov/).

The EARLINET is currently supported by the ACTRIS-2 project, funded by the European Union Research Infrastructure Action under the H2020 specific program for Integrating and opening existing national and regional research infrastructures of European interest under Grant Agreement No. 654109 and No. 739530. The University of Warsaw participates in

ACTRIS-2 project as an associate partner without funding.

The authors gratefully acknowledge the NOAA Air Resources Laboratory (ARL) for the provision of the HYSPLIT transport and dispersion model and/or READY website (http://www.ready.noaa.gov) used in this publication.

We acknowledge the use of data and imagery from LANCE FIRMS operated by NASA's Earth Science Data and Information System (ESDIS) with funding provided by NASA Headquarters.

The authors acknowledge the Institute of Meteorology and Water Management—The National Research Institute in Poland IMGW-PIB for the use of the weather charts.

**Competing Interests:** The authors declare no conflict of interest. The authors equally contributed to this manuscript.

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



**Table 1.** Mean values of the intensive optical properties in the selected layers (Fig. 6) with the dominating aerosol type. The colours and designation of the layers are analogous as in Fig. 6 (schematic plot of the layers), Fig. 7 and Appendix B. In Appendix B the scatter plots (left and middle panels) presents optical values in all sub-layers used for calculation of the means.

| Dominating aerosol type | Selected layer | $LR_{355}$ [sr] | $LR_{532}$ [sr] | RH [%] | $\delta_{par355}$ [%] | $\delta_{par532}$ [%] | ÅE 355/532 | ÅB 355/532 | ÅB 532/1064 | CRLR |
|---|---|---|---|---|---|---|---|---|---|---|
| | | | | | | 09/10August 2015 | | | | |
| pollution | L1(a) | 55 ± 6 | 43 ± 4 | 76 ± 2 | 1.5 ± 0.1 | 3.6 ± 0.3 | 1.48 ± 0.10 | 0.86 ± 0.08 | 0.40 ± 0.04 | 0.78 ± 0.04 |
| BBA mod. fresh | L2(a) | 76 ± 7 | 62 ± 4 | 68 ± 4 | 1.6 ± 0.2 | 3.5 ± 0.3 | 1.35 ± 0.18 | 0.85 ± 0.14 | 0.46 ± 0.05 | 0.82 ± 0.09 |
| BBA fresh | L3(a) | 81 ± 6 | 60 ± 6 | 53 ± 4 | 2.5 ± 0.4 | 5.0 ± 0.3 | 1.34 ± 0.33 | 0.60 ± 0.13 | 0.52 ± 0.07 | 0.75 ± 0.10 |
| mixed | L4(a) | 96 ± 1 | 85 ± 5 | 62 ± 4 | 2.0 ± 0.6 | 4.1 ± 0.8 | 1.15 ± 0.09 | 0.86 ± 0.04 | 0.35 ± 0.10 | 0.89 ± 0.05 |
| BBA mod. aged | L5(a) | 71 ± 10 | 57 ± 8 | 85 ± 6 | 1.0 ± 0.2 | 2.3 ± 0.3 | 1.60 ± 0.22 | 1.05 ± 0.28 | 0.65 ± 0.14 | 0.81 ± 0.10 |
| BBA aged /early cloud | L6(a) | 60 ± 12 | 67 ± 7 | 82 ± 8 | 1.1 ± 0.4 | 2.5 ± 0.7 | 0.60 ± 0.32 | 0.89 ± 0.34 | 0.68 ± 0.12 | 1.13 ± 0.14 |
| | | | | | | 10/11 August 2015 | | | | |
| pollution | L1(b) | 53 ± 5 | 31 ± 4 | 55 ± 1 | 1.6 ± 0.1 | 3.8 ± 0.2 | 2.03 ± 0.21 | 0.69 ± 0.13 | 0.36 ± 0.04 | 0.58 ± 0.06 |
| pollution + pollen | L2(b) | 73 ± 12 | 45 ± 11 | 58 ± 2 | 1.8 ± 0.1 | 4.8 ± 0.3 | 1.82 ± 0.37 | 0.56 ± 0.10 | 0.26 ± 0.05 | 0.60 ± 0.07 |
| BBA fresh | L3(b) | 114 ± 33 | 78 ± 25 | 48 ± 4 | 1.6 ± 0.3 | 3.8 ± 0.5 | 1.50 ± 0.19 | 0.58 ± 0.24 | 0.44 ± 0.03 | 0.69 ± 0.07 |
| BBA mod. fresh | L4(b) | 80 ± 7 | 66 ± 9 | 64 ± 2 | 1.1 ± 0.1 | 2.8 ± 0.1 | 1.41 ± 0.12 | 0.90 ± 0.18 | 0.39 ± 0.05 | 0.82 ± 0.09 |
| BBA mod. aged | L5(b) | 69 ± 9 | 61 ± 7 | 76 ± 4 | 1.0 ± 0.1 | 2.3 ± 0.3 | 1.30 ± 0.15 | 0.99 ± 0.10 | 0.47 ± 0.06 | 0.88 ± 0.05 |





**Table 2. Aerosol optical depth calculated from the lidar measurements for free troposphere aerosols observed on August 9th-11th, 2015 in Warsaw in comparison with the PolandAOD-NET MFR-7 radiometer data (Warsaw).**

| | Time UTC | $AOD_{CL}$ 414 nm PolandAOD-NET | $AOD_{FT}$ 355 nm free troposphere lidar | $AOD_{CL}$ 496 nm PolandAOD-NET | $AOD_{FT}$ 532 nm free troposphere lidar |
|---|---|---|---|---|---|
| daily mean | 09.08.2015 | 0.689 | - | 0.516 | - |
| 09/10.08.2015 free troposphere $AOD_{FT}$ lidar 1.2- 4 km | 16:57 | 0.0645 | - | 0.475 | - |
| | 19:15 | | 0.85 | - | 0.44 |
| | 20:00 | | 0.71 | - | 0.41 |
| | 20:30 | | 0.8 | - | 0.43 |
| | 21:00 | | 0.79 | - | 0.41 |
| | 21:30 | | 0.66 | - | 0.4 |
| | 23:00 | | 0.69 | - | 0.36 |
| | 23:30 | | 0.63 | - | 0.38 |
| | 00:00 | | 0.64 | - | 0.44 |
| | 01:30 | | 0.79 | - | 0.46 |
| | 04:44 | 0.819 | - | 0.596 | - |
| daily mean | 10.08.2015 | 0.527 | - | 0.402 | - |
| 10/11.08.2015 free troposphere $AOD_{FT}$ lidar 1.2-2.5 km | 16:56 | 0.365 | - | 0.284 | - |
| | 19:00 | | 0.39 | - | 0.23 |
| | 19:30 | | 0.34 | - | 0.22 |
| | 20:00 | | 0.3 | - | 0.21 |
| | 20:30 | | 0.29 | - | 0.19 |
| | 21:17 | | 0.25 | - | 0.15 |
| | 21:30 | | 0.26 | - | 0.18 |
| | 22:00 | | 0.29 | - | 0.19 |
| | 22:30 | | 0.3 | - | 0.17 |
| | 23:00 | | 0.3 | - | 0.18 |
| | 23:30 | | 0.32 | - | 0.19 |
| | 00:00 | | 0.27 | - | 0.17 |
| | 00:30 | | 0.34 | - | 0.2 |
| | 01:00 | | 0.38 | - | 0.23 |
| | 01:30 | | 0.38 | - | 0.26 |
| | 04:46 | 0.415 | - | 0.332 | - |
| daily mean | 11.08.2015 | 0.249 | - | 0.184 | - |
| | Long term 2013-2017 mean lidar $AOD_{BL}$ (Wang et al., 2018) | 0.27±0.17 | | - | 0.15±0.10 |





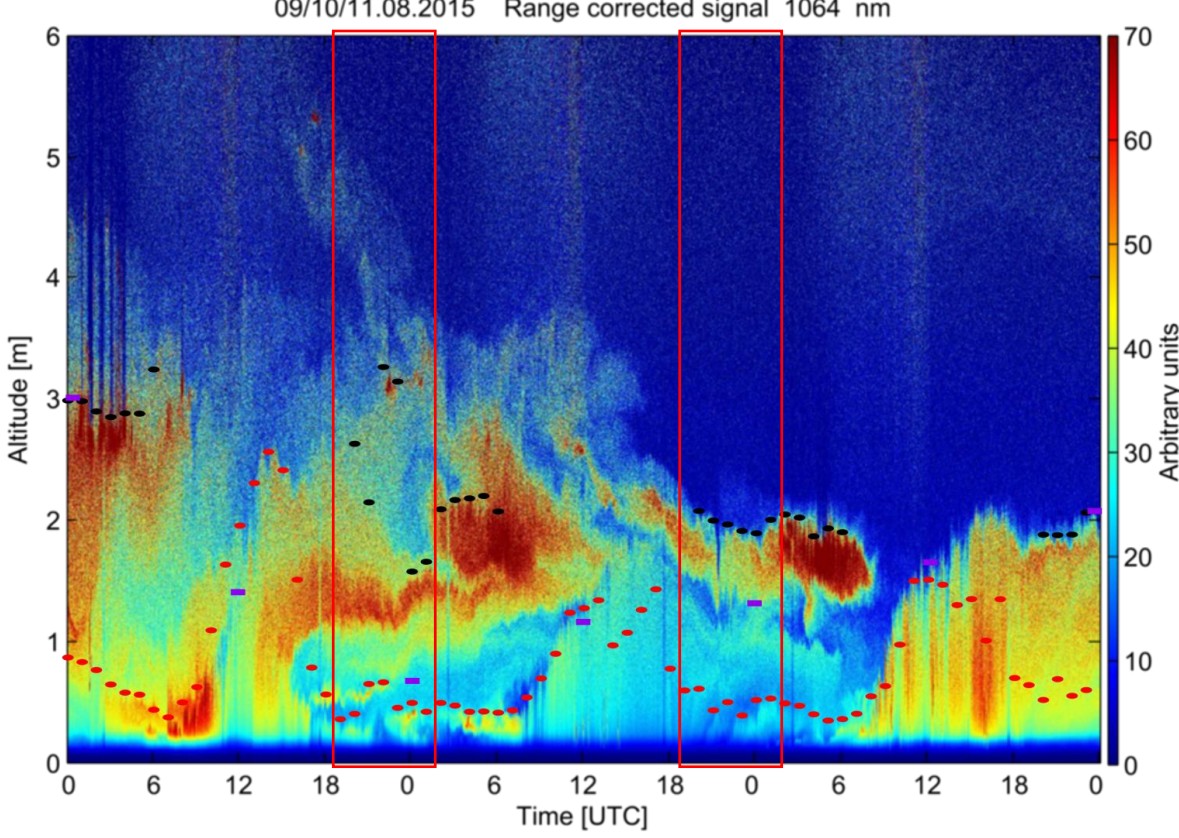

**Figure 1: Range corrected signal at 1064 nm for the period of August 9ᵗʰ-11ᵗʰ, 2015 with the boundary layer height marked (red dots depict the aerosol boundary layer, black dots depicts residual layer calculated with wavelet method, purple bars depicts boundary layer estimated form the radiosounding). The red rectangles depict the time periods selected to the evaluation.**



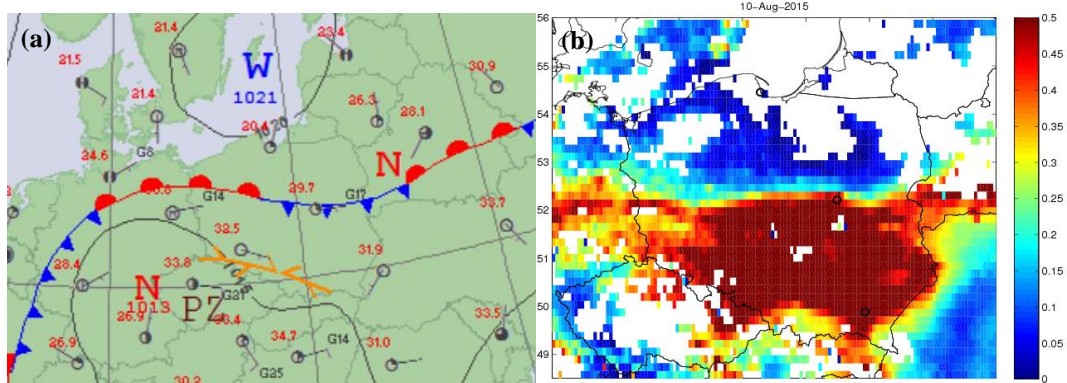

**Figure 2: (a) The synoptic map of 10.08.2015 12 UTC (source: the online Weather Service of the Institute of Meteorology and Water Management, IMGW, www.pogodynka.pl); (b) The spatial distribution of AOD at 550 nm on 10.08.2015 from the satellite MODIS detector (source: www.polandaod.pl).**





**Figure 3: The HYSPLIT 6-days backward trajectories calculated for Warsaw location for the altitudes of 1.2-5.5 km with 0.3 km interval for 19 UTC 09.08.2015 (a), 02 UTC 10.08.2015 (b) (c) and calculated for the altitudes of 1.2-2.7 km with 0.3 km interval for 19 UTC 10.08.2015 (d), 02 UTC 11.08.2015 (e) (f). The left and the middle panels show calculation with reanalysis meteorogical data (a) (b) (d) (e), the right panels with GDAS model data (c) (f).**



**Figure 4: The potential sources of the biomass burning aerosols at 02 UTC on August 10th, 2015 (a) and at 02 UTC on August 11th, 2015 (b). In colours the areas from which the air at the altitudes of 1.2 – 5.5 km (a) and 1.2 – 2.7 km (b) have originated for the particular time period; the trajectory altitude is rising north-westward. With black dots the collection 6 MODIS Active Fire Data from 4th to 10th of August are presented; active fire data are in the time collocation with each time period depicted in colours.**



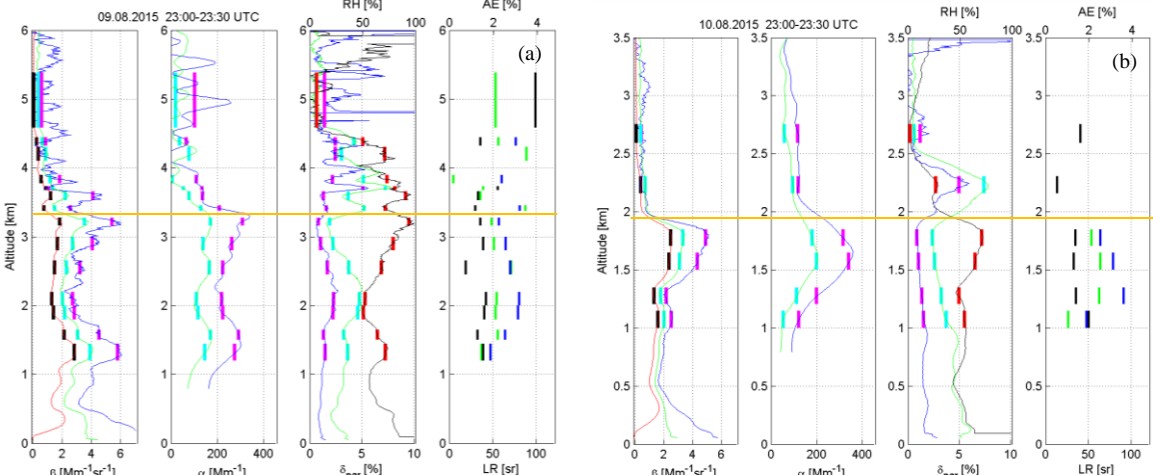

**Figure 5: The optical properties profiles for 23:00-23:30 UTC on August 9th (a) and 10th (b), 2015. In blue (magenta), green (cyan) and red (black) lines (bars) the particle backscatter (β) and extinction (α) coefficients and particle depolarization ratio (δ_par) profiles (mean values in the layers) for 355, 532 and 1064 nm respectively are presented; in black line (red bars) the relative humidity profile (RH) (mean values in the layers) is shown; in blue and green bars the mean lidar ratios (LR) for 355 and 532 nm are shown; the mean extinction related Ångström exponent 355/532 (AE) is depicted in black bars. The yellow line depicts the limit up to which the intensive optical properties were taken into account in further analysis.**





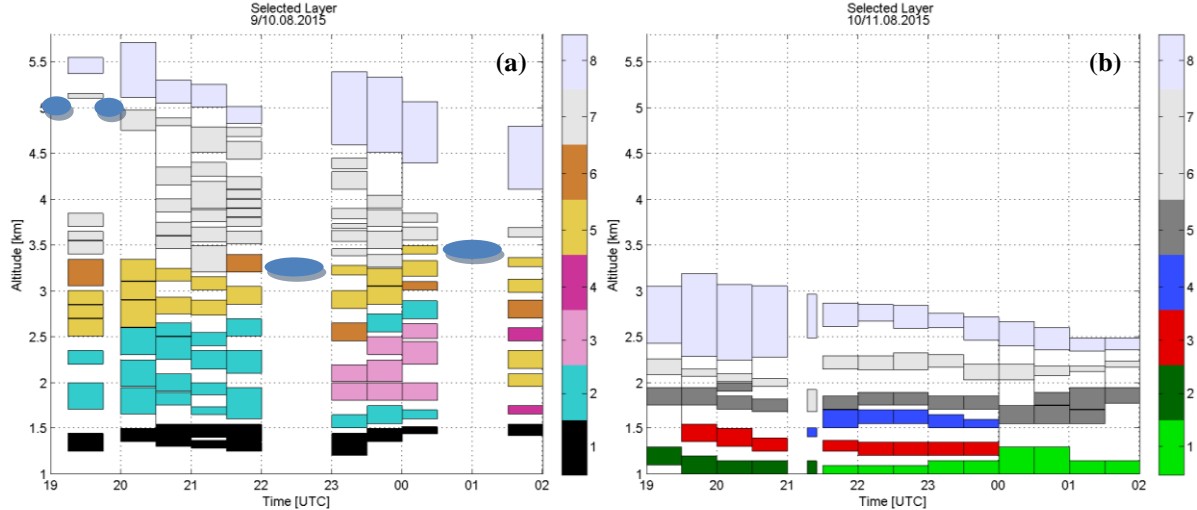

**Figure 6: Schematic plot of the same aerosol type layers selected in the 30-minutes profiles in the nights of 9/10 and 10/11 of August 2015. The same colour depict one aerosol type. The layers numbered 1-6 in the panel (a) and 1-5 in the panel (b) depict the biomass burning and pollution mixed layers considered in the further study; the layer 7 in the panel (a) and 6-7 in the panel (b) (light grey) depict other aerosols with higher uncertainty not analyzed in the further study; the layer number 8 (light purple) depict the arctic air layer (not analyzed in the further study). Blue ovals in the panel (a) depict small clouds occurred.**




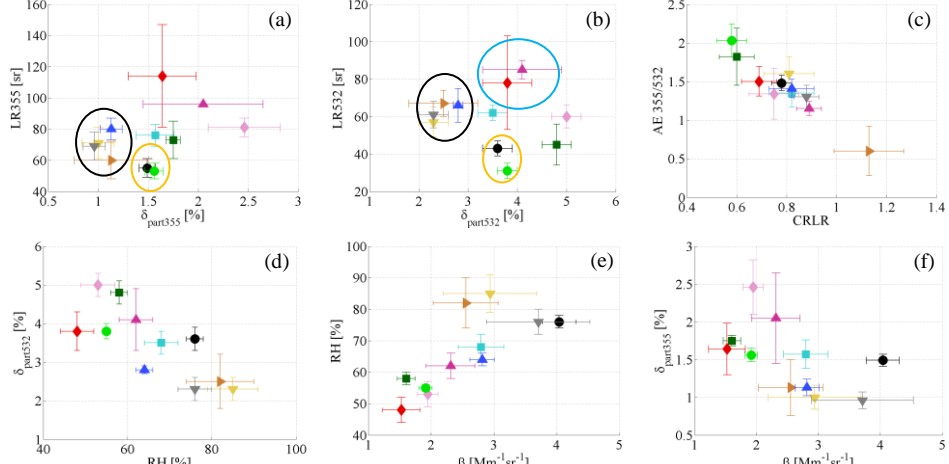

**Figure 7: Scatter plots of mean optical properties in the layers (Table 1). The lidar ratio versus depolarization ratio is depicted in panels: (a) 355 nm and (b) 532 nm; the extinction related Ångström exponent 355/532 versus colour ratio of lidar ratios (532/355) is depicted in panel (c); the particle depolarization ratio at 532 nm, relative humidity and backscatter coefficient at 532 nm are depicted in panels (d) (e) (f). In circles the properties which are clearly grouping in the plot for anthropogenic pollution (yellow) and biomass burning aerosol (black) mixtures dominated are marked.**



# Appendix A

Sets of the Raman optical properties profiles averaged over nine 30-minutes periods from 19:15 UTC to 02:00 UTC on the night of 9/10 August 2015 (Fig. A1) and over thirteen 30-minutes periods and one 8-minute period from 19:00 UTC to 02:00 UTC on the night of 10/11 August 2015 (Fig. A2). In blue (magenta), green (cyan) and red (black) lines (bars) the particle

5  backscatter (β) and extinction (α) coefficients and the particle depolarization ratio ($\delta_{par}$) profiles (mean values in the layers) for 355, 532 and 1064 nm, respectively, are presented; in black line (red bars) the relative humidity profile (RH) (mean values in the layers) is shown; in blue and green bars the mean lidar ratios (LR) for 355 and 532 nm are shown; the mean extinction related Ångström exponent 355/532 (AE) is depicted in black bars. The yellow line depicts the limit up to which the intensive optical properties were taken into account in further analysis.

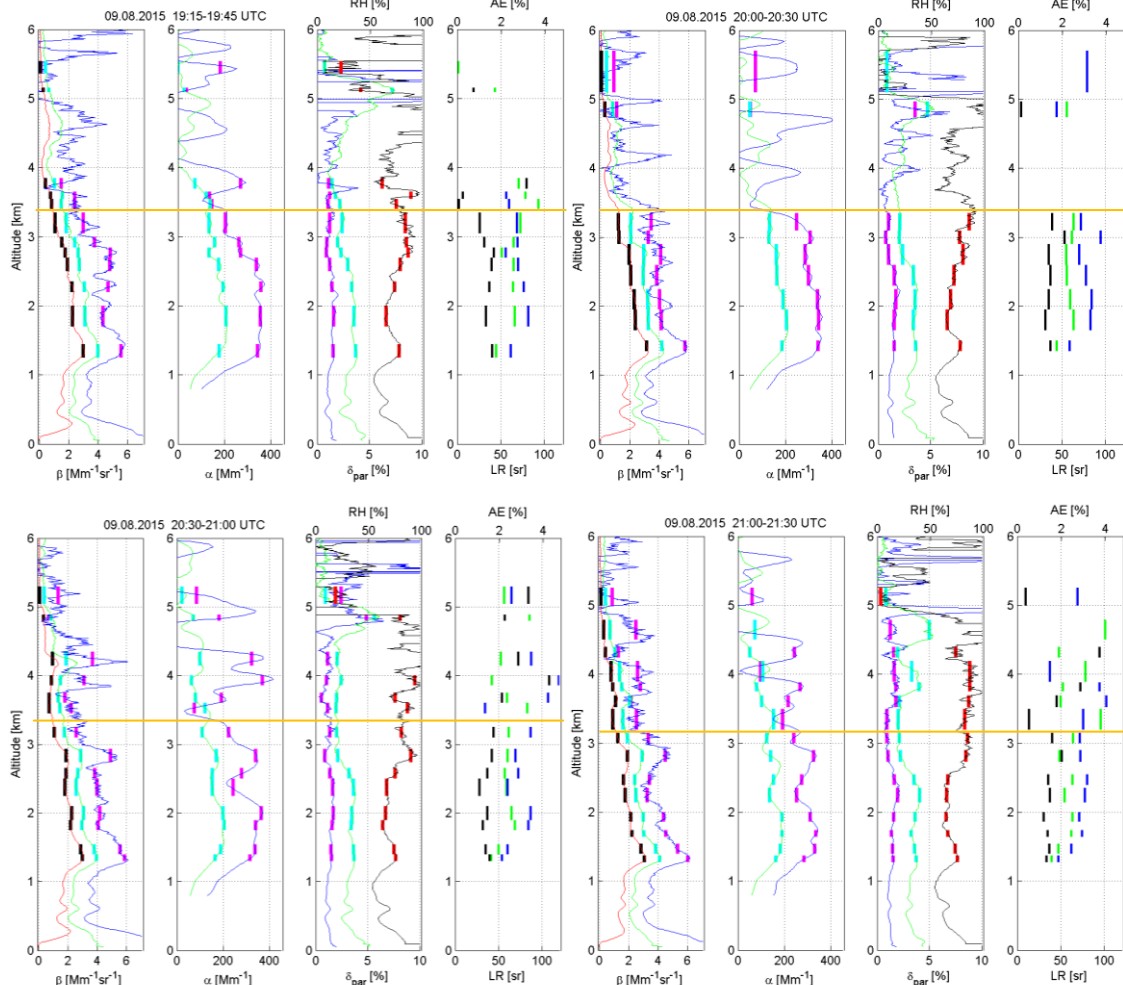





**Figure A1: Sets of the Raman optical properties profiles averaged over nine 30-minutes periods from 19:15 UTC to 02:00 UTC on the night of 9/10 August 2015.**

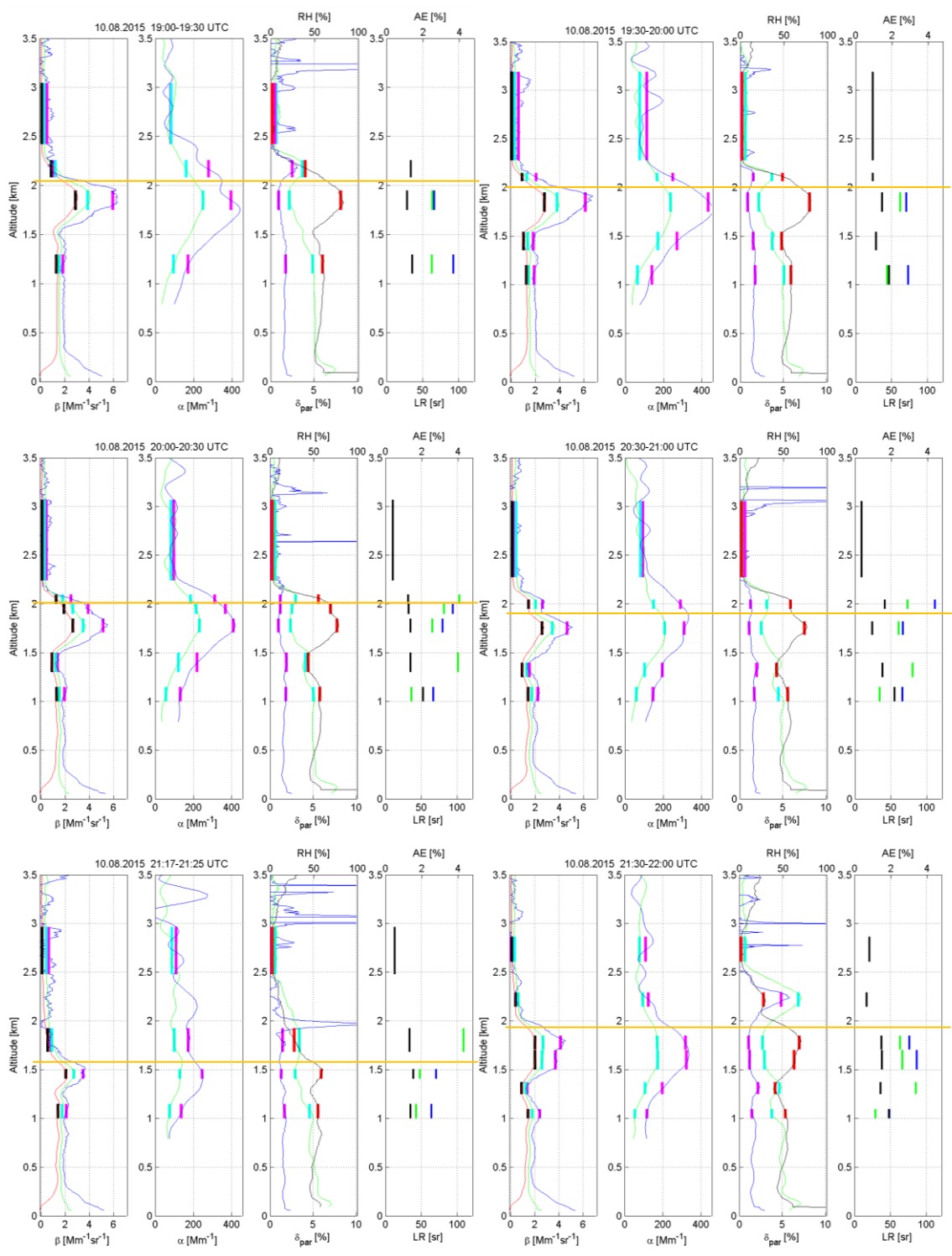

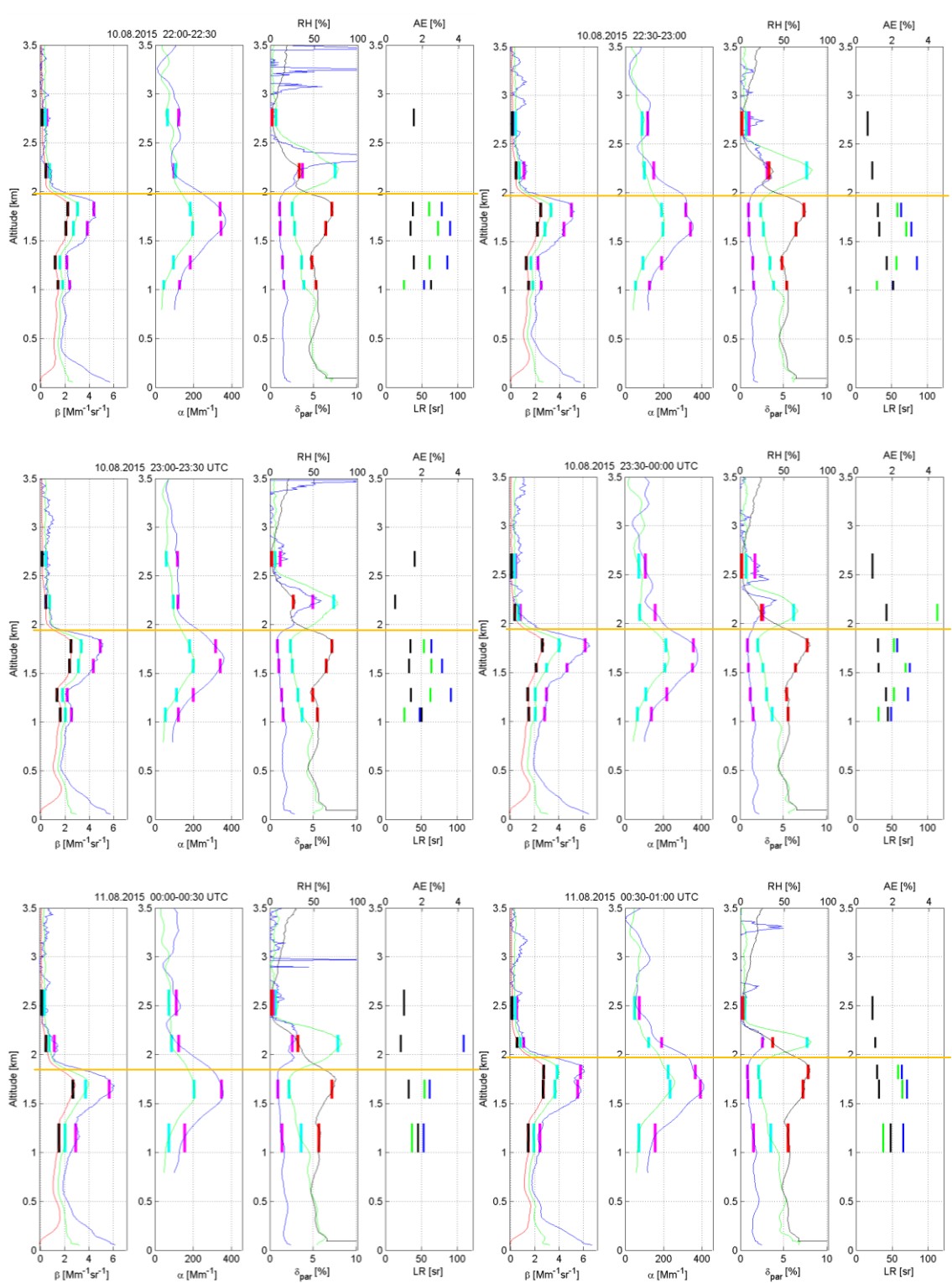



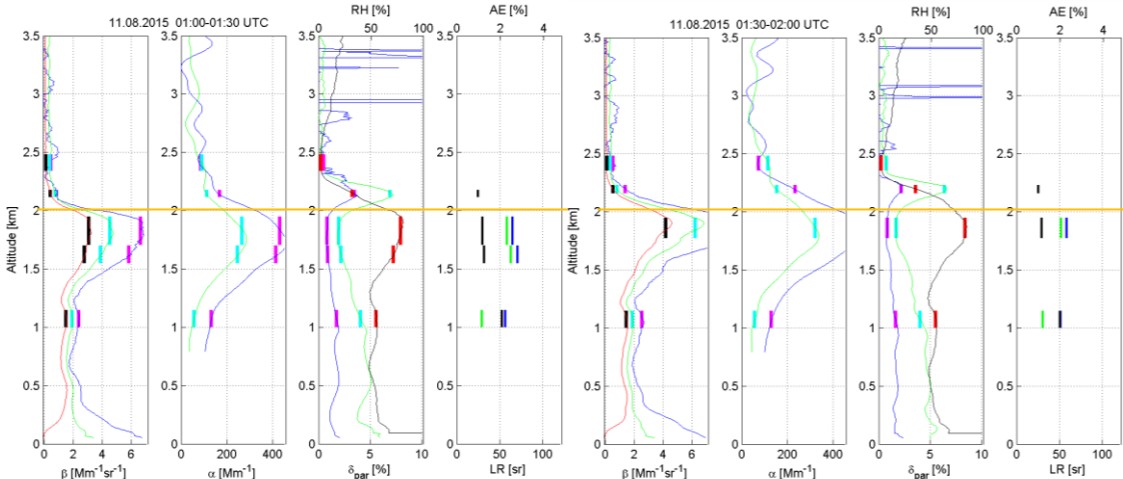

**Figure A2: Sets of the Raman optical properties profiles averaged over thirteen 30-minutes periods and one 8-minute period (21:17-21:25 UTC) from 19:00 UTC to 02:00 UTC on the night of 10/11 August 2015.**



## Appendix B

Relations between the intensive optical properties for 113 selected aerosol sub-layers (69 sub-layers on the night of 9/10 August 2015 and 44 sub-layers on the night of 10/11 August 2015) (left and middle panels) as well as the mean optical properties (Table 1) of each aerosol/mixture type selected as the individual layer (right panels). For transparency, the
5  detailed results in left and middle panels are shown for each night separately, while the plots with the mean values in right panels present data of both nights together. The colours and designation of the layers are analogous as in Table 1, Fig. 6 (schematic plot of the layers) and Fig. 7.

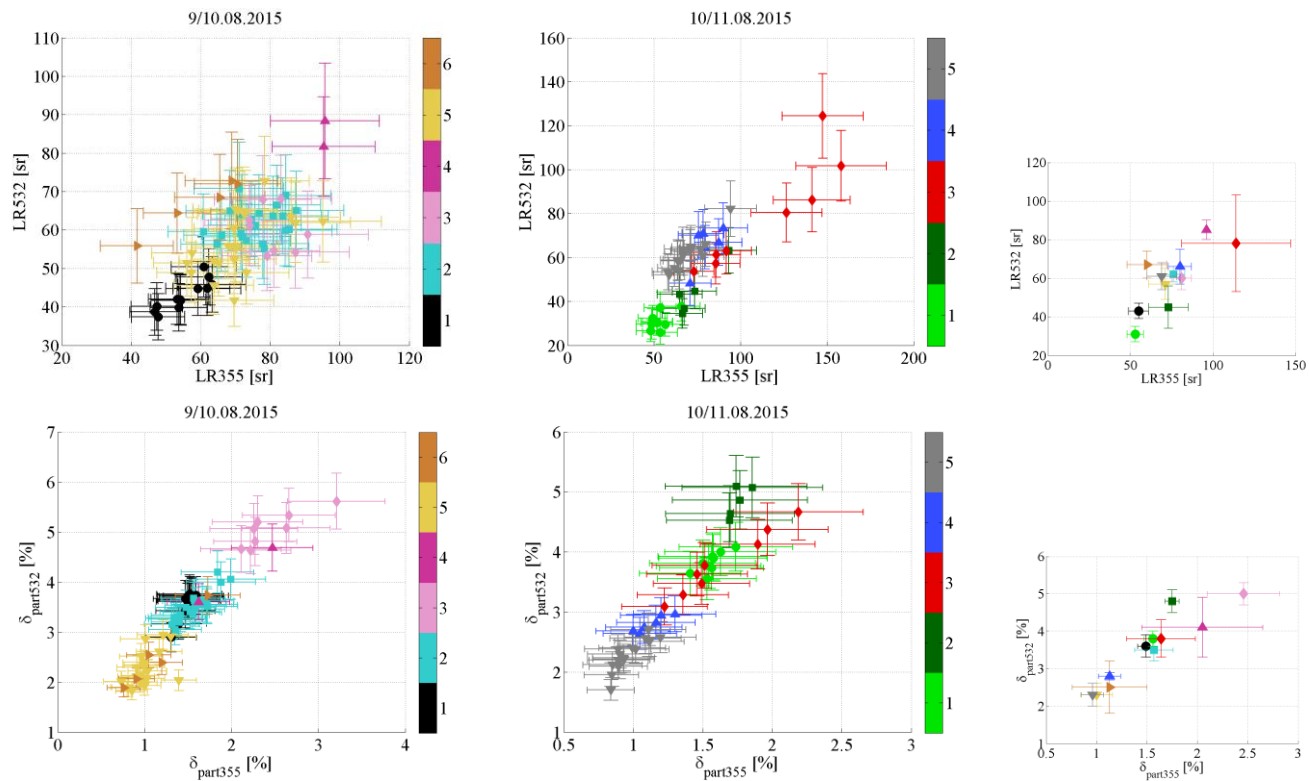

**Figure B1: Scatter plots of the lidar ratio at 355 versus 532 nm (upper panels) and the particle depolarization ratio at 355 versus 532 nm (bottom panels).**





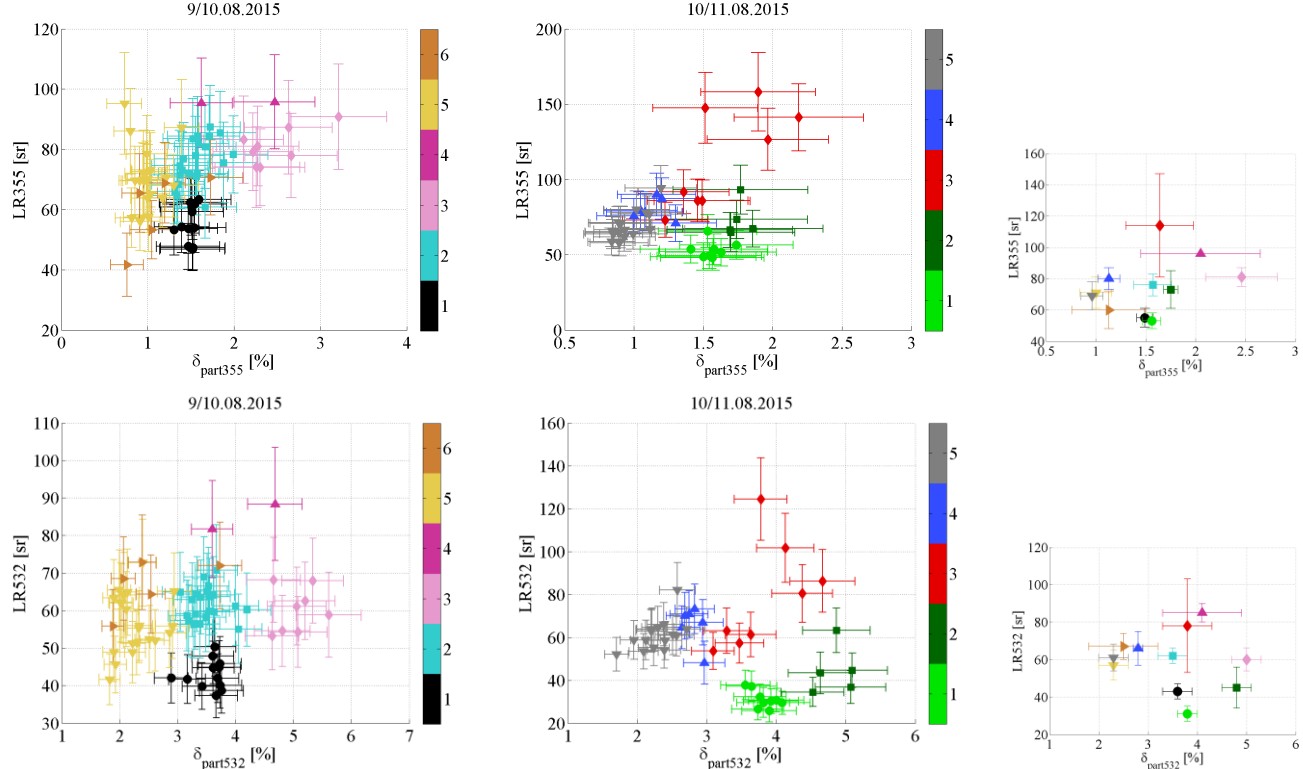

**Figure B2: Scatter plots of the particle depolarization ratio versus the lidar ratio both at 355 nm (upper panels) and 532 nm (bottom panels).**

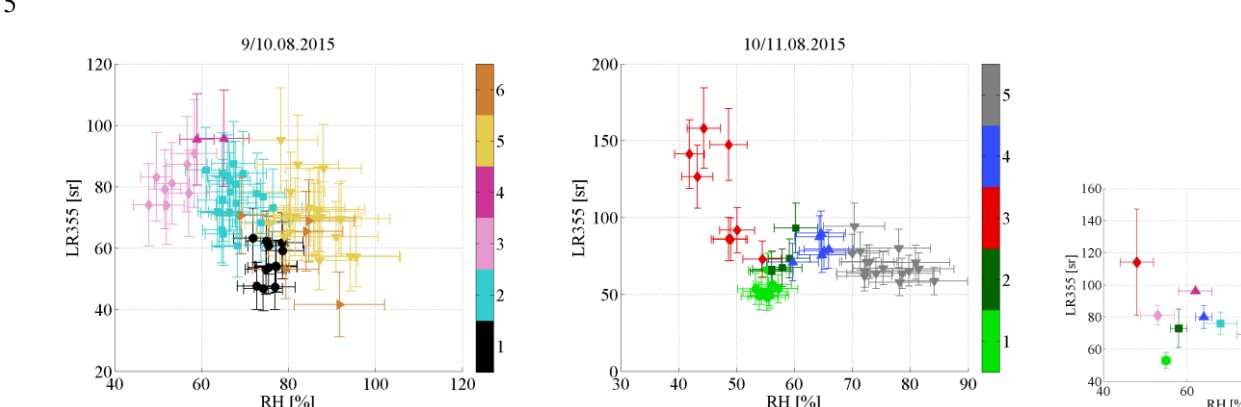





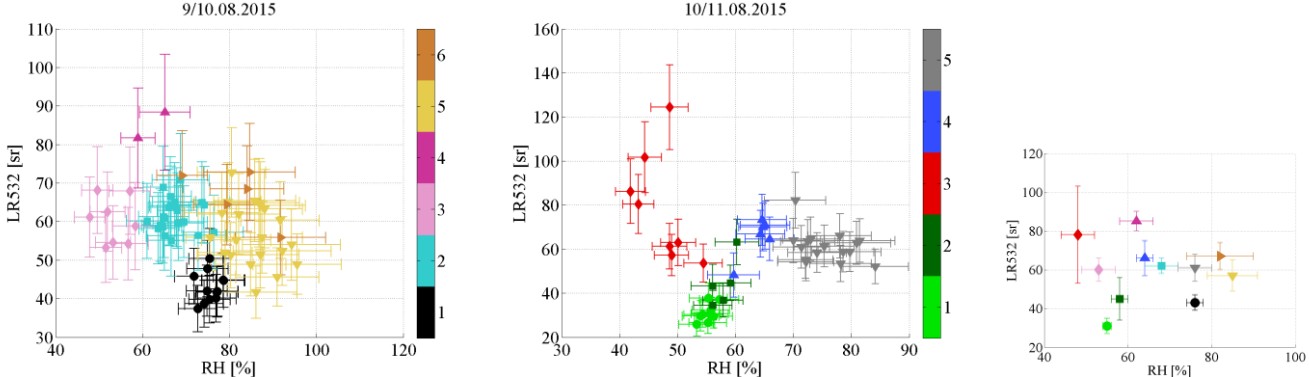

**Figure B3: Scatter plots of the relative humidity versus the lidar ratio at 355 nm (upper panels) and 532 nm (bottom panels).**

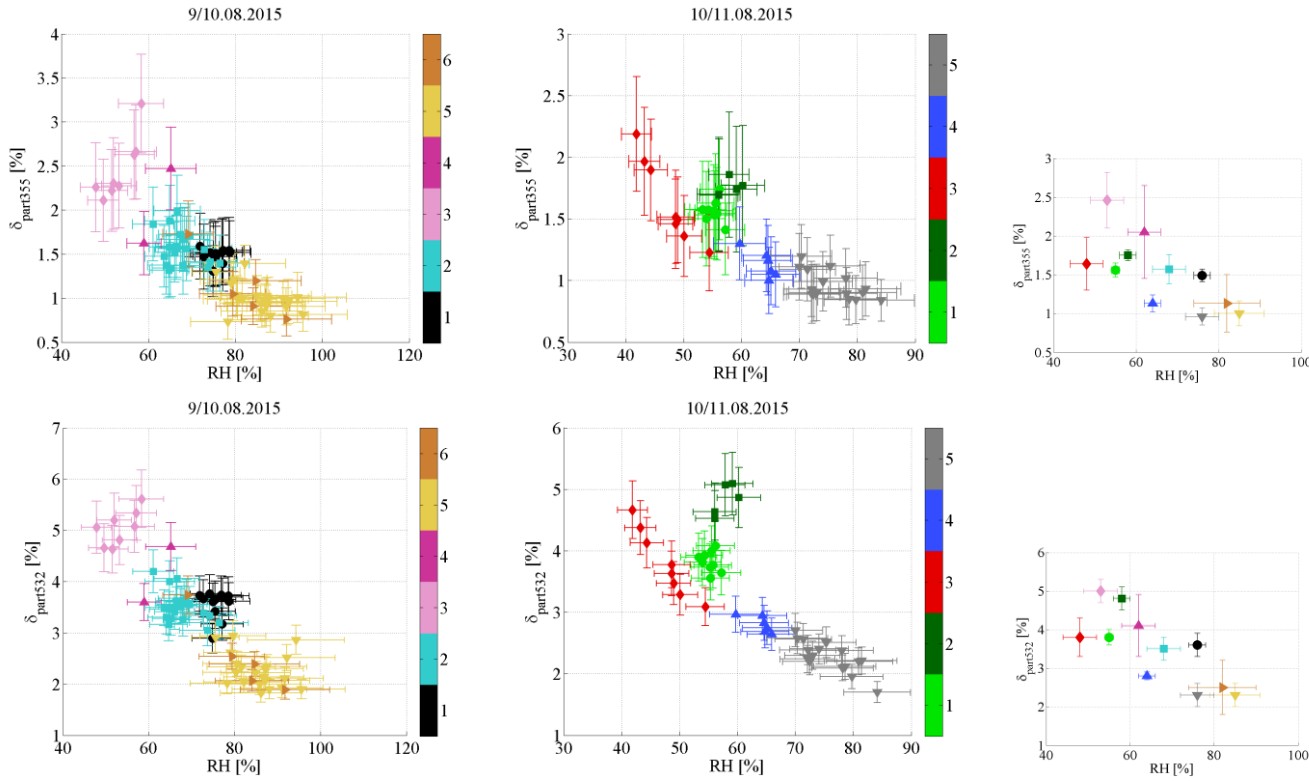

**Figure B4: Scatter plots of the relative humidity versus the particle depolarization ratio at 355 nm (upper panels) and 532 nm (bottom panels).**





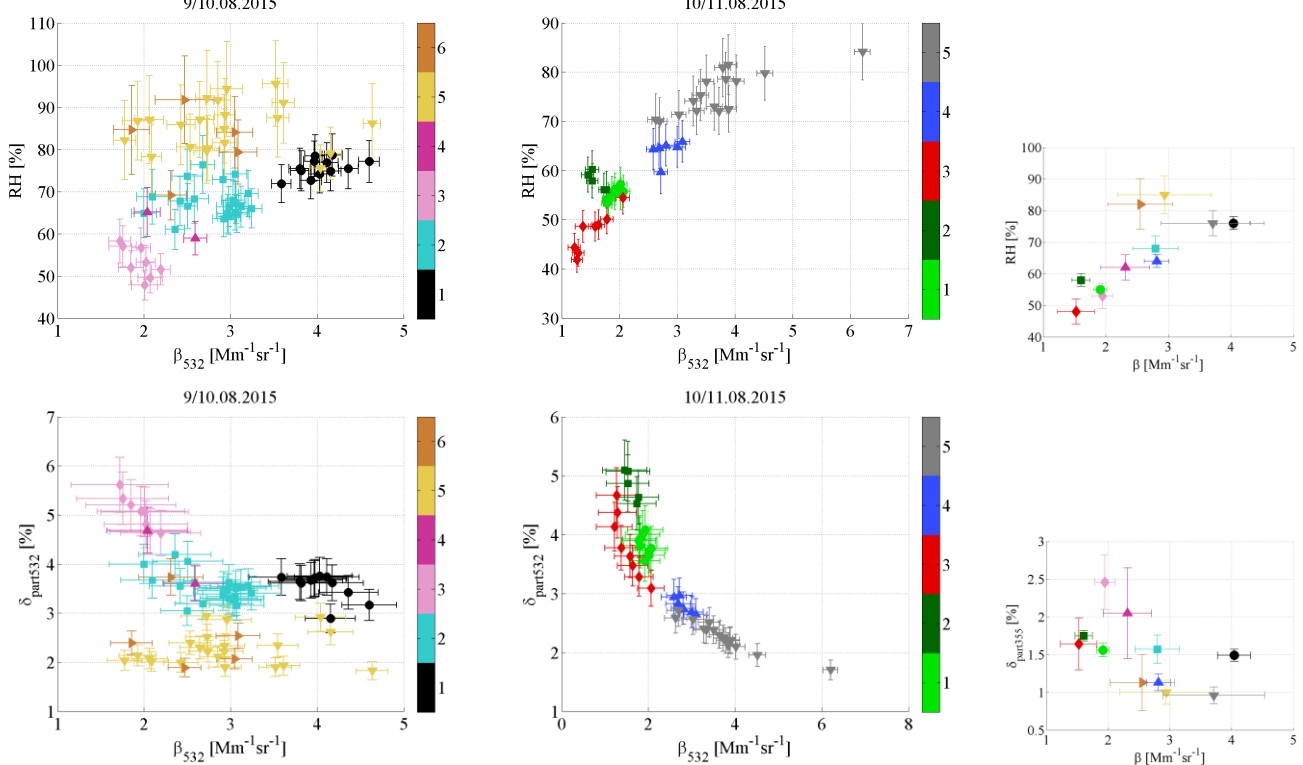

**Figure B5: Scatter plots of the backscatter coefficient at 532 nm versus the relative humidity (upper panels) and versus the particle depolarization ratio at 532 nm (bottom panels).**

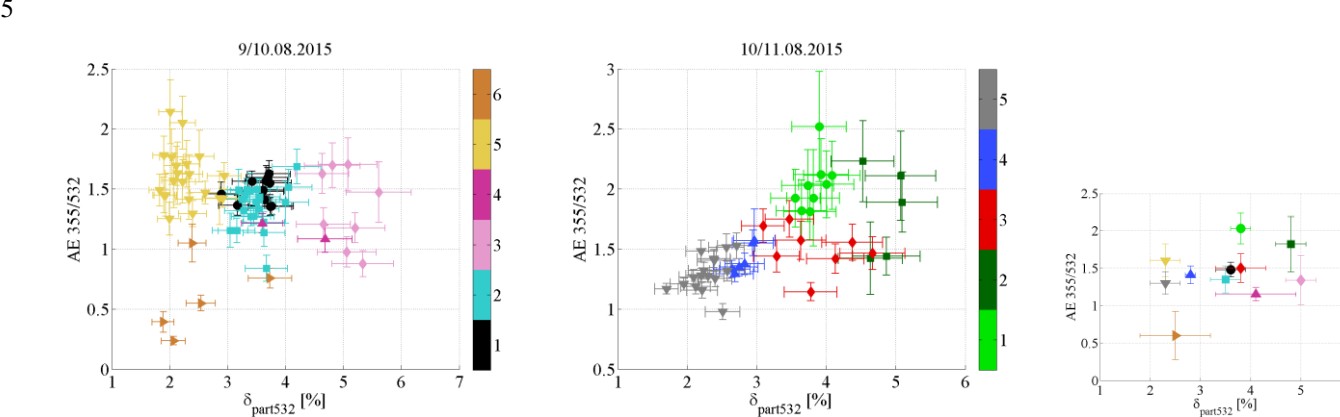





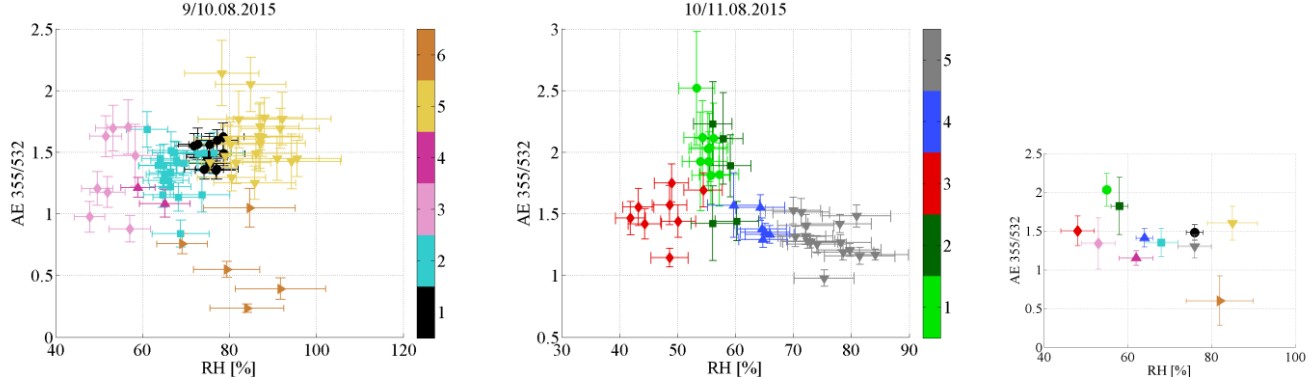

**Figure B6:** Scatter plots of the particle depolarization ratio at 532 nm (upper panels) and relative humidity (bottom panels) versus the extinction related Ångström exponent 355/532.

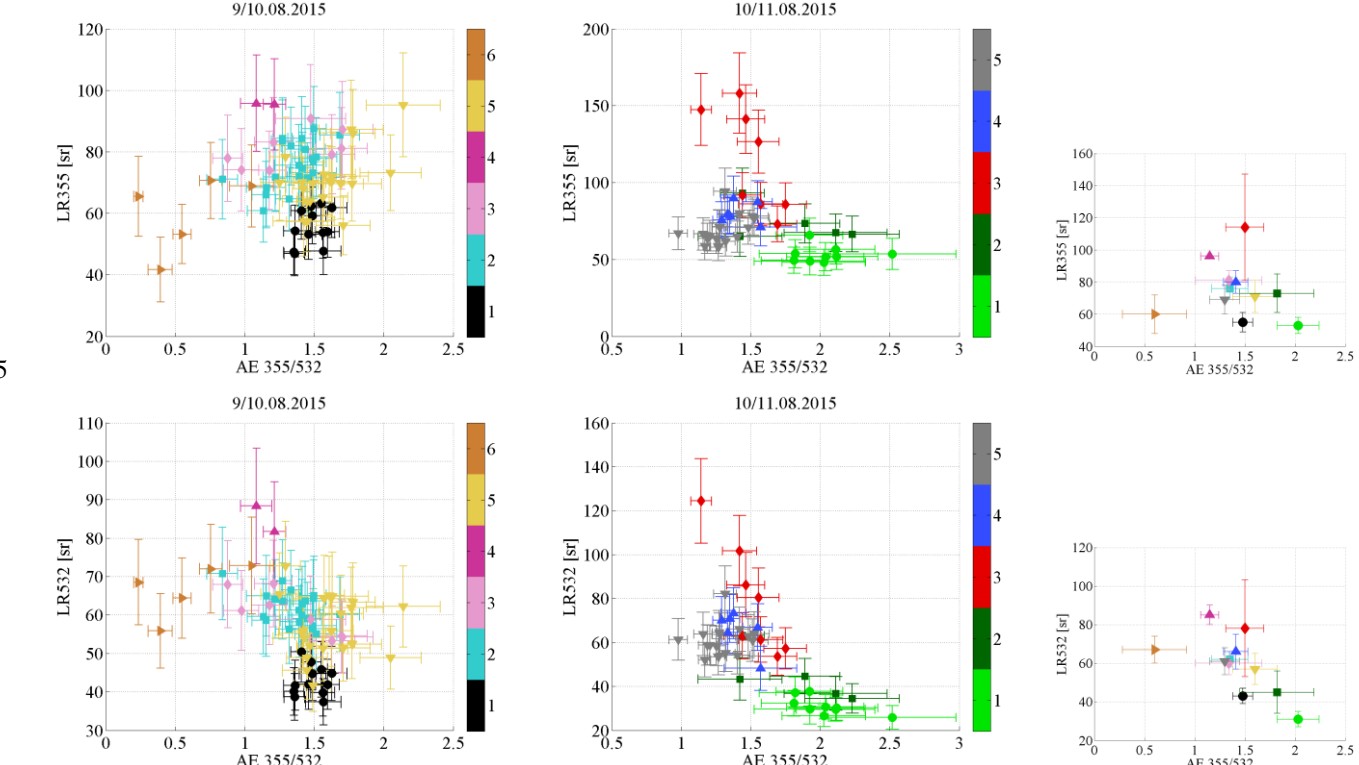

**Figure B7:** Scatter plots of the extinction related Ångström exponent 355/532 versus the lidar ratio at 355 nm (upper panels) and 532 nm (bottom panels).





**Figure B8: Scatter plots of the extinction related Ångström exponent 355/532 versus the backscattering related Ångström exponent 355/532 (upper panels), and versus the backscattering related Ångström exponent 532/1064 (middle panels). Scatter plots of the backscattering related Ångström exponent 355/532 versus the backscattering related Ångström exponent 532/1064 (bottom panels).**





**Figure B9: Scatter plots of the colour ratio of lidar ratios (532/355) versus the particle depolarization ratio at 532 nm (upper panels), versus the relative humidity (middle panels) and versus the extinction related Ångström exponent 355/532 (bottom panels).**