# Peer review of "Properties of biomass burning aerosol mixtures derived at fine temporal and spatial scales from Raman lidar measurements: Part I optical properties"

_Atmospheric Chemistry and Physics, 2019_

## Short Comment (SC1) · 16 Apr 2019

-The Paper by Janicka & Stachlewska (ACPD, 2019) shows an impressive dataset of aerosol optical properties derived forover a 100 layers within only 2 overnights. Each of those layers is selected and charecterised based on unique full-set of optical properties derived from lidar data, namely $3\beta$ + $2\alpha$ + $2\delta$ + wv profiles, which are interpreted with help of the estensive backward trajectory simulations. This amount of lidar-derived data products, as well as much care put on the layer definitions, allowed the authors to assess the spatio-temporal extent of aerosol/mixture types. In total, 11

different aerosol types and aerosol mixtures were identified and characterized by the mean values of optical properties. -Therefore, this study provides an excellent dataset for microphysical inversion. Why authors did not conducted any of those? As there is an indication that this is 'part one' paper, will the 'part two' paper contain extention of this study to microphysical parameter inversion? Moreover, it would be beneficial to put the results obtained on the 2 nights in the context of aerosol properties derived for multi-year analyses. Were such analyses performed? -Finally, the discussed data set comprises diferent mixtures of aerosols, therefore it would be beneficial to test those retreivals on automated aerosol typing algorithms, e.g. as in Nicolae et al. ACP, 2018, especially for multi-year analysis.

---

## Referee Comment (RC1) · Anonymous Referee #1 · 19 Apr 2019

The paper by Janicka and Stachlewska discusses the optical properties of biomass burning aerosol during a heat wave event over Warsaw. The study employs a multi-wavelength Raman lidar and model simulations to identify and characterize the different biomass burning layers. The results constitute an interesting database of biomass burning particles as well as mixtures over Eastern Europe. In many parts, the paper is worded poorly and some important aspects are not fully covered, hence there is lack of clarity in the presentation. I suggest that the English must be improved. I have several technical comments addressing this. The paper is not yet suitable for

publication in ACP unless some major issues are addressed.

Firstly, important and minor revisions are given for consideration in Major and Minor Specific Comments. Secondly, I tried to give Technical Comments although it was impossible to include all of them. Pg1Ln1 means line 1 of page 1.

**Major Specific Comments**

Pg1Ln11: What is a sub-layer? I think in the abstract this term is confusing.

§3: It would be nice to see some discussion on how you retrieve the 'sub-layer' along with an example. Similarly, you should discuss in detail the aerosol typing process. It is not clear to me what information you use to make your aerosol mask. I think, also, that in the last paragraph of this section you can add a summarizing table with the available measurements and the averaging time.

Pg5Ln23: If I understand well, you identified the layers using the particle extinction coefficient and RH data. Can you discuss more on this method? You can give an example.

Pg5Ln26: For the identification of the layer boundaries, which extensive parameter are you using? Is this parameter at full resolution? Does the smoothing affect this decision? I am asking all these because the minimum layer thickness is 89 m.

Pg5Ln27-30: Why do you use mean backscatter and extinction coefficient values in order to estimate the intensive parameters? To my knowledge, the profiles of the intensive parameters, which are type dependent, indicate layer homogeneity etc. Therefore, the intensive properties profiles will serve the purpose of your study.

Pg6Ln28: Why do you use reanalysis and GDAS? I think that reanalysis data perform better and should be preferred.

Pg7Ln16: Where does the statement 'look less reliable' come from? Expand please.

§4.2: I recommend to remove section 4.2 and the description of the event with backward trajectory analysis moved to section 4.1. The comparison of reanalysis and GDAS falls out of the scope of this study and the findings do not support the preference for one dataset over the other.

§4.3: Is the layer characterization assisted with the backward trajectory simulations? If this is the case, the way you classify biomass burning aerosol should be part of the methodology (I think I made this comment already). Could you specify the time window you used in order to characterize the BBA layers as fresh, moderately fresh, etc. I presume that this information is incorporated from the model simulations, therefore I insist that this description should be part of the methodology.

§5: I did not find connection with other studies on the topic. You could create a summarizing table with your findings and the findings from other studies. I think this point is important.

Pg8Ln30: 'The sub-layers... $\alpha$ profiles'. Will you explain how this selection is made?

Pg9Ln8-10: The assumption of dust particles over Warsaw for this specific case is far-fetched when taking into account the simulated trajectories. The depolarization ratio value of 8 % is rather too low. Does this come from a 'sub-layer' (Fig. 5a)? What do the other intensive properties show?

Pg9Ln15: What does this anti-correlation mean? Is it due to dust contamination? Checking Wiegner et al. (2011) there is no such thing observed.

Pg9Ln25-31: Could you expand on the technique for the aerosol typing? Is this technique based only on RH and $\delta$? Are these two parameters sufficient? However, in lines 27-28 you refer to intensive properties, do you use them for the typing? If yes, do you use lookup tables or other information? Furthermore, the different colours refer to different layer aging, is this information solely based on intensive properties or do you

use Hysplit info? I am a bit confused. Please clarify.

Pg10Ln23-25: I would expect that pollen is depolarizing, whereas in your measurements I do not observe depolarizing layers. Can you comment? Moreover, the event presented by Sicard et al. (2016) is during the pollination period (i.e., Spring). Could pollen be observed in August? Can you support this statement?

Pg10Ln27: What does this ratio show? Please, describe. How is this ratio connected to pollen?

Pg11Ln7: Give literature values for fresh BBA and link it with your findings.

Pg11Ln17: Give literature values for aged BBA and link it with your findings.

Pg11Ln17: Give more information on the CRLR threshold.

Pg12Ln23: In Fig. 7c, where is the aging effect in the scatter plot?

Pg12Ln30-31: Please expand.

Pg13Ln1-2: How can you deduce that the particle size is increasing from the $\beta$-RH plot? Do you imply that the RH is linearly correlated with particle size? Please expand. Also, which is the reference?

Pg13Ln25: Try to revisit the Conclusions as this section is very confusing.

**Minor Specific Comments**

Pg1Ln1: I counted 8 distinct layers when I checked Table 1. L1a and L1b are the same as well as the pair L2a and L4b.

Pg1Ln15: You mention fresh, moderately fresh, and moderately aged. In Table 1, you also refer to aged BBA. Why not include it in this sentence?

Pg1Ln15-17: Why do you refer to this specific layer in the abstract?

Pg4Ln11: Do the data shown in this study follow the QA and QC procedures of EAR-LINET?

Pg8Ln14: Do you have any literature reference to support this claim?

Pg8Ln31: Give more information for the PolandAOD database.

Pg9Ln19-24: Aren't the lidar signals too noisy to make any deductions? I would omit this discussion.

Pg10Ln24: You mention high values of depolarization ratio, whilst 4.8 % is definitely a small value. Please clarify.

Pg11Ln5-7: What did the model simulations show?

Pg11L11: I think the statement 'small depolarizing particles' is untrue. Please rephrase.

Pg13Ln20: Can you quantify 'partly consistent'?

Pg13Ln31-34: Better remove this sentence.

Pg14Ln15-16: What does this finding show?

Pg28: Can you overlay the position of the EARLINET sites of Poland? If the empty dots correspond to the sites, make sure to change the size, or colour, or icon.

Pg29: Consider removing this figure or keep the figures related to the reanalysis data.

Pg31: A legend is required for Fig. 5. Also, why don't you report the profiles of lidar ratio and Ångström exponent? The profiles will give an insight into the aerosol layers. Furthermore, the figures are poorly rendered. Make sure to improve the quality of the Figures 5, 6, and 7.

Pg32: Consider a legend describing the colours or incorporate this information in the colour-bar. In Fig. 7a, the layers 7 and 8 are not discernible. In Fig. 7b, merge layers 6-7. You could have the same colour for layers that correspond to the same aerosol

type. E.g. L1 in both layers corresponds to pollution.

**Technical Comments**

Pg1Ln8: Add 'the' after 'during'.

Pg1Ln8: The 'th' is not needed in the date.

Pg1Ln9: Add 'the' before 'so-called'.

Pg1Ln10: Replace 'The' with 'A'.

Pg1Ln11: Write 'optical properties of 116 layers' instead of 'properties within 116 sub-layers in the profiles'.

Pg1Ln13: Add 'the' before 'aerosol/mixture'.

Pg1Ln16: Replace 'characteristic for' with 'characteristic for'.

Pg1Ln16: Delete 'scattered'.

Pg1Ln18: Delete '4.8'.

Pg1Ln18: Replace 'were' with 'was'.

Pg1Ln24: Replace 'studying of' with 'studying the'.

Pg1Ln26: Add 'the' before 'microphysics'.

Pg1Ln26: Delete 'forming in the presence of aerosols'.

Pg1Ln27: Replace 'particles' with 'particle'.

Pg1Ln28: Add 'the' before 'aerosol'.

Pg1Ln28: Delete 'suspension'.

Pg2Ln2: Replace 'what' with 'which'.

Pg2Ln8: Replace 'consist' with 'consists'.

Pg2Ln9: Replace 'is the subject of' with 'is subject to'.

Pg2Ln9: Replace 'processes' with 'process'.

Pg2Ln9: Replace 'lead' with 'leads'.

Pg2Ln10: Replace 'evolutions' with 'change'.

Pg2Ln11: Delete ',' after 'fact'.

Pg2Ln13: Add 'the' before 'development'.

Pg2Ln13: I do not understand 'yield in the'.

Pg2Ln13: Replace 'allow' with 'allows'.

Pg2Ln14: Add 'the' before 'characterization'.

Pg2Ln14-15: Please rephrase 'a signal depolarization by unspherical particles'.

Pg2Ln15: Replace 'unspherical' with 'aspherical'.

Pg2Ln17: Delete 'the' after 'during'.

Pg2Ln19: Replace 'the' with 'a'.

Pg2Ln21: Delete 'the' after 'by'.

Pg2Ln21: Please rephrase 'to increase the research quality and reliability'.

Pg2Ln23-24: Please rephrase 'The automatic aerosol typing algorithms operate with. . .'.

Pg2Ln24: The abbreviation is missing. Make sure to include abbreviations throughout the manuscript with specific references and website links when needed.

Pg2Ln27: Add 'a' before 'maximum'.

Pg2Ln28: Replace 'This' with 'these'.

Pg2Ln29: Delete 'a very'.

Pg2Ln29: Please rephrase 'to use the lidar data by the specialist'.

Pg2Ln31: Delete 'a' before 'high'.

Pg2Ln33-34: Please rephrase the last sentence.

Pg3Ln1-2: Please rephrase the first sentence.

Pg3Ln2: Delete 'A'.

Pg3Ln3: Delete 'a' before 'long'.

Pg3Ln5-6: What do you mean 'found in the systematic observations'?

Pg3Ln6-8: Also, what does this sentence mean? The message is not clear.

Pg3Ln9: Delete 'a' before 'favourable'.

Pg3Ln14: Replace 'according with' with 'according to', however, I do not think that fits here.

Pg3Ln16: Replace 'were' with 'was'.

Pg3Ln16: Replace 'based on' with 'using'.

Pg3Ln18: 'which coincide with the analysed in the paper'. What do you mean?

Pg3Ln19: Replace 'aerosol' with 'aerosols'.

Pg3Ln19: Write 'as biomass burning smoke from Ukrainian wildfires'.

Pg3Ln20-21: What do you mean by 'deepening study'?

Pg3Ln21: Please rephrase 'to reflect atmospheric variability and to catch all individual sublayers'.

[Figure]

Pg3Ln22: What do you mean by 'can be likely kinds of mixtures'?

Pg3Ln23: Delete 'the' before 'separation'.

Pg3Ln28-29: Please rephrase 'as the one... in each sub-layer'.

Pg3Ln29: Delete 'mentioned'.

Pg3Ln30: Please rephrase 'performed as a standard'.

Pg3Ln31: Add 'the' before 'study'.

Pg3Ln32-34: Please rephrase the last sentence. Also, a reference is missing.

Pg4Ln1: Replace 'contain' with 'contains'.

Pg4Ln1-6: The abbreviation 'Sect.' should be used when it appears in running text.

Pg4Ln5: Replace 'is' with 'are'.

Pg4Ln5: Please rephrase 'outlooks potential use of the obtained results'.

Pg4Ln10: Replace 'as' with 'following'.

Pg4Ln15: Replace 'consist of the' with 'consists of a'.

Pg4Ln20: Add 'an' after 'where'.

Pg4Ln23-24: Add 'the' before 'particle' and 'water'.

Pg4Ln26: Replace 'Baars et al., 2016' with 'Baars et al. (2016)'. Make sure when you are citing a paper to comply with the guidelines of ACP. Several mistakes of this kind were found throughout and it is not possible to list them all out.

Pg4Ln28: For the dates, consider using dashes and not slashes. E.g. 9-10 August.

Pg5Ln2: Replace 'meteorology' with 'meteorological'.

Pg5Ln4: Add 'the' before 'retrieval'.

[Figure]

Pg5Ln9: Add 'the' before 'free'.

Pg5Ln14: You refer to 'the latter', which is the former?

Pg5Ln19: Delete 'of' after 'Comparing'.

Pg5Ln19: Delete 'the' after 'showed'.

Pg5Ln20: Add 'the' before 'calculation'.

Pg5Ln20: Delete 'the' before 'mean'.

Pg5Ln23: Add 'for' before 'further'.

Pg5Ln24-25: Delete 'of the intensive optical properties'.

Pg5Ln25: Replace 'sets' with 'set'.

Pg5Ln32: What do you mean by 'aerosol layers occurrence'?

Pg5Ln33: Add 'the' before 'residual'.

Pg6Ln8: Replace 'On the' with 'At the'.

Pg6Ln8: Add 'a' before 'persistent'.

Pg6Ln10: Write 'Inflow of warm air from Western Africa'.

Pg6Ln10: What do you mean by 'was dominating in the troposphere'?

Pg6Ln14: Delete 'the' after 'within'.

Pg6Ln14: I cannot understand 'insensitively'?

Pg6Ln15: Delete 'territory'.

Pg6Ln15: Replace 'In the three days period' with 'In the next three days'.

Pg6Ln15: Replace 'at the direct' with 'located in direct'.

Pg6Ln16 and Pg6Ln18: Delete 'panel'.

Pg6Ln18: Replace 'depict' with 'depicts'.

Pg6Ln18: Add 'the' before 'distinct'.

Pg6Ln19: Replace 'splitted' with 'split'.

Pg6Ln22: Delete 'occurrence'.

Pg6Ln22: What does the sentence 'air settlement in the conditions of high pressure' mean?

Pg6Ln23: Delete 'of aerosol structures'.

Pg6Ln25: Replace 'directions of the aerosols inflow' with 'aerosol origin'.

Pg6Ln31: What is a 'high interval'?

Pg7Ln1: What does 'reflect' mean?

Pg7Ln2: Replace 'have' with 'has'.

Pg7Ln3: Add 'the' before 'general'.

Pg7Ln3: Please rephrase 'ran starting on the territory'?

Pg7Ln5: Replace 'At the altitudes' with 'For the altitudes'.

Pg7Ln6: Add 'the' before 'trajectories'.

Pg7Ln6-7: Please rephrase 'from the mentioned before direction of the lower trajectories to the direction from western Europe'.

Pg7Ln7: Add 'the' before 'Czech'.

Pg7Ln7-8: Please rephrase 'Over the time. . . over Spain'.

Pg7Ln9-11: Please rephrase 'At the altitude. . . Sahara Desert'.

Pg7Ln11: Replace 'originate' with 'originates'.

[Figure]

Pg7Ln14: Add 'the' before 'consistency'.

Pg7Ln17: Write 'the two simulations is the origin of the aerosols over Warsaw'.

Pg7Ln21: Replace 'analogous' with 'same'.

Pg7Ln23: Delete 'at the analogous altitudes'.

Pg7Ln28: Delete 'with'.

Pg7Ln28: Add 'the' before 'GDAS'.

Pg8Ln2: The period should be 10-11 August and not 9-11.

Pg8Ln3: Replace 'panel (a)' with 'Fig. 4a'.

Pg8Ln4-5 and Ln14: The same change applies to these sentences.

Pg8Ln5: What do you mean by 'The trajectory altitude is rising. . .'?

Pg8Ln6: Replace 'as in' with 'to'.

Pg8Ln7: Delete 'the' before 'satellite'.

Pg8Ln9-10: Please rephrase.

Pg8Ln12: Replace 'point that biomass' with 'point out that the biomass'.

Pg8Ln15: Replace 'most likely originates' with 'might have come'.

Pg8Ln16: Delete 'the' before 'copper'.

Pg8Ln16: Add 'the' before 'possible'.

Pg8Ln16: Delete 'the' before 'pollution'.

Pg8Ln16: Add 'the' before 'altitude'.

Pg8Ln17: Replace 'origin' with 'originate'.

Pg8Ln19: Add 'the' before 'Iberian'.

[Figure]

Pg8Ln20: Replace 'origin' with 'originate'.

Pg8Ln21: Replace 'Hungry' with 'Hungary'.

Pg8Ln22: Delete 'the' before 'moderately'.

Pg8Ln22-23: Please rephrase

Pg8Ln26: Replace 'origin' with 'originate' everywhere.

Pg8Ln29: What do you mean by 'representative for the layers'?

Pg8Ln29: Delete 'the' before 'mean'.

Pg9Ln1: Replace 'in further study' with 'for further study'.

Pg9Ln1: Delete 'of the intensive properties'.

Pg9Ln11: Add 'the' after 'course of'.

Pg9Ln11: Add 'A' before 'similar'.

Pg9Ln14: Add 'the' before 'data'.

Pg9Ln17: Add 'the' before 'slight'.

Pg9Ln17: Replace 'from over Sahara' with 'from the Sahara Desert'.

Pg9Ln25: Replace 'performed' with 'made'.

Pg9Ln31: Please explain 'have some features'.

Pg10Ln8: Replace 'splitted' with 'split'.

Pg10Ln14: Replace 'in the first' with 'for the first'.

Pg10Ln14: Replace 'cooper' with 'copper'.

Pg10Ln23: Replace 'domination' with 'dominance'.

Pg10Ln23-25: Consider to rephrase this sentence.

Pg10Ln26: Please rephrase 'To easy compare'.

Pg10Ln28: Replace 'has' with 'have'.

Pg10Ln32: Replace 'have' with 'has'.

Pg10Ln34: What do you mean by 'stronger contamination'?

Pg11Ln1: Delete 'i' from '0.12i' and '0.30i'.

Pg11Ln5: Replace 'of' with ','.

Pg11Ln7: Replace 'weakly' with 'less'.

Pg11Ln8-12: Please rephrase, the message is not clear.

Pg11Ln13-14: Please rephrase.

Pg11Ln18: Replace 'this' with 'these'.

Pg11Ln18: 'This. . . sr).' What do you mean?

Pg11Ln18: Delete 'of' after 'was'.

Pg11Ln21: Delete 'of' before 'the BBA'.

Pg11Ln27: Delete 'one' before 'case'.

Pg12Ln4: Delete ',' after 'shows'.

Pg12Ln4: Delete 'rather'.

Pg12Ln7: Add 'the' before 'linear'.

Pg12Ln7: Delete 'the' before 'details'.

Pg12Ln9: I do not get the meaning of 'alternatively'.

Pg12Ln9: Write 'the dependence is not linear'.

Pg12Ln9-12: Please rephrase.

Pg12Ln13: Please rephrase.

Pg12Ln14: Replace 'mixture' with 'mixtures'.

Pg12Ln16: Replace 'another' with 'other'.

Pg12Ln16: Please rephrase 'another stage… atmosphere'.

Pg12Ln19: Replace 'splitted' with 'split'.

Pg12Ln21: Replace 'this' with 'these'.

Pg12Ln23: Replace 'hygroscopicity' with 'the hygroscopic growth'.

Pg12Ln24-25: Please rephrase the two sentences. It is impossible to understand.

Pg12Ln25-27: Please explain.

Pg12Ln27: Delete 'the' before 'model'.

Pg12Ln29: Replace 'for' with 'to'.

Pg12Ln29: Add 'the' before 'imaginary'.

Pg12Ln32: Replace 'shows' with 'show'.

Pg12Ln33: Delete 'the' before 'negative'.

Pg13Ln1-2: Please rephrase. It is impossible to understand.

Pg13Ln2-4: Please rephrase. It is impossible to understand.

Pg13Ln6: What do you mean by 'assessed'?

Pg13Ln17: Add 'the' before 'troposphere'.

Pg13Ln17: Add 'the' before 'aerosol'.

Pg13Ln17: Delete 'the' before 'observed on that days'.

Pg13Ln17: Replace 'were' with 'was'.

Pg13Ln18: Pleas rephrase 'present a consistent course'.

Pg13Ln19: Add 'the' before 'radiometer'.

Pg13Ln20: Replace 'this' with 'the'.

Pg13Ln22: I cannot understand 'what'.

Pg13Ln26: Delete 'an' before 'aerosol'.

Pg13Ln26: Delete 'the' before 'biomass'.

Pg13Ln28: Replace 'show' with 'showed'.

Pg13Ln29: Write 'that 2-3 days old air from Germany. . .'.

Pg13Ln30: Delete 'about'.

Pg13Ln31: Replace 'is' with 'in'.

Pg14Ln1: 'specifying. . . sub-layers'. Please rephrase.

Pg14Ln3: Add 'A' before 'total'.

Pg14Ln4: Replace 'o' with 'of'.

Pg14Ln4-5: 'general. . . approach'. Please rephrase.

Pg14Ln8-9: 'slight. . . possible'. Please rephrase.

Pg14Ln9: Add 'the' before 'upper'.

Pg14Ln9: Replace 'were' with 'was'.

Pg14Ln10-11: 'In one layer. . . information'. Please rephrase.

Pg14Ln11: Replace 'do' with 'does'.

Pg14Ln11: Replace 'an' with 'a'.

Pg14Ln11-13: Please rephrase 'occurs. . . formation)'.

Pg14Ln17: Replace 'of some kinds of mixtures' with 'mixed'.

Pg14Ln23-24: Please rephrase.

Pg27: Insert the following 'black dots depict the', 'calculated with the', 'purple bars depict the', 'estimated from'.

Pg27: Replace 'selected to the evaluation' with 'selected for evaluation'.

Pg29: Please improve the caption of Figure 3.

Pg30: What do you mean by 'the trajectory altitude is rising north-westward'?

Pg31: Replace 'in further analysis' with 'for further analysis'.

Pg32: Insert the following 'the same colour depicts' and 'layer 8 depicts'.

Pg32: Delete 'further' wherever appears in the caption.

Pg33: Please rephrase the last sentence of the caption.

---

## Referee Comment (RC2) · Anonymous Referee #2 · 25 Apr 2019

The manuscript introduces a dataset of lidar observations of aerosol backscatter, aerosol extinction, and aerosol depolarization at multiple wavelengths plus relative humidity, for a heat wave event in Warsaw comprising many identified aerosol layers over two nights. It provides aerosol type identification for these layers and gives the mean lidar properties for layers of each type.

Unfortunately, most aspects of aerosol type analysis are not clear and there are many results with inadequate support or no support whatsoever. The overall motivation or objective of the manuscript also seems confused. The lack of clarity about the objec-

tives and especially the flaws in the analysis make it impossible for me to recommend the manuscript for publication in ACP.

Before getting into the details, I will say that I agree that the dataset itself (as distinct from the aerosol type analysis) is potentially quite valuable, and I am particularly impressed by the inclusion of relative humidity profile information which adds significant potential for science value when combined with the lidar aerosol measurements. One option might be to make a streamlined manuscript without the aerosol type analysis, but with a more in-depth uncertainty analysis for the primary lidar measurements (backscatter, extinction, intensive parameters, and RH), plus an explanation of the quality control, and submit this alternate paper as a data paper in a data journal such as Earth System Science Data.

I will organize my more detailed comments by manuscript section.

Section "Introduction":

My primary criticism in the introduction is that it does not adequately state the objective and motivation. The key statement of the objective in the introduction seems to be at page 2, lines 30-33, "The algorithms that deal with the inversion problem of microphysical retrieval require an accurate aerosol layer selection and a high quality 3beta + 2alpha optical data set as well as the depolarization information [Veselovskii et al. 2002; Bockmann et al. 2005; Muller et al. 2016]. Thus the manual data evaluation which allows for an insightful analysis of the lidar signals with the individual approach to the considered case is still much-needed." In this study, the high quality 3beta + 2alpha optical data set and the QA on that dataset is provided by the retrieval algorithm which is considered to be an input to this study and not part of the study itself. Similarly, although the layer selection is done here, the explanation of how it's done is presented as a black box. The statement is a non-sequitur if the "insightful analysis" is taken to mean the aerosol typing analysis, since that is not needed for microphysical retrievals. I also don't understand why automated aerosol typing algorithms are acknowledged

but discarded. In short, I just don't understand the motivation for doing this study this way.

A less important point is that there is a lot of information in the introduction, such as the first several paragraphs about the effect of aerosol properties on radiative forcing, whose relevance to this study is not explained at all. On a positive note, I appreciate the introduction to the specific heat wave event being studied, and also the statement about the uniqueness of the dataset.

Section "Methodology":

The methodology describes the retrieval methodology for obtaining the measurements of e.g. backscatter and extinction but not the analysis methodology. Most of the analysis in this paper is about aerosol type attribution, but there is no description of the methodology for this (here or anywhere in the paper), leaving me completely without a foundation for understanding the rest of the paper. For a few other key aspects of the analysis, the hints in the methodology are inadequate, for example "The sub-layers selection was based mainly on RH and extinction profiles". This is too vague to be useful for understanding how the layers are selected.

Also, I think it is important to understand the uncertainties on the lidar measurements. Page 5, line 30 says the uncertainties for the intensive parameters are propagated from the backscatter and extinction uncertainties, but does not say how the backscatter and extinction uncertainties are calculated, and anyway, it doesn't seem that the uncertainties are actually presented in the paper. Error bars in the figures and tables seem to be just the standard deviation calculated for selected layers, not measurement uncertainties. Standard deviation might be a reasonable stand-in for measurement random uncertainties, but for some of the analysis, such as understanding quite small particle depolarization values, the authors need to develop an understanding of the expected systematic uncertainty.

The statement "The columnar value of the extinction related Angstrom exponent of 1

was assumed for the extinction coefficient profiles calculations" confuses me. I don't understand why you would need this assumption and it seems that this assumption would have too much impact on the layer angstrom exponents. I think I'm probably just misunderstanding what this means. I would like to be able to read a reference about the retrieval to be sure, but I can't find one cited. Did I miss it?

Section 4.3 "Aerosol source analysis"

On first read-through of the paper, I thought the aerosol source analysis was meant to be a major part of the analysis. There is a significant amount of analysis to calculate and collate the backtrajectories in an attempt to identify source regions, and this could potentially be very informative in the inference of aerosol type and source for the observed layers. However, the analysis stops far short of that. There is very little attempt to link the backtrajectories to specific layers and the two segments of analysis (aerosol source analysis vs. aerosol typing) seem to be separate, disjointed sections.

There are also several confusing or contradictory statements. For example, page 8, lines 14-15 say where two-days old smoke would have come from but then lines 17-19 says that above 3 km, it's about 3 days old and below 3 km, it's older; so where is the 2-days old smoke? Then at line 20 is a statement that aged biomass burning aerosol is possible in the lowermost levels but at line 22, it's "moderately fresh". I'm not sure if all of this is actually contradictory or if the authors are expressing the thought that there are multiple possibilities and a lot of uncertainty. If there are any layers at all where the airmass analysis allows for assessing probable sources or a range of ages of the aerosol layer, it would be very helpful to see that information included explicitly in Table 1 where it can be easily used for understanding the aerosol types.

Section 5 "Optical properties of the BBA and pollution mixtures"

In this section, individual aerosol layers are labeled according to aerosol types. In a few cases, the backtrajectories are referred to, but mostly the methodology seems to be based on haphazardly comparing the aerosol intensive parameters to a few case

studies in prior literature. There is no discussion of how exactly this is done, no consistent thresholds are presented, and there is no quantitative comparison with prior literature that present ranges of parameters that might be expected for different types (e.g. Muller et al. 2007, Gross et al. 2012, Burton et al. 2012).

Unfortunately, there are many, many examples of type labels being applied without adequate explanation, support, or comparison:

Page 9, line 29: "properties typical for the aged biomass burning aerosol (CRLR >1)". No support is given for this value being "typical" for aged biomass burning aerosol. Although it is not cited in reference to this statement, I believe this argument follows from the case studies presented by Nicolae et al. (2013). However, since those are just a few case studies, it may be an overgeneralization to describe this relationship as "typical".

Page 10, line 22: "The relatively low LR and high AE . . . indicates pollution domination." No reference is given for LR and AE of pollution or a description of how to determine them.

Page 10, line 22: The low lidar ratios that are said to indicate pollution seem very low compared to prior literature that I know of. No references are given here. How do these values compare to other published values for pollution cases?

Page 10, line 24, "relatively high value of 532 nm particulate depolarization of 4.8 plus or minus 0.3 ... may reflect contamination by pollen". This is not a very high value of particulate depolarization, and is only for the earlier layer (L1b), and the larger AE for the earlier layer would suggest smaller particles rather than larger. These seem like subtle and confusing trends. Is it clear that these distinctions are actually significant compared to measurement uncertainty (random + systematic)? Also, why is the conclusion that pollen is present, and not some other depolarizing type such as dust? 4.8% is small enough that even smoke could have such a value. The Sicard reference cited here does not help me understand this, since it's not about this case, but is just a

general reference for lidar measurements of pollen. (And a minor related point: if the purpose is to credit lidar measurements of pollen, much earlier papers such as Sassen et al. 2008 should get that credit, doi: 10.1029/2008gl035085).

Page 10, line 26-29 Contradictory statements that the ratio of particulate depolarization at two wavelengths "did not vary significantly" but the difference between them "confirms the hypothesis of pollen contamination". If the variation is not significant, it doesn't confirm the hypothesis.

Page 10, line 31. Higher lidar ratio values "indicate domination by the biomass burning aerosol." Again, based on what methodology and what references? All the papers I'm familiar with (e.g. Muller et al. 2007, Gross et al. 2012, Burton et al. 2012, Papagiannopoulos et al. 2018) have a significant overlap for the lidar ratio of biomass burning and pollution, at whichever wavelength they look at.

Page 11, line 1, imaginary refractive index values of 0.12i or 0.20-0.30i seem extremely high. This is quote from a paper in preparation. It doesn't seem appropriate to present something so unexpected without any support or discussion.

Page 11, line 5, "as the LR values were relatively high of 81 plus or minus 6 sr and 60 plus or minus 6 sr at 355 nm and 532 nm, respectively, the layer can be treated as fresh biomass burning aerosol . . . although the angstrom exponent value of 1.34 plus or minus 0.3 is rather low for fresh BBA". How is this conclusion reached? What values are these measurements being compared to? Couldn't this be urban pollution?

Page 11, lines 15-16 "Such low depolarization ratios are characteristic for aged biomass burning". No references are given to support this statement. Aren't low depolarization ratios also characteristic of pollution and other aerosol types?

Page 11, line 21. "The AE value of about 0.6 [i.e. the same value as measured in this study] was reported by Muller et al. 2007". Actually, Mueller et al. 2007 reported a value of 1.0 plus or minus 0.5. Maybe that could be considered "about 0.6" but it's a

misleading way to say it.

Page 12, line 4, appears to suggests that the backscatter angstrom exponent is expected to be monotonically related to the aerosol age. I don't know of any reason to think this. Are there references?

This section also contains statements two pages apart which directly contradict each other, giving me the impression that the authors do not fully understand their subject material and this manuscript was really not ready for submission in the first place. Regarding an apparent correlation between backscatter and depolarization, on page 11, "it is probable that increased depolarization is associated with the higher participation of background aerosol in the tropical air containing some mineral dust particles." And on page 13, "the negative dependence of backscatter and particulate depolarization is unlikely related to mineral dust contamination".

Another question about the measurements. On page 11, line 28, a lidar ratio of 114 plus or minus 33 sr at 355 nm. Isn't this really very high? Are there other published observations of 355 nm lidar ratios this high? I see from the figure that this very high value occurs where there is significant slope in the backscatter and extinction profiles, on the underside of a layer with larger backscatter. The peak of the backscatter and extinction have very different shapes, with the extinction looking like a smoothed version of backscatter. If there is a difference in the resolutions of the backscatter and extinction profiles such that the edges of the layers are not being captured the same way, it creates spurious values in the ratio. In this case, it looks like the lidar ratio is large in consequence of the fact that the backscatter and extinction show the edge with different slopes. Are the backscatter and extinction resolutions compatible when the lidar ratio is computed?

The wording is quite rough. If this manuscript were to be published or resubmitted, it would be very helpful to have a round of English-language editing.

Finally, where are the data available? Given that the paper is primarily about introducing a new dataset, I would like to see a statement of data availability.

---

## Author Comment (AC1) · 29 Apr 2019

We are thankful for your comment. We truly appreciate that you and both Anonymous Reviewers value the dataset presented in our paper, the dataset that is derived manually using the Raman lidar data evaluation in combination with lidar water vapour profiling.

Aerosol interpretation (not typing) that is currently presented in the paper was based on the trajectory simulations, the inspection of the MODIS fire data and the literature

review. In this sense it is the first step of complex analysis and, indeed, the 'part two' will focus on the microphysical inversion of presented dataset. We regret, that this information was not clearly stated in our paper.

We agree that it would be beneficial to use the derived dataset as an input for automated aerosol typing algorithms, such as the NATALI code (Nicolae et al. ACP, 2018). We are open to cooperation in this matter.

Finally, the multi-year analysis is not in a scope of the current paper. However, such analysis, based on the standard EARLINET data from Warsaw site as available via the ACTRIS Database is currently under preparation.

---

## Author Comment (AC2) · 15 May 2019

**Response to Reviewer Report #1 by Janicka & Stachlewska on 14 May 2019**

**Dear Referee, Dear Editor,**

**We are grateful to Referee for comments and suggestions that allowed us to improve this manuscript. In the following, the answers to Referee's comments and issues raised are reported directly below each related comment (in bold). All modifications of the initial version of the manuscript as well as additions are reported in color highlight in the revised version of the manuscript. We believe that we have fulfilled required changes in the final version of the manuscript.**

The paper by Janicka and Stachlewska discusses the optical properties of biomass burning aerosol during a heat wave event over Warsaw. The study employs a multi-wavelength Raman lidar and model simulations to identify and characterize the different biomass burning layers. The results constitute an interesting database of biomass burning particles as well as mixtures over Eastern Europe.

**Thank you. We appreciate that you value the dataset presented in our study, which was evaluated manually to create a specific data base that allows to assess the representativeness of each result.**

In many parts, the paper is worded poorly and some important aspects are not fully covered, hence there is lack of clarity in the presentation. I suggest that the English must be improved. I have several technical comments addressing this. The paper is not yet suitable for publication in ACP unless some major issues are addressed.

**Thank you for your comment. Of course we are not native speakers and it is our mistake, not sent manuscript for professional English editing before the first submission of the manuscript. Language proof will be done by the professional English proofreading service before submission revised version. We realize that this was one of the reasons due to which important aspects were not well described.**
**One of them is that the aim of this paper was never set on aerosol typing. There was never a procedure for aerosol typing introduced nor aerosol mask was applied. What we did was a rather classical layer interpretation approach done for every single layer separately. It was done in a manual way and interpreted based on the backward trajectory simulations, the satellite data and the literature review.**
**The second aspect is the microphysical inversion, which as performed on the dataset discussed in this paper (part I) will be the subject of the 'part II' manuscript. We truly regret that this information was not clearly stated in the text, which was confusing to the reader.**

Firstly, important and minor revisions are given for consideration in Major and Minor Specific Comments. Secondly, I tried to give Technical Comments although it was impossible to include all of them. Pg1Ln1 means line 1 of page 1.

**Thank you very much for such many and detailed comments. Below we are addressing them point by point.**

**Major Specific Comments**

Pg1Ln11: What is a sub-layer? I think in the abstract this term is confusing.

**Indeed, we agree. In the abstract the term 'sub-layer' is changed to 'layer'.**

§3: It would be nice to see some discussion on how you retrieve the 'sub-layer' along with an example.

**Thank you for the comment. We add in the text of Methodology Pg5Ln21 following clarification:**
**'The definition of the sub-layer in this paper is not standard as it is not based on the typically used gradient of the signal. The sub-layers were discriminated to fulfill the quality requirements to further microphysical retrieval (part II paper). The sub-layer selection is appropriate for such inversion if the extensive optical properties ($\alpha$, $\beta$) follow the same tendency at each wavelength (i.e. constant, increasing or decreasing).**
**In the sub-layer discrimination, the first step was to select the constant sections in the relative humidity profile and compare them with the extinction coefficient profiles. In the next step, the correspond to the backscatter coefficients profiles as well as to the sub-layers visible in the depolarization ratio were checked and corrected if necessarily. Examples of how the sub-layers were selected are shown on the profiles in Fig. 5 and in the Appendix A.**
**Finally, the sub-layers were collected into groups (layers) characterized by similar properties and described in the statistical way by the mean values of the intensive optical properties and the relative humidity with the standard deviation. The layers were discriminated based on the relative humidity and depolarization plots, as the differences between the layers were the most pronounced using these properties. Then, the first choice was revised with the spread between the properties using the plots of mutual relations of intensive optical properties and relative humidity shown in Appendix B. If particular layer had too much noise (especially upper layers) it was no further analyzed.'**

Similarly, you should discuss in detail the aerosol typing process. It is not clear to me what information you use to make your aerosol mask.

**Thank you, it was misleading in the text.**
**The aim of this paper was never set on aerosol typing. There was never a procedure for aerosol typing introduced nor aerosol mask was applied. Therefore, the data showed in Table 1 are interpretational results based on the backward trajectories simulation and satellite data.**

I think, also, that in the last paragraph of this section you can add a summarizing table with the available measurements and the averaging time.

**We are not sure that we correctly understood the comment. The time averaging for each set of profiles (available measurements) is explicitly given in Appendix A on each plot. Moreover, explanation of used averaging times is given in section Methodology in Pg4Ln27 to Pg5Ln3. We fell there is no need to repeat this information in an extra table. Unless, we did not understood this comment of the Reviewer.**

Pg5Ln23: If I understand well, you identified the layers using the particle extinction coefficient and RH data. Can you discuss more on this method? You can give an example.

**It has been answered above.**

Pg5Ln26: For the identification of the layer boundaries, which extensive parameter are you using? Is this parameter at full resolution? Does the smoothing affect this decision? I am asking all these because the minimum layer thickness is 89 m.

**For all profiles initial smoothing with running mean of 112.5 m was applied only on the raw signals (Pg5Ln10-11). The extinction coefficient profiles were additionally smoothed as given in section Methodology Pg5Ln16-20. Sub-layers boundaries are, at first, obtained on relative humidity and extinction coefficient profiles. Then, the choice was revised with the backscatter coefficient profiles (no additional smoothing).**

**Moreover, as the sub-layer choice was done to come up within the layers with coherent properties (useful for microphysical inversion) we were not looking at the gradient but for the coherent parts of the profiles, thus the effect of the smoothing is negligible even for this very thin layers.**

Pg5Ln27-30: Why do you use mean backscatter and extinction coefficient values In order to estimate the intensive parameters? To my knowledge, the profiles of the intensive parameters, which are type dependent, indicate layer homogeneity etc. Therefore, the intensive properties profiles will serve the purpose of your study.

**The procedures used for aerosol typing are based on the intensive properties profiles but here we defined the layers based on the extensive optical properties as being useful for the microphysical retrieval. The intensive properties were calculated for inspection of the results in already defined layers. It is also the reason that in the plots in the Appendix A the intensive properties are shown as the mean values not the profiles.**

Pg6Ln28: Why do you use reanalysis and GDAS? I think that reanalysis data perform
better and should be preferred.

**Please note, that in this paper for interpretation we used only the simulation with the reanalysis data. The simulation with the GDAS model was shown only to stress that for the chosen cases it resembled the results obtained with reanalysis in surprisingly good manner.**
**Therefore, we agree to remove the HYSPLIT trajectories modeled with GDAS data, which shall also reduce this already lengthy paper. Thank you.**

Pg7Ln16: Where does the statement 'look less reliable' come from? Expand please.

**As suggested, the analysis of trajectories with GDAS model is excluded (Pg7Ln13-19 and Pg7Ln26-3).**

§4.2: I recommend to remove section 4.2 and the description of the event with backward trajectory analysis moved to section 4.1. The comparison of reanalysis and GDAS falls out of the scope of this study and the findings do not support the preference for one dataset over the other.

**As recommended, the analysis of trajectories with GDAS model is excluded. Then the paragraph 4.1 and 4.2 are combined under name 'Meteorological situation and air-mass transport'.**

§4.3: Is the layer characterization assisted with the backward trajectory simulations? If this is the case, the way you classify biomass burning aerosol should be part of the methodology (I think I made this comment already).

**Thank you for the comment. In section Methodology following clarification has been added (Pg5Ln31): 'The interpretation of the layers shown in Fig. 6 was based on the HYSPLIT backward trajectory simulations performed using the reanalysis meteorological data (Fig. 3), the inspection of the MODIS fire data (Fig. 4) and the literature review. The biomass burning aerosol was classified as follow: semi fresh (up to 2 days of transport), moderately fresh (2-3 days of transport), moderately aged (3-4 days of transport), aged (5-6 days of transport).'**

Could you specify the time window you used in order to characterize the BBA layers as fresh, moderately fresh, etc. I presume that this information is incorporated from the model simulations, therefore I insist that this description should be part of the methodology.

**Yes you, are right. It has been clarified as above.**

§5: I did not find connection with other studies on the topic. You could create a summarizing table with your findings and the findings from other studies. I think this point is important.

**We reckon that it is a valid point, however there is several publications that presents nice comparison tables for values ranges reported in literature (e.g. Baars et al., 2016, Papagiannopoulos et al., 2018, Nicolae et al., 2018). A very recent, excellent table is presented by Nicolae, et al.: A neural network aerosol typing algorithm based on lidar data, Atmos. Chem. Phys., 18, 14511-14537, 2018.**
**This is why we did not see the necessity to repeat such table in our paper (especially as both papers are in the same issue). In revised version we reference to this table. Moreover, we checked obtained in our paper results against the values reported by Nicolae et al. (2018).**
**In our case, lidar ratio for biomass burning particles only in a very few values goes beyond the range listed in Table of Nicolae et al. For pollution, we observed slightly lower lidar ratio values than reported in this Table. Our particle depolarization ratios (below 5%) is within the range reported for biomass burning aerosol and pollution in this Table. Also our extinction-related Angstrom exponent for both aerosol types is similar and only for a few cases slightly higher than reported in this Table.**

Pg8Ln30: 'The sub-layers…α profiles'. Will you explain how this selection is made?

**Thank you, this is now elaborated in Methodology as answered above (In Pg8Ln30 this sentence is removed).**

Pg9Ln8-10: The assumption of dust particles over Warsaw for this specific case is farfetched when taking into account the simulated trajectories. The depolarization ratio value of 8% is rather too low. Does this come from a 'sub-layer' (Fig. 5a)? What do the other intensive properties show?

**Thank you, we revised this sentence as following: 'Above the height of 4 km in Fig. 5 (a) weak layer with depolarization ratio of $\delta_{par532}$ ~8% (the West coast of Africa pointed by the trajectories at 4 km; not shown for clarity of the figure). However, this layer is no further analyzed due to high noise in the profiles.'.**

Pg9Ln15: What does this anti-correlation mean? Is it due to dust contamination? Checking Wiegner et al. (2011) there is no such thing observed.

**We would like to address this issue. In the paper Wiegner et al.: The May/June 2008 Saharan dust event over Munich: Intensive aerosol parameters from lidar measurements, J. Geophys. Res., 116, D23213, 2011; there is no backscattering profiles shown. Therefore we cannot agree with Referee that anticorrelation was not observed by Wiegner et al. (2011).**
**Unless, there is another paper Wiegner et al. (2011)?**
**Moreover, in the text we removed the sentence of dust contamination (Pg9Ln16-18) and added more explanation and rephrased Pg13Ln1-4.**
**Interpretation of anticorrelation of δ and β was reported in our study (mixed biomass burning and pollution), as well as Chazette et al. (2017) (polluted continental, biomass burning) and Haarig et al. (2017) (marine aerosol) can be attributed to particle size (sphericity) increase with water uptake.**

Pg9Ln25-31: Could you expand on the technique for the aerosol typing? Is this technique based only on RH and δ? Are these two parameters sufficient? However, In lines 27-28 you refer to intensive properties, do you use them for the typing? If yes, do you use lookup tables or other information? Furthermore, the different colours refer to different layer aging, is this information solely based on intensive properties or do you use Hysplit info? I am a bit confused. Please clarify.

**Sorry for the confusion. The current paper was never meant to be used in a sense of aerosol typing nor lookup tables. Instead we performed aerosol layer interpretation. The approach is**

now better described in the Methodology section (full chain of steps, which are far beyond RH and delta analysis). The different colors denote layers, which differ with particular aerosol property and not precisely with the age. However, for different ages we obtained different properties.

Pg10Ln23-25: I would expect that pollen is depolarizing, whereas in your measurements I do not observe depolarizing layers. Can you comment? Moreover, the event presented by Sicard et al. (2016) is during the pollination period (i.e., Spring). Could pollen be observed in August? Can you support this statement?

**Thank you for this questions. First of all the peak for the (widely understood) pollen measurements (pollen plant and fungi spores) in Warsaw, it is observed during daytime (4-18 UTC), as it is strictly confined to development of daytime boundary layer. The pollination periods depend on polinating species. For some of them it can start in early spring (e.g. birch) and last till late autumn (e.g. nettle). Of course not all pollen particles are aspherical (e.g. utrica), but fungi spores are elongated and can produce depolarization signature. Thus, observations of pollen in August is not unusual over Warsaw. Depolarization values between 8-12 % at 532 are characteristic for cladosporium dominant in summer. Note that the lack of strong depolarization due to pollen at nighttime is expected.**
**Currently we have a publication on this topic in review; we will add the reference in the revised version. However, if the reviewer finds it important, we can elaborate more on this in the text of the current paper.**

Pg10Ln27: What does this ratio show? Please, describe. How is this ratio connected to pollen?

**Thank you we revise the sentence as follows: 'the ratio of $\delta_{355}/\delta_{532}$ ($CR_\delta$) was calculated to emphasise spectral dependence of depolarization between the layers (the lower $CR_\delta$ the more pollen contamination)'.**
**For pollen particles we expect small values of the $CR_\delta$ (values around 0.35, this study) as the depolarization at 355 nm in the case of very large size (>10μm) aspherical pollen/fungi particles is expected to be low. On the other hand, the $CR_\delta$ above 1 will indicate likely existence of small size depolarizing particles (eg. Haarig et al. (2018), this issue).**

Pg11Ln7: Give literature values for fresh BBA and link it with your findings.

**Thank you, references are added (Nicolae et al. (2013), Baars et al.: Aerosol profiling with lidar in the Amazon Basin during the wet and dry season, J. Geophys. Res., 117, D21201, 2012.).**

Pg11Ln17: Give literature values for aged BBA and link it with your findings.

**Thank you, references are added (Ortiz-Amezcua et al. (2016), Ritter et al. (2018)).**

Pg11Ln17: Give more information on the CRLR threshold.

**Following clarification has been added into sentence (Pg11Ln17): 'The threshold of $CR_{LR}>1$, which is indicative for aged BBA, so as referring to the reported in the literature relation of values of lidar ratio being higher at 532 nm than at 355 nm (e.g. Nicolae et al., 2013, Ortiz et al., 2016), was obtained only in five sub-layers.'.**

Pg12Ln23: In Fig. 7c, where is the aging effect in the scatter plot?

**Please note, that in Pg12Ln23 we cite the results of Samaras et al. (2015), not our results. The following sentence has been added to clarify: 'In fig 7 (c) the ageing effect of BBA is marked by the layer color (semi fresh in pink and red, moderately fresh in blue and aged in orange). This**

ageing relation to the relative humidity is confirmed by the plots in Appendix B Fig. B6 (less humid in pink and red, moderately humid in blue, and humid in orange).

Pg12Ln30-31: Please expand.

**The explicit definition is now given in section Methodology (Pg5Ln27):**
**'… colour ratio of lidar ratios ($CR_{LR}$ = $LR_{532}/LR_{355}$ = exp(-ln(355/532)*($ÅB_{355/532}$- $ÅE_{355/532}$))), colour ratio of depolarization ratios ($CR_\delta$=$\delta_{355}/\delta_{532}$)…'. In Pg12Ln30-31 the sentence is removed. Moreover we rephrased Pg12Ln24-27 as follows: 'Taking into account the definition of $CR_{LR}$ it is visible that the expressions CRLR>1 and $ÅB_{355/532}$- $ÅE_{355/532}$ are the same indicators of aged biomass burning aerosol.'.**

Pg13Ln1-2: How can you deduce that the particle size is increasing from the β-RH plot? Do you imply that the RH is linearly correlated with particle size? Please expand. Also, which is the reference?

**Sorry for not being clear. We do not imply that RH is linearly correlated with particle size, however there is tendency seen in the β and RH scatter plots (Fig. B5) and also in Fig. B6 in the ÅE and RH scatter plots referring to the particle size. The reference it to Haarig et al. (2017) is added.**

Pg13Ln25: Try to revisit the Conclusions as this section is very confusing.

**Conclusions have been revised to point out advantages and limitations of this study. We clearly state that for the identified aerosol layers the microphysical parameters are actually retrieved, which is the topic of the 'part II' paper.**

**Minor Specific Comments**

Pg1Ln1: I counted 8 distinct layers when I checked Table 1. L1a and L1b are the same as well as the pair L2a and L4b.

**The information in Table 1 was indeed somewhat misleading. In fact different colors denote different layers, whereby these differences can be due either due to the optical properties and/or the particle origin and transport. Therefore we elaborated Table 1 as below. Moreover, in the abstract we decided not to give explicitly number of layers.**

| Selected layer | $LR_{355}$ [sr] | $LR_{532}$ [sr] | RH [%] | $\delta_{par355}$ [%] | $\delta_{par532}$ [%] | $ÅE_{355/532}$ | $ÅB_{355/532}$ | $ÅB_{532/1064}$ | $CR_{LR}$ |
|---|---|---|---|---|---|---|---|---|---|
| **9/10 August 2015** | | | | | | | | | |
| **L1(a)   H (1.3 – 1.5 km); T (19 – 02 UTC)** **pollution wet** | | | | | | | | | |
| | $55 \pm 6$ | $43 \pm 4$ | $76 \pm 2$ | $1.5 \pm 0.1$ | $3.6 \pm 0.3$ | $1.48 \pm 0.10$ | $0.86 \pm 0.08$ | $0.40 \pm 0.04$ | $0.78 \pm 0.04$ |
| **L2(a)   H (1.6 – 2.6 km); T (19 – 01 UTC)** **BBA moderately fresh (2-3 days) more depolarizing** | | | | | | | | | |
| | $76 \pm 7$ | $62 \pm 4$ | $68 \pm 4$ | $1.6 \pm 0.2$ | $3.5 \pm 0.3$ | $1.35 \pm 0.18$ | $0.85 \pm 0.14$ | $0.46 \pm 0.05$ | $0.82 \pm 0.09$ |
| **L3(a)   H (1.8 – 2.6 km); T(23 – 01 UTC)** **BBA semi fresh (up to 2 days) lower lidar ratio, more depolarizing** | | | | | | | | | |
| | $81 \pm 6$ | $60 \pm 6$ | $53 \pm 4$ | $2.5 \pm 0.4$ | $5.0 \pm 0.3$ | $1.34 \pm 0.33$ | $0.60 \pm 0.13$ | $0.52 \pm 0.07$ | $0.75 \pm 0.10$ |
| **L4(a)   H (1.7; 2.5 km); T (02 UTC)** **mixed** | | | | | | | | | |
| | $96 \pm 1$ | $85 \pm 5$ | $62 \pm 4$ | $2.0 \pm 0.6$ | $4.1 \pm 0.8$ | $1.15 \pm 0.09$ | $0.86 \pm 0.04$ | $0.35 \pm 0.10$ | $0.89 \pm 0.05$ |
| **L5(a)   H (2 - 3.5 km); T (19 – 02 UTC)** **BBA moderately aged (3-4 days) more wet** | | | | | | | | | |
| | $71 \pm 10$ | $57 \pm 8$ | $85 \pm 6$ | $1.0 \pm 0.2$ | $2.3 \pm 0.3$ | $1.60 \pm 0.22$ | $1.05 \pm 0.28$ | $0.65 \pm 0.14$ | $0.81 \pm 0.10$ |
| **L6(a)   H (2.5 – 3.5 km); T (19 – 02 UTC)** **BBA aged (5-6 days) / early cloud** | | | | | | | | | |

| | | | | | | | | |
|---|---|---|---|---|---|---|---|---|
| 60 ± 12 | 67 ± 7 | 82 ± 8 | 1.1 ± 0.4 | 2.5 ± 0.7 | 0.60 ± 0.32 | 0.89 ± 0.34 | 0.68 ± 0.12 | 1.13 ± 0.14 |
| **10/11 August 2015** | | | | | | | | |
| **L1(b)   H (1 – 1.3 km); T (21 – 02 UTC)** | | | | | | | | |
| **pollution dry** | | | | | | | | |
| 53 ± 5 | 31 ± 4 | 55 ± 1 | 1.6 ± 0.1 | 3.8 ± 0.2 | 2.03 ± 0.21 | 0.69 ± 0.13 | 0.36 ± 0.04 | 0.58 ± 0.06 |
| **L2(b)   H (1 – 1.3 km); T (19 – 21 UTC)** | | | | | | | | |
| **pollution dry + pollen** | | | | | | | | |
| 73 ± 12 | 45 ± 11 | 58 ± 2 | 1.8 ± 0.1 | 4.8 ± 0.3 | 1.82 ± 0.37 | 0.56 ± 0.10 | 0.26 ± 0.05 | 0.60 ± 0.07 |
| **L3(b)   H (1.3 – 1.5 km); T (19 – 00 UTC)** | | | | | | | | |
| **BBA semi fresh (up to 2 days) higher lidar ratio, less depolarizing** | | | | | | | | |
| 114 ± 33 | 78 ± 25 | 48 ± 4 | 1.6 ± 0.3 | 3.8 ± 0.5 | 1.50 ± 0.19 | 0.58 ± 0.24 | 0.44 ± 0.03 | 0.69 ± 0.07 |
| **L4(b)   H (1.5 – 1.7 km); T (21 – 00 UTC)** | | | | | | | | |
| **BBA moderately fresh (2-3 days) less depolarizing** | | | | | | | | |
| 80 ± 7 | 66 ± 9 | 64 ± 2 | 1.1 ± 0.1 | 2.8 ± 0.1 | 1.41 ± 0.12 | 0.90 ± 0.18 | 0.39 ± 0.05 | 0.82 ± 0.09 |
| **L5(b)   H (1.5 – 2.0 km); T (19 – 02 UTC)** | | | | | | | | |
| **BBA moderately aged (3-4 days) more dry** | | | | | | | | |
| 69 ± 9 | 61 ± 7 | 76 ± 4 | 1.0 ± 0.1 | 2.3 ± 0.3 | 1.30 ± 0.15 | 0.99 ± 0.10 | 0.47 ± 0.06 | 0.88 ± 0.05 |

Pg1Ln15: You mention fresh, moderately fresh, and moderately aged. In Table 1, you also refer to aged BBA. Why not include it in this sentence?

**Sorry for being not clear. In fact we talked about the aged BBA in lines 15-18, but the composition of the text was, indeed, confusing. It is revised now.**

Pg1Ln15-17: Why do you refer to this specific layer in the abstract?

**It is an interesting aged BBA layer you just asked for before. We revised the abstract and now we are referring mainly to the most different layers (polluted, semi fresh bb and aged bb).**

Pg4Ln11: Do the data shown in this study follow the QA and QC procedures of EARLINET?

**Yes. We added respective sentences.**

Pg8Ln14: Do you have any literature reference to support this claim?

**Well, we have several press release reports regarding local fires, which were not seen on the MODIS data. We did not make links to press news as it is in Polish language only, e.g. https://www.lublin112.pl/kolejne-pozary-pol-lasow-radawcu-gaszenia-wezwano-samolot-zdjecia/#close_info_content**

Pg8Ln31: Give more information for the PolandAOD database.

**The Polish Aerosol Research Network (PolandAOD) conducts the measurements of aerosol properties and radiation budget in several cites in Poland (supplementary materials of Markowicz et al. 2016: Study of aerosol optical properties during long-range transport of biomass burning from Canada to Central Europe in July 2013, Journal of Aerosol Science, 101, 156-173, 2016). We added this reference. More information on the PolandAOD Data Base is available via the website http://polandaod.pl/. Corresponding information is added in the text.**

Pg9Ln19-24: Aren't the lidar signals too noisy to make any deductions? I would omit this discussion.

**We would like to keep this description, as it is not unusual to see at Warsaw such intrusions of arctic air. The long term study analysis even within boundary layer indicates that at least 5% of the long range intrusion had arctic marine origin (Wang et al., 2018, this issue). Therefore it is even more likely to see an arctic air at higher altitudes.**

Pg10Ln24: You mention high values of depolarization ratio, whilst 4.8% is definitely a small value. Please clarify.

**Sorry for being unclear. Of course 4.8% is relatively low depolarization. We rephrased Pg10Ln24-25 to: 'For polluted layer in Warsaw we expect values of $\delta_{par532}$ below 1% (Wang et al., 2018), thus the relatively high value of 4.8 % (which is decreasing during the night, along with a slight increase of $ÅE_{355/532}$) can reflect contamination of this polluted layer with a pollen particles waning after the sunset.'**

Pg11Ln5-7: What did the model simulations show?

**We assume Referee means the HYSPLIT model backward trajectory simulations. If yes, we add in the text following sentence: 'The HYSPLIT backward trajectory simulation indicate, that at the altitude of ~2 km on 9/10 August, more likely semi fresh biomass burning aerosol (up to 2 days old) was advected from over German and the Czech Republic.'**

Pg11L11: I think the statement 'small depolarizing particles' is untrue. Please rephrase.

**Sorry for being unclear. We rephrase the sentence Pg11Ln8-12 as follows: 'The $CR_\delta$ value for this layer was the highest (~0.49), in comparison with all other layers.'.**

Pg13Ln20: Can you quantify 'partly consistent'?

**Thank you for the comment. We rephrased the fragment Pg13Ln20-23 as follows: 'Results of MFR-7 Shadowband Radiometer (PolandAOD-NET, http://polandaod.pl/) are consistent in terms of higher AOD derived at shorter wavelength with the data of C318 Sun Photometer (AERONET, https://aeronet.gsfc.nasa.gov/) site in Belsk (~50 km to the south of Warsaw). However, the data obtained at both sites are not directly comparable in this period due to complicated meteorological conditions of the quasi-stationary weather front.'.**

Pg13Ln31-34: Better remove this sentence.

**Thank you. This is removed.**

Pg14Ln15-16: What does this finding show?

**The following sentence is added: 'This being in relation to water uptake by the particles.'.**

Pg28: Can you overlay the position of the EARLINET sites of Poland? If the empty dots correspond to the sites, make sure to change the size, or colour, or icon.

**This paper concerns the data sets obtained at the only one Raman lidar EARLINET site in Poland. We indicated the Warsaw EARLINET site with an arrow.**

Pg29: Consider removing this figure or keep the figures related to the reanalysis data.

**The comment was followed as suggested; we left only the trajectories obtained using reanalysis data. Thank you.**

Pg31: A legend is required for Fig. 5. Also, why don't you report the profiles of lidar ratio and Ångström exponent? The profiles will give an insight into the aerosol layers. Furthermore, the figures are poorly rendered. Make sure to improve the quality of the Figures 5, 6, and 7.

**Thank you for the comment. The legend has been added to Fig 5. We will improve the quality of the Figures 5, 6, 7. We agree that it is possible to have the selection of each layer based on the**

**intensive parameters but a lot of averaging/smoothing have to be done to do properly this selection. This mean that the extensive parameters will be also averaged a lot when one interprets the data obtained in such way. Nevertheless the microphysical inversion depends on the quality of the optical data. Thus the selection of many thin layers is in advantage for the microphysical inversion. This is why we prefer to stick with the proposed approach.**

Pg32: Consider a legend describing the colours or incorporate this information in the colour-bar. In Fig. 7a, the layers 7 and 8 are not discernible. In Fig. 7b, merge layers 6-7. You could have the same colour for layers that correspond to the same aerosol type. E.g. L1 in both layers corresponds to pollution.

**Thank you for the comment. Several legends have been added to Figures 5, 6, and 7.**
**We prefer not to combine the pollution layers together, as they are distinctly different due to the relative humidity.**

**Technical Comments**

Thank you for all the technical comments. We have considered all of them in revised version, including the revision of sentences.

Pg1Ln8: Add 'the' after 'during'.        **Authors: Done.**
Pg1Ln8: The 'th' is not needed in the date.     **Authors: Done.**
Pg1Ln9: Add 'the' before 'so-called'.       **Authors: Done.**
Pg1Ln10: Replace 'The' with 'A'.        **Authors: Done.**
Pg1Ln11: Write 'optical properties of 116 layers' instead of 'properties within 116 sublayers in the profiles'.           **Authors: Done.**
Pg1Ln13: Add 'the' before 'aerosol/mixture'.     **Authors: Done.**
Pg1Ln16: Replace 'characteristic for' with 'characteristic for'.  **Authors: Done.**
Pg1Ln16: Delete 'scattered'.          **Authors: Done.**
Pg1Ln18: Delete '4.8'.            **Authors: Done.**
Pg1Ln18: Replace 'were' with 'was'.       **Authors: Done.**
Pg1Ln24: Replace 'studying of' with 'studying the'.    **Authors: Done.**
Pg1Ln26: Add 'the' before 'microphysics'.      **Authors: Done.**
Pg1Ln26: Delete 'forming in the presence of aerosols'.   **Authors: Done.**
Pg1Ln27: Replace 'particles' with 'particle'.     **Authors: Done.**
Pg1Ln28: Add 'the' before 'aerosol'.       **Authors: Done.**
Pg1Ln28: Delete 'suspension'.         **Authors: Done.**
Pg2Ln2: Replace 'what' with 'which'.       **Authors: Done.**
Pg2Ln8: Replace 'consist' with 'consists'.      **Authors: Done.**
Pg2Ln9: Replace 'is the subject of' with 'is subject to'.   **Authors: Done.**
Pg2Ln9: Replace 'processes' with 'process'.     **Authors: Done.**
Pg2Ln9: Replace 'lead' with 'leads'.       **Authors: Done.**
Pg2Ln10: Replace 'evolutions' with 'change'.     **Authors: Done.**
Pg2Ln11: Delete ',' after 'fact'.        **Authors: Done.**
Pg2Ln13: Add 'the' before 'development'.     **Authors: Done.**
Pg2Ln13: I do not understand 'yield in the'.     **Authors: Rephrased.**
Pg2Ln13: Replace 'allow' with 'allows'.      **Authors: Done.**
Pg2Ln14: Add 'the' before 'characterization'.    **Authors: Done.**
Pg2Ln14-15: Please rephrase 'a signal depolarization by unspherical particles'. **Authors: Rephrased.**
Pg2Ln15: Replace 'unspherical' with 'aspherical'.    **Authors: Done.**
Pg2Ln17: Delete 'the' after 'during'.      **Authors: Done.**
Pg2Ln19: Replace 'the' with 'a'.        **Authors: Done.**
Pg2Ln21: Delete 'the' after 'by'.        **Authors: Done.**
Pg2Ln21: Please rephrase 'to increase the research quality and reliability'.  **Authors: Rephrased.**
Pg2Ln23-24: Please rephrase 'The automatic aerosol typing algorithms operate with…'.

**Authors: Rephrased.**

Pg2Ln24: The abbreviation is missing. Make sure to include abbreviations throughout the manuscript with specific references and website links when needed.     **Authors: Done.**

Pg2Ln27: Add 'a' before 'maximum'.     **Authors: Done.**

Pg2Ln28: Replace 'This' with 'these'.     **Authors: Done.**

Pg2Ln29: Delete 'a very'.     **Authors: Done.**

Pg2Ln29: Please rephrase 'to use the lidar data by the specialist'.     **Authors: Rephrased.**

Pg2Ln31: Delete 'a' before 'high'.     **Authors: Done.**

Pg2Ln33-34: Please rephrase the last sentence.     **Authors: Rephrased.**

Pg3Ln1-2: Please rephrase the first sentence.     **Authors: Rephrased.**

Pg3Ln2: Delete 'A'.     **Authors: Done.**

Pg3Ln3: Delete 'a' before 'long'.     **Authors: Done.**

Pg3Ln5-6: What do you mean 'found in the systematic observations'?     **Authors: Rephrased.**

Pg3Ln6-8: Also, what does this sentence mean? The message is not clear.     **Authors: Rephrased.**

Pg3Ln9: Delete 'a' before 'favourable'.     **Authors: Done.**

Pg3Ln14: Replace 'according with' with 'according to', however, I do not think that fits here. **Authors: Rephrased.**

Pg3Ln16: Replace 'were' with 'was'.     **Authors: Done.**

Pg3Ln16: Replace 'based on' with 'using'.     **Authors: Done.**

Pg3Ln18: 'which coincide with the analysed in the paper'. What do you mean? **Authors: Rephrased.**

Pg3Ln19: Replace 'aerosol' with 'aerosols'.     **Authors: Done.**

Pg3Ln19: Write 'as biomass burning smoke from Ukrainian wildfires'. **Authors: Done.**

Pg3Ln20-21: What do you mean by 'deepening study'?     **Authors: Rephrased.**

Pg3Ln21: Please rephrase 'to reflect atmospheric variability and to catch all individual sublayers'. **Authors: Rephrased.**

Pg3Ln22: What do you mean by 'can be likely kinds of mixtures'?     **Authors: Rephrased.**

Pg3Ln23: Delete 'the' before 'separation'.     **Authors: Done.**

Pg3Ln28-29: Please rephrase 'as the one… in each sub-layer'. **Authors: Rephrased.**

Pg3Ln29: Delete 'mentioned'.     **Authors: Done.**

Pg3Ln30: Please rephrase 'performed as a standard'.     **Authors: Rephrased.**

Pg3Ln31: Add 'the' before 'study'.     **Authors: Done.**

Pg3Ln32-34: Please rephrase the last sentence. Also, a reference is missing.     **Authors: Rephrased.**

Pg4Ln1: Replace 'contain' with 'contains'.     **Authors: Done.**

Pg4Ln1-6: The abbreviation 'Sect.' should be used when it appears in running text. **Authors: Done.**

Pg4Ln5: Replace 'is' with 'are'.     **Authors: Done.**

Pg4Ln5: Please rephrase 'outlooks potential use of the obtained results'.     **Authors: Rephrased.**

Pg4Ln10: Replace 'as' with 'following'.     **Authors: Done.**

Pg4Ln15: Replace 'consist of the' with 'consists of a'.     **Authors: Done.**

Pg4Ln20: Add 'an' after 'where'.     **Authors: Done.**

Pg4Ln23-24: Add 'the' before 'particle' and 'water'.     **Authors: Done.**

Pg4Ln26: Replace 'Baars et al., 2016' with 'Baars et al. (2016)'. Make sure when you are citing a paper to comply with the guidelines of ACP. Several mistakes of this kind were found throughout and it is not possible to list them all out.     **Authors: Done.**

Pg4Ln28: For the dates, consider using dashes and not slashes. E.g. 9-10 August.     **Authors: Done.**

Pg5Ln2: Replace 'meteorology' with 'meteorological'.     **Authors: Done.**

Pg5Ln4: Add 'the' before 'retrieval'.     **Authors: Done.**

Pg5Ln9: Add 'the' before 'free'.     **Authors: Done.**

Pg5Ln14: You refer to 'the latter', which is the former?     **Authors: Rephrased.**

Pg5Ln19: Delete 'of' after 'Comparing'.     **Authors: Done.**

Pg5Ln19: Delete 'the' after 'showed'.     **Authors: Done.**

Pg5Ln20: Add 'the' before 'calculation'.     **Authors: Done.**

Pg5Ln20: Delete 'the' before 'mean'.     **Authors: Done.**

Pg5Ln23: Add 'for' before 'further'.     **Authors: Done.**

Pg5Ln24-25: Delete 'of the intensive optical properties'.     **Authors: Done.**

Pg5Ln25: Replace 'sets' with 'set'.     **Authors: Done.**

Pg5Ln32: What do you mean by 'aerosol layers occurrence'?   **Authors: Rephrased.**
Pg5Ln33: Add 'the' before 'residual'.   **Authors: Done.**
Pg6Ln8: Replace 'On the' with 'At the'.   **Authors: Done.**
Pg6Ln8: Add 'a' before 'persistent'.   **Authors: Done.**
Pg6Ln10: Write 'Inflow of warm air from Western Africa'.   **Authors: Done.**
Pg6Ln10: What do you mean by 'was dominating in the troposphere'?   **Authors: Rephrased.**
Pg6Ln14: Delete 'the' after 'within'.   **Authors: Done.**
Pg6Ln14: I cannot understand 'insensitively'?   **Authors: Rephrased.**
Pg6Ln15: Delete 'territory'.   **Authors: Done.**
Pg6Ln15: Replace 'In the three days period' with 'In the next three days'.   **Authors: Done.**
Pg6Ln15: Replace 'at the direct' with 'located in direct'.   **Authors: Done.**
Pg6Ln16 and Pg6Ln18: Delete 'panel'.   **Authors: Done.**
Pg6Ln18: Replace 'depict' with 'depicts'.   **Authors: Done.**
Pg6Ln18: Add 'the' before 'distinct'.   **Authors: Done.**
Pg6Ln19: Replace 'splitted' with 'split'.   **Authors: Done.**
Pg6Ln22: Delete 'occurrence'.   **Authors: Done.**
Pg6Ln22: What does the sentence 'air settlement in the conditions of high pressure' mean?
**Authors: Rephrased.**
Pg6Ln23: Delete 'of aerosol structures'.   **Authors: Done.**
Pg6Ln25: Replace 'directions of the aerosols inflow' with 'aerosol origin'.   **Authors: Done.**
Pg6Ln31: What is a 'high interval'?   **Authors: Rephrased.**
Pg7Ln1: What does 'reflect' mean?   **Authors: Rephrased.**
Pg7Ln2: Replace 'have' with 'has'.   **Authors: Done.**
Pg7Ln3: Add 'the' before 'general'.   **Authors: Done.**
Pg7Ln3: Please rephrase 'ran starting on the territory'?   **Authors: Rephrased.**
Pg7Ln5: Replace 'At the altitudes' with 'For the altitudes'.   **Authors: Done.**
Pg7Ln6: Add 'the' before 'trajectories'.   **Authors: Done.**
Pg7Ln6-7: Please rephrase 'from the mentioned before direction of the lower trajectories to the direction from western Europe'.   **Authors: Rephrased.**
Pg7Ln7: Add 'the' before 'Czech'.   **Authors: Done.**
Pg7Ln7-8: Please rephrase 'Over the time: : : over Spain'.   **Authors: Rephrased.**
Pg7Ln9-11: Please rephrase 'At the altitude: : : Sahara Desert'.   **Authors: Rephrased.**
Pg7Ln11: Replace 'originate' with 'originates'.   **Authors: Done.**
Pg7Ln14: Add 'the' before 'consistency'.   **Authors: Done.**
Pg7Ln17: Write 'the two simulations is the origin of the aerosols over Warsaw'.   **Authors: Done.**
Pg7Ln21: Replace 'analogous' with 'same'.   **Authors: Done.**
Pg7Ln23: Delete 'at the analogous altitudes'.   **Authors: Done.**
Pg7Ln28: Delete 'with'.   **Authors: Done.**
Pg7Ln28: Add 'the' before 'GDAS'.   **Authors: Done.**
Pg8Ln2: The period should be 10-11 August and not 9-11.   **Authors: Done.**
Pg8Ln3: Replace 'panel (a)' with 'Fig. 4a'.   **Authors: Done.**
Pg8Ln4-5 and Ln14: The same change applies to these sentences.   **Authors: Done.**
Pg8Ln5: What do you mean by 'The trajectory altitude is rising…'?   **Authors: Rephrased.**
Pg8Ln6: Replace 'as in' with 'to'.   **Authors: Done.**
Pg8Ln7: Delete 'the' before 'satellite'.   **Authors: Done.**
Pg8Ln9-10: Please rephrase.   **Authors: Rephrased.**
Pg8Ln12: Replace 'point that biomass' with 'point out that the biomass'.   **Authors: Done.**
Pg8Ln15: Replace 'most likely originates' with 'might have come'.   **Authors: Done.**
Pg8Ln16: Delete 'the' before 'copper'.   **Authors: Done.**
Pg8Ln16: Add 'the' before 'possible'.   **Authors: Done.**
Pg8Ln16: Delete 'the' before 'pollution'.   **Authors: Done.**
Pg8Ln16: Add 'the' before 'altitude'.   **Authors: Done.**
Pg8Ln17: Replace 'origin' with 'originate'.   **Authors: Done.**
Pg8Ln19: Add 'the' before 'Iberian'.   **Authors: Done.**
Pg8Ln20: Replace 'origin' with 'originate'.   **Authors: Done.**

Pg8Ln21: Replace 'Hungry' with 'Hungary'. **Authors: Done.**
Pg8Ln22: Delete 'the' before 'moderately'. **Authors: Done.**
Pg8Ln22-23: Please rephrase **Authors: Rephrased.**
Pg8Ln26: Replace 'origin' with 'originate' everywhere. **Authors: Done.**
Pg8Ln29: What do you mean by 'representative for the layers'? **Authors: Rephrased.**
Pg8Ln29: Delete 'the' before 'mean'. **Authors: Done.**
Pg9Ln1: Replace 'in further study' with 'for further study'. **Authors: Done.**
Pg9Ln1: Delete 'of the intensive properties'. **Authors: Done.**
Pg9Ln11: Add 'the' after 'course of'. **Authors: Done.**
Pg9Ln11: Add 'A' before 'similar'. **Authors: Done.**
Pg9Ln14: Add 'the' before 'data'. **Authors: Done.**
Pg9Ln17: Add 'the' before 'slight'. **Authors: Done.**
Pg9Ln17: Replace 'from over Sahara' with 'from the Sahara Desert'. **Authors: Done.**
Pg9Ln25: Replace 'performed' with 'made'. **Authors: Done.**
Pg9Ln31: Please explain 'have some features'. **Authors: Rephrased.**
Pg10Ln8: Replace 'splitted' with 'split'. **Authors: Done.**
Pg10Ln14: Replace 'in the first' with 'for the first'. **Authors: Done.**
Pg10Ln14: Replace 'cooper' with 'copper'. **Authors: Done.**
Pg10Ln23: Replace 'domination' with 'dominance'. **Authors: Done.**
Pg10Ln23-25: Consider to rephrase this sentence. **Authors: Rephrased.**
Pg10Ln26: Please rephrase 'To easy compare'. **Authors: Rephrased.**
Pg10Ln28: Replace 'has' with 'have'. **Authors: Done.**
Pg10Ln32: Replace 'have' with 'has'. **Authors: Done.**
Pg10Ln34: What do you mean by 'stronger contamination'? **Authors: Rephrased.**
Pg11Ln1: Delete 'i' from '0.12i' and '0.30i'. **Authors: Done.**
Pg11Ln5: Replace 'of' with ','. **Authors: Done.**
Pg11Ln7: Replace 'weakly' with 'less'. **Authors: Done.**
Pg11Ln8-12: Please rephrase, the message is not clear. **Authors: Rephrased.**
Pg11Ln13-14: Please rephrase. **Authors: Rephrased.**
Pg11Ln18: Replace 'this' with 'these'. **Authors: Done.**
Pg11Ln18: 'This… sr).' What do you mean? **Authors: Rephrased.**
Pg11Ln18: Delete 'of' after 'was'. **Authors: Done.**
Pg11Ln21: Delete 'of' before 'the BBA'. **Authors: Done.**
Pg11Ln27: Delete 'one' before 'case'. **Authors: Done.**
Pg12Ln4: Delete ',' after 'shows'. **Authors: Done.**
Pg12Ln4: Delete 'rather'. **Authors: Done.**
Pg12Ln7: Add 'the' before 'linear'. **Authors: Done.**
Pg12Ln7: Delete 'the' before 'details'. **Authors: Done.**
Pg12Ln9: I do not get the meaning of 'alternatively'. **Authors: Rephrased.**
Pg12Ln9: Write 'the dependence is not linear'. **Authors: Done.**
Pg12Ln9-12: Please rephrase. **Authors: Rephrased.**
Pg12Ln13: Please rephrase. **Authors: Rephrased.**
Pg12Ln14: Replace 'mixture' with 'mixtures'. **Authors: Done.**
Pg12Ln16: Replace 'another' with 'other'. **Authors: Done.**
Pg12Ln16: Please rephrase 'another stage… atmosphere'. **Authors: Rephrased.**
Pg12Ln19: Replace 'splitted' with 'split'. **Authors: Done.**
Pg12Ln21: Replace 'this' with 'these'. **Authors: Done.**
Pg12Ln23: Replace 'hygroscopicity' with 'the hygroscopic growth'. **Authors: Done.**
Pg12Ln24-25: Please rephrase the two sentences. It is impossible to understand.**Authors: Rephrased.**
Pg12Ln25-27: Please explain. **Authors: Rephrased.**
Pg12Ln27: Delete 'the' before 'model'. **Authors: Done.**
Pg12Ln29: Replace 'for' with 'to'. **Authors: Done.**
Pg12Ln29: Add 'the' before 'imaginary'. **Authors: Done.**
Pg12Ln32: Replace 'shows' with 'show'. **Authors: Done.**
Pg12Ln33: Delete 'the' before 'negative'. **Authors: Done.**

Pg13Ln1-2: Please rephrase. It is impossible to understand.     **Authors: Sorry, rephrased.**
Pg13Ln2-4: Please rephrase. It is impossible to understand.     **Authors: Sorry, rephrased.**
Pg13Ln6: What do you mean by 'assessed'?     **Authors: Rephrased.**
Pg13Ln17: Add 'the' before 'troposphere'.     **Authors: Done.**
Pg13Ln17: Add 'the' before 'aerosol'.     **Authors: Done.**
Pg13Ln17: Delete 'the' before 'observed on that days'.     **Authors: Done.**
Pg13Ln17: Replace 'were' with 'was'.     **Authors: Done.**
Pg13Ln18: Pleas rephrase 'present a consistent course'.     **Authors: Rephrased.**
Pg13Ln19: Add 'the' before 'radiometer'.     **Authors: Done.**
Pg13Ln20: Replace 'this' with 'the'.     **Authors: Done.**
Pg13Ln22: I cannot understand 'what'.     **Authors: Rephrased.**
Pg13Ln26: Delete 'an' before 'aerosol'.     **Authors: Done.**
Pg13Ln26: Delete 'the' before 'biomass'.     **Authors: Done.**
Pg13Ln28: Replace 'show' with 'showed'.     **Authors: Done.**
Pg13Ln29: Write 'that 2-3 days old air from Germany…'.     **Authors: Done.**
Pg13Ln30: Delete 'about'.     **Authors: Done.**
Pg13Ln31: Replace 'is' with 'in'.     **Authors: Done.**
Pg14Ln1: 'specifying: : : sub-layers'. Please rephrase.     **Authors: Rephrased.**
Pg14Ln3: Add 'A' before 'total'.     **Authors: Done.**
Pg14Ln4: Replace 'o' with 'of'.     **Authors: Done.**
Pg14Ln4-5: 'general: : : approach'. Please rephrase.     **Authors: Rephrased.**
Pg14Ln8-9: 'slight: : : possible'. Please rephrase.     **Authors: Rephrased.**
Pg14Ln9: Add 'the' before 'upper'.     **Authors: Done.**
Pg14Ln9: Replace 'were' with 'was'.     **Authors: Done.**
Pg14Ln10-11: 'In one layer: : : information'. Please rephrase.     **Authors: Rephrased.**
Pg14Ln11: Replace 'do' with 'does'.     **Authors: Done.**
Pg14Ln11: Replace 'an' with 'a'.     **Authors: Done.**
Pg14Ln11-13: Please rephrase 'occurs: : : formation)'.     **Authors: Rephrased.**
Pg14Ln17: Replace 'of some kinds of mixtures' with 'mixed'.     **Authors: Done.**
Pg14Ln23-24: Please rephrase.     **Authors: Rephrased.**
Pg27: Insert the following 'black dots depict the', 'calculated with the', 'purple bars depict the', 'estimated from'.     **Authors: Done.**
Pg27: Replace 'selected to the evaluation' with 'selected for evaluation'.     **Authors: Done.**
Pg29: Please improve the caption of Figure 3.     **Authors: Done.**
Pg30: What do you mean by 'the trajectory altitude is rising north-westward'?     **Authors: Rephrased.**
Pg31: Replace 'in further analysis' with 'for further analysis'.     **Authors: Done.**
Pg32: Insert the following 'the same colour depicts' and 'layer 8 depicts'.     **Authors: Done.**
Pg32: Delete 'further' wherever appears in the caption.     **Authors: Done.**
Pg33: Please rephrase the last sentence of the caption.     **Authors: Rephrased.**

---

## Author Comment (AC3) · 15 May 2019

Response to Reviewer Report #2 by Janicka & Stachlewska on 14 May 2019

Dear Referee, Dear Editor,

We are grateful to Referee for comments and suggestions that allowed us to improve this manuscript.

In the following, the answers to Referee's comments and issues raised are reported directly below each related comment. All modifications of the initial version of the manuscript as well as additions are reported in color highlight in the revised version of the manuscript. We believe that we have fulfilled required changes in the final version of the manuscript.

We would like to thank Referee for her/his opinion on the initially submitted manuscript. We have carefully considered all of the comments and suggestions; below point-by-point answers are given (in blue). In general, we would like to stress that crucial points raised by the Referee have been revised, so as following:

- the methodology for interpretation of aerosol layers, as not described well enough

Now: Aerosol source/composition/advection better combined, plus referred to values reported in literature

- the uncertainties of obtained properties, as not addressed sufficiently

Now: Systematic and statistical measurement uncertainties discussed in terms of signal-to-noise ratio (averaging/smoothing), data evaluation methodology, etc.

Therefore, we are grateful to Referee for helping in substantially improving the initial version of the manuscript.

The manuscript introduces a dataset of lidar observations of aerosol backscatter,aerosol extinction, and aerosol depolarization at multiple wavelengths plus relative humidity, for a heat wave event in Warsaw comprising many identified aerosol layers over two nights. It provides aerosol type identification for these layers and gives the mean lidar properties for layers of each type.

In fact, it was never meant to focus this study on the aerosol typing in the sense of an application of an automated aerosol typing routine (so as e.g. Papagiannopoulos et al., 2018, Nicolae et al., 2018, *this issue*). This was the reason why we avoided using the wording *aerosol typing* throughout this manuscript. We performed a classical aerosol layer interpretation, this using both the aerosol optical properties (as reported in literature), which were combined with backward trajectory analyses as well as information provided by the satellite sensors and synoptic charts).

Unfortunately, most aspects of aerosol type analysis are not clear and there are many results with inadequate support or no support whatsoever.

In the revised version an effort was put on better describing the approach undertaken for the interpretation of the identified aerosol layers (and sub-layers), through adding also a methodology work-flow scheme, a better description of the aerosol layers in the Tables and Figures, and most of all – relevant references to place this research in a wider range.

The overall motivation or objective of the manuscript also seems confused.

This is true, We regret not to state this clear enough. The introduction has been modified accordingly with focus on motivation and objectives. This is explicitly addressed in the answered to specific comments of the Referee, below.

The lack of clarity about the objectives and especially the flaws in the analysis make it impossible for me to recommend the manuscript for publication in ACP.

Several aspects have been improved, all accordingly to both Referees suggestions and comments, and therefore we believe the revised version will satisfy Referees.

Before getting into the details, I will say that I agree that the dataset itself (as distinct from the aerosol type analysis) is potentially quite valuable, and I am particularly impressed by the inclusion of relative humidity profile information which adds significant potential for science value when combined with the lidar aerosol measurements.

Thank you for this comment. In fact, adding information on lidar-derived RH in the presented data set is an advantage over the EARLINET data sets, as for RH profiles being not yet reported in the EARLINET-ACTRIS Data Base. All profiles derived in this study are published in this data base. The evaluation scheme was introduced with much detail in Baars et al, 2016. All of the obtained sets of profiles are accomplished for the measurements comprising with the QA of the EARLINET-ACTRIS recommendations. Moreover also the QC tests were performed these data, whereby the Warsaw site has one of the highest scores of according to last evaluation in March 2019 for 97% of all Warsaw profiles stored fulfill the QC). Now, this all is in regard of all profiles reported in our paper but RH profiles, due to the lack of QA&QC procedures for RH in the EARLINET-ACTRIS Data Base (majority of sites are not working yet with RH). The evaluation scheme and results validation for RH are reported by Stachlewska et al 2017, where details of the retrieval, including uncertainty analyses are given.

One option might be to make a streamlined manuscript without the aerosol type analysis, but with a more in-depth uncertainty analysis for the primary lidar measurements (backscatter, extinction, intensive parameters, and RH), plus an explanation of the quality control, and submit this alternate paper as a data paper in a data journal such as Earth System Science Data.

We would like to thank Referee for this recommendation, as publishing the data sets in such a highly values journal as the Earth System Science Data of Copernicus Publications would be precious. However, we feel that it is not feasible for the current study and the discussed data set. First of all, the quality assurance and control of the obtained data set has been done accordingly to the EARLINET-ACTRIS procedures. Secondly, the data has been already published in the EARLINET-ACTRIS Data Base. The EARLINET Community (e.g. Nemuc 2019, Short Comment, *this issue*) show potential interest for using this data set. Even Referee himself seems to have interest to use it – questions on data public availability). Thirdly, we added a new section related to measurement uncertainty analyses, as suggested. This is why we reckon that submitting this research to the current EARLINET Special Issue of the ACP is appropriate and valuable for this issue/community.

I will organize my more detailed comments by manuscript section.

Thank you, below we answered them point-by-point.
* * *
Section "Introduction":

My primary criticism in the introduction is that it does not adequately state the objective and motivation.

The introduction has been revised taking to account following:

The main objective:
Derive unique set of comprehensive profiles that can be used in future for further exploration in e.g. microphysical inversion (as in Veselovskii et al., 2002; Böckmann et al., 2005; Müller et al., 2016); testing of aerosol typing algorithms (e.g. Nicolae et al., 2018; Papagiannopoulos et al., 2018); testing the aerosol separation algorithms (e.g. Mamouri and Ansmann, 2017).

The motivation for this study:
The aerosol properties reported in literature are often limited in terms of: i) existence of wavelength dependent depolarization and/or RH, ii) profiles obtained for a single measurement (1-2h average, at single location, at particular time, no temporal dependence), iii) profiles evaluated for the cases of well-defined aerosol (practically pure aerosol type/source) in well-defined layers (geometrically and optically thick).
But how to understand such reported in literature profiles?
What can be said on their representability in vertical and temporal extend?
Is the significant averaging (over time/height) necessary?
Is it meaningful to divide atmosphere to a high spatial-temporal resolution *sectors* and assess the aerosol properties within them?
Is it feasible to find in these many very thin *sub-layers* a coherence of the aerosol properties?
Are those connected to air-mass transport to unambiguously interpret/estimate their possible origin?
Finally, to confirm the results with advances microphysics inversion applied in a 2-dim height/time space, and yet be able to provide consistent microphysical parameter retrieval?

(Note that the latter is the topic of the related part 2 paper. The first results of microphysical retrieval show promising results ( paper accepted for presenting at the next ILRC-2019).

The key statement of the objective in the introduction seems to beat page 2, lines 30-33, "The algorithms that deal with the inversion problem of micro-physical retrieval require an accurate aerosol layer selection and a high quality 3beta+ 2alpha optical data set as well as the depolarization information [Veselovskii et al.2002; Bockmann et al. 2005; Muller et al. 2016]. Thus the manual data evaluation which allows for an insightful analysis of the lidar signals with the individual approach to the considered case is still much-needed."
In this study, the high quality 3beta + 2alpha optical data set and the QA on that dataset is provided by the retrieval algorithm which is considered to be an input to this study and not part of the study itself.

The algorithm is published in ACP by Baars et al 2016.
The QA&QC of the EARLINET-ACTRIS are in place for all data profiles obtained.
We doubt repetitions are necessary, however we will comment on both in the revised text.

Similarly, although the layer selection is done here, the explanation of how it's done is presented as a black box.

Thank you, this is very important point, especially as the layers in our work are not defined in a classical approach (not based on gradients as marking the layers borders).
The explicit layer selection work-flow applied in our work is added in the revised version.

The statement is a non-sequitur if the "insightful analysis" is taken to mean the aerosol typing analysis, since that is not needed for microphysical retrievals.

In the fragment Referee cites above we spoke of *aerosol layer selection* - not interpretation, nor typing. We shall rephrase, as this seem is not clear to the reader.

I also don't understand why automated aerosol typing algorithms are acknowledged but discarded.

Well, we acknowledge the automated aerosol typing algorithms them as professional and great software tools. They are not disregarded, as they will be likely used in the future.
Indeed, we do not aim at coming up with another typing-algorithm ourselves but we are looking for collaboration! In part 2 paper, we explore microphysics inversion of this data set but we are willing to consider also an automated aerosol typing (and this, not as an option, but as a must). Thank you.

In short, I just don't understand the motivation for doing this study this way.

We apologize for not being able to convince Referee to the proposed analyses approach in the initial version of the manuscript. On the positive site, at least potentially nothing can stop Referee from performing this study in her/his way. We are keen to openly publish or provide directly all raw data files (additionally to the profiles stored in the ERLINET ACTRIS Data Base), so that the Referee can perform analyses.

A less important point is that there is a lot of information in the introduction, such as the first several paragraphs about the effect of aerosol properties on radiative forcing, whose relevance to this study is not explained at all.

Thank you, we removed from Introduction all comments/statements which are indirectly related, thus irrelevant to this paper.

On a positive note, I appreciate the introduction to the specific heat wave event being studied, and also the statement about the uniqueness of the dataset.

Thank you, in the revised version we strongly focused on including only relevant information and commenting on why we include it in the sense of supporting objectives and motivation of this study.
* * *
Section "Methodology":

The methodology describes the retrieval methodology for obtaining the measurements of e.g. backscatter and extinction but not the analysis methodology.
Most of the analysis in this paper is about aerosol type attribution, but there is no description of the methodology for this (here or anywhere in the paper), leaving me completely without a foundation for understanding the rest of the paper.

Thank you, we added the analysis methodology work-flow diagram.
We also improved the Reference list and added more descriptive information in the aerosol attribution Table.

For a few other key aspects of the analysis, the hints in the methodology are inadequate, for example "The sub-layers selection was based mainly on RH and extinction profiles". This is too vague to be useful for understanding how the layers are selected.

This sentence has been rephrased to: 'The definition of the sub-layer in this paper is not standard as it is not based on the typically used gradient of the signal, e.g. ref. The sub-layers were discriminated to fulfill the quality requirements to further microphysical retrieval (part II paper). The sub-layer selection is appropriate for such inversion if the extensive optical properties (α, β) follow the same tendency at each wavelength (i.e. constant, increasing or decreasing).
In the sub-layer discrimination, the first step was to select the constant sections in the relative humidity profile and compare them with the extinction coefficient profiles. In the next step, the correspond to the backscatter coefficients profiles as well as to the sub-layers visible in the depolarization ratio were checked and corrected if necessarily. Examples of how the sub-layers were selected are shown on the profiles in Fig. 5 and in the Appendix A.

Finally, the sub-layers were collected into groups (layers) characterized by similar properties and described in the statistical way by the mean values of the intensive optical properties and the relative humidity with the standard deviation. The layers were discriminated based on the relative humidity and depolarization plots, as the differences between the layers were the most pronounced using these properties. Then, the first choice was revised with the spread between the properties using the plots of mutual relations of intensive optical properties and relative humidity shown in Appendix B. If particular layer had too much noise (especially upper layers) it was no further analyzed.'.

Also, I think it is important to understand the uncertainties on the lidar measurements.
Page 5, line 30 says the uncertainties for the intensive parameters are propagated from the backscatter and extinction uncertainties, but does not say how the backscatter and extinction uncertainties are calculated, and anyway, it doesn't seem that the uncertainties are actually presented in the paper. Error bars in the figures and tables seem to be just the standard deviation calculated for selected layers, not measurement uncertainties. Standard deviation might be a reasonable stand-in for measurement random uncertainties, but for some of the analysis, such as understanding quite small particle depolarization values, the authors need to develop an understanding of the expected systematic uncertainty.

Indeed, in the initial version the measurement uncertainties were not discussed. Although in the Appendix B the uncertainties for all values plotted in scatter plots are given with uncertainty. Only the mean values (right hand side plots) show standard deviations.
Section on uncertainties of lidar measurements has been added, as recommended.

We do not really understand why Referee thinks it is necessary to comment on *quite small particle depolarization values*. The values of depolarization obtained in our study align in the possible ranges of depolarization values reported for different aerosol types, as listed in the extensive table based on the most recent literature review in Nicolae et al. (2018).

The statement "The columnar value of the extinction related Angstrom exponent of 1 was assumed for the extinction coefficient profiles calculations" confuses me. I don't understand why you would need this assumption and it seems that this assumption would have too much impact on the layer angstrom exponents. I think I'm probably just misunderstanding what this means. I would like to be able to read a reference about the retrieval to be sure, but I can't find one cited. Did I miss it?

We clarified this in the revised manuscript. The AE=1 is recommendable and applied in the classical Raman approach introduced by Ansmann et al. (1990). The uncertainty related to this assumption is assessed based on the sensitivity study (for AE threshold values of 0 and 3). This is not a dominating source of uncertainty, more important is SNR (for extinction coefficient and RH) and ref. height (for backscatter coefficient and depolarization ratio).
The sources of the measurement uncertainties are addressed in the revised version.
* * *
Section 4.3 "Aerosol source analysis"

On first read-through of the paper, I thought the aerosol source analysis was meant to be a major part of the analysis. There is a significant amount of analysis to calculate and collate the backtrajectories in an attempt to identify source regions, and this could potentially be very informative in the inference of aerosol type and source for the observed layers.

Thank you, this is what we actually aimed at: an independent analysis of the source regions. Backward trajectories were calculated at regular height intervals at the beginning /end of each night. Then we constrained the trajectories with possible aerosol sources (satellite data). It was necessary to proceed this way, as the aerosol sub-layers in our case are not defined based on the search for significant gradients in the profiles. Then, the individual layers were (one by one!) *connected* to the transport/source origin based on height-space apportioning.

However, the analysis stops far short of that. There is very little attempt to link the backtrajectories to specific layers and the two segments of analysis (aerosol source analysis vs. aerosol typing) seem to be separate, disjointed sections.

As said above, there was careful apportioning performed. In the revised version of the manuscript we put more effort to explicitly state the link between each layer vs aerosol source, i.e. in Fig. 6 (layers) we add legend relating them to Fig.4 (aerosol source), using the same nomenclature and in Table 1.

There are also several confusing or contradictory statements.
For example, page 8, lines 14-15 say where two-days old smoke would have come from but then lines 17-19 says that above 3 km, it's about 3 days old and below 3 km, it's older; so where is the 2-days old smoke?
Then at line 20 is a statement that aged biomass burning aerosol is possible in the lowermost levels but at line 22, it's "moderately fresh".
I'm not sure if all of this is actually contradictory or if the authors are expressing the thought that there are multiple possibilities and a lot of uncertainty.

Well, yes. We were suggesting that there are multiple possibilities for interpretation. We took wrong approach – we wanted to point out all possible explanations and then argue to eliminate the less likely. As this seem confusing to the reader we revised these sentences and left unequivocal interpretation (or none).

If there are any layers at all where the airmass analysis allows for assessing probable sources or a range of ages of the aerosol layer, it would be very helpful to see that information included explicitly in Table1 where it can be easily used for understanding the aerosol types.

Thank you for this suggestion, it has been added as suggested.
* * *
Section 5 "Optical properties of the BBA and pollution mixtures"

In this section, individual aerosol layers are labeled according to aerosol types. In a few cases, the backtrajectories are referred to, but mostly the methodology seems to be based on haphazardly comparing the aerosol intensive parameters to a few case studies in prior literature.

After adding the methodology flow it is much clearer. Also addition of the respective legends and more descriptive information in Table 1 helps out. In fact trajectories were related to each sub-layer. Thank you.

There is no discussion of how exactly this is done, no consistent thresholds are presented, and there is no quantitative comparison with prior literature that present ranges of parameters that might be expected for different types (e.g. Muller et al. 2007, Gross et al. 2012, Burton et al. 2012).

As a manual interpretation of data and not an automated aerosol typing is applied in this study, therefore there is/was no intention to *fix thresholds*. However, indeed the quantitative comparisons with prior literature can be improved, and so was done in the revised version, and special case was sent on referring to work of Müller et al. (2007), Groß et al. (2012), Burton et al. (2012).

Unfortunately, there are many, many examples of type labels being applied without adequate explanation, support, or comparison:

Page 9, line 29: "properties typical for the aged biomass burning aerosol (CRLR >1)".No support is given for this value being "typical" for aged biomass burning aerosol. Although it is not cited in reference to this statement, I believe this argument follows from the case studies presented by Nicolae

et al. (2013). However, since those are just a few case studies, it may be an overgeneralization to describe this relationship as "typical".

*Thank you, references are added, sentences rephrased, wording for* typical *is no longer used.*

Page 10, line 22: "The relatively low LR and high AE...indicates pollution domination. " No reference is given for LR and AE of pollution or a description of how to determine them.

*Thank you, references are added, sentences rephrased, discussion is elaborated.*

Page 10, line 22: The low lidar ratios that are said to indicate pollution seem very low compared to prior literature that I know of. No references are given here. How do these values compare to other published values for pollution cases?

*Thank you, references are added, sentences rephrased, results are discussed against literature.*

Page 10, line 24, "relatively high value of 532 nm particulate depolarization of 4.8 plus or minus 0.3 ... may reflect contamination by pollen". This is not a very high value of particulate depolarization, and is only for the earlier layer (L1b), and the larger AE for the earlier layer would suggest smaller particles rather than larger. These seem like subtle and confusing trends. Is it clear that these distinctions are actually significant compared to measurement uncertainty (random + systematic)? Also, why is the conclusion that pollen is present, and not some other depolarizing type such as dust? 4.8% is small enough that even smoke could have such a value. The Sicard reference cited here does not help me understand this, since it's not about this case, but is just a general reference for lidar measurements of pollen.
(And a minor related point: if the purpose is to credit lidar measurements of pollen, much earlier papers such as Sassen et al. 2008 should get that credit, doi: 10.1029/2008gl035085).

*Thank you, references are added, sentences rephrased.*
*Special care was dedicate to more discussion of this dry polluted layer with only a slight contamination of the residual pollen particles, which were observed during pollination event at daytime of 10 August 2015). For pure pollen values of 6-12% over Warsaw are reported. Obtained here 4.8% is attributed to pollen contamination of polluted layer. Pollen contamination is more likely than dust/smoke, as for this layer there is evidence of local origin no long-range transport.*
*As suggested, the reference to Sassen et al. (2008) is added and the reference to Sicard et al. (2018) is removed.*

Page 10, line 26-29 Contradictory statements that the ratio of particulate depolarization at two wavelengths "did not vary significantly" but the difference between them "confirms the hypothesis of pollen contamination". If the variation is not significant, it doesn't confirm the hypothesis.

*Thank you, references are added, sentences rephrased. We apologize for misleading phrasing.*
*In fact, what was meant is that values are low, being lowermost for pollution layer contaminated with residual pollen particles.*

Page 10, line 31. Higher lidar ratio values "indicate domination by the biomass burning aerosol." Again, based on what methodology and what references? All the papers I'm familiar with (e.g. Muller et al. 2007, Gross et al. 2012, Burton et al. 2012,Papagiannopoulos et al. 2018) have a significant overlap for the lidar ratio of biomass burning and pollution, at whichever wavelength they look at.

*Thank you, suggested references are cited, sentences rephrased, comparability of results addressed.*

Page 11, line 1, imaginary refractive index values of 0.12i or 0.20-0.30i seem extremely high. This is quote from a paper in preparation. It doesn't seem appropriate to present something so unexpected without any support or discussion.

Thank you, as suggested these sentences were removed, note there was typo in values of the imaginary part of the refractive index (order of magnitude).

Page 11, line 5, "as the LR values were relatively high of 81 plus or minus 6 sr and 60 plus or minus 6 sr at 355 nm and 532 nm, respectively, the layer can be treated as fresh biomass burning aerosol...although the angstrom exponent value of 1.34 plus or minus 0.3 is rather low for fresh BBA". How is this conclusion reached? What values are these measurements being compared to? Couldn't this be urban pollution?

Thank you, references added and discussed, sentences rephrased. It is unlikely pollution, as trajectories indicate possible smoke. Comparison with values reported in literature is added.

Page 11, lines 15-16 "Such low depolarization ratios are characteristic for aged biomass burning". No references are given to support this statement. Aren't low depolarization ratios also characteristic of pollution and other aerosol types?

Thank you, references are added, sentences rephrased. In Warsaw, for boundary layer pure pollution the lidar ratios below 1% are reported, this is why slight contamination with smoke cannot be excluded.

Page 11, line 21. "The AE value of about 0.6 [i.e. the same value as measured in this study] was reported by Muller et al. 2007". Actually, Mueller et al. 2007 reported a value of 1.0 plus or minus 0.5. Maybe that could be considered "about 0.6" but it's misleading way to say it.

Thank you, sentences were rephrased.

Page 12, line 4, appears to suggests that the backscatter angstrom exponent is expected to be monotonically related to the aerosol age. I don't know of any reason to think this. Are there references?

We are not sure which sentence is meant here.
Muller et al. 2007, related the extinction-AE to actual advected BBA age.
Now, CRLR can be used as indicator of the BBA age. In Appendix B there is a relation of extinction-AE versus CRLR, which indicates a negative trend of the two, i.e. for aged BBA particle size is larger, and vice versa.

This section also contains statements two pages apart which directly contradict each other, giving me the impression that the authors do not fully understand their subject material and this manuscript was really not ready for submission in the first place. Regarding an apparent correlation between backscatter and depolarization, on page 11,"it is probable that increased depolarization is associated with the higher participation of background aerosol in the tropical air containing some mineral dust particles." And on page 13, "the negative dependence of backscatter and particulate depolarization is unlikely related to mineral dust contamination".

No, there is no contradiction, its's misleading writing… The interpretation of the described relation was assess at first as due to possible dust contamination, but then this hypothesis was disregarded in favor of another explanation – water uptake by the particles. Thus, the apparent contradiction. We revised this in the final version and kept only one possible explanation; the latter.

Another question about the measurements.
On page 11, line 28, a lidar ratio of 114 plus or minus 33 sr at 355 nm. Isn't this really very high? Are there other published observations of 355 nm lidar ratios this high? I see from the figure that this very high value occurs where there is significant slope in the backscatter and extinction profiles, on the underside of a layer with larger backscatter. The peak of the backscatter and extinction have very

different shapes, with the extinction looking like a smoothed version of backscatter. If there is a difference in the resolutions of the backscatter and extinction profiles such that the edges of the layers are not being captured the same way, it creates spurious values in the ratio. In this case, it looks like the lidar ratio is large in consequence of the fact that the backscatter and extinction show the edge with different slopes. Are the backscatter and extinction resolutions compatible when the lidar ratio is computed?

Note that all available profiles in the EARLINET data base are contained in so called e and b files, which by definition are stored with different resolution and/or smoothing applied to the data (they contain: e- extinction and b- backscattering and depolarization). This is a standard EARLINET procedure. The EARLINET Warsaw site is one of a few that provide in the b-file not only also extinction (all in the same resolutions!).
Now, according to Referee comment, it seems essentially not possible to derive correctly lidar ratios from profiles stores in this data base. Taking to account vast literature based on the EARLINET data base (e.g. Mattis et al. (2004); Matthias et al. (2004); Papayannis et al. (2008); Papagiannopoulos et al. (2018)) using the LR obtained with different resolutions... I have difficulty to agree with this comment.

Secondly, all of the profiles we calculated are shown in the Appendix A, and looking at these We can easily find the ranges for which even for high slope (in B and A) we get low values of LR. This is why it is difficult to agree that such slope shall *in general* causing large LR (we have large values in only 5-7 layers).

Moreover, LR values of > 100 sr are not impossible, the EARLINET data base threshold for LRmax is at 150sr. In the present study only 4 values are ranging >100 sr. Also, such high LRs were discussed as possible to exist, e.g. during the ILRC-2017 Summer School (lecture course by known lidar expert Prof. A.Pappayannis. Thus, we should not exclude such values for the analyses.

Now, as for the slope itself – in our case it is not much of a slope for the 4 values of high LRs (red points in Fig.6), they also show temporal dependence (likely due to aging, as it is the same aerosol arriving at the site in this layers). If you like, there is a strong temporal change in these high LR (> 100sr) values, if one explore individual values as given on the Appendix A plots (the high LR is becoming lower with time during the night). We shall doubt that this is a coincidence or an artefact.

We regret we did not stress this well enough in the initial manuscript. Note that a sensitivity study was performed to assess if LRs are not overestimated, whereby discussion on backscatter and extinction coefficient calculated with the same resolutions vs applied smoothing to extinction and its effect on resulting LR was addressed. For the presented case the high values of LR are obtained for both cases.

The wording is quite rough. If this manuscript were to be published or resubmitted, it would be very helpful to have a round of English-language editing.

Thank you, obviously neither of us is English native speaker. We apologize for ignoring this fact, and not sending the initial manuscript for language proof. The resubmitted version of the manuscript will be sent for professional English proof.

Finally, where are the data available? Given that the paper is primarily about introducing a new dataset, I would like to see a statement of data availability.

Thank you for this comment, all of the derived data sets are already stored in the EARLINET-ACTRIS Data Base as (b and e files). The RH is currently not therein but it is available via PolandAOD (www.poland aod.pl).
We will extra explore the possibility to add the RH profiles to the EARLINET b-files category instead of volume depolarization profiles. If we obtain agreement form the AQ&AC team, we will add the full sets of profiles to the data base.